# Microbiota-induced T cell plasticity enables immune-mediated tumour control

Tariq A. Najar[1], Yuan Hao[2,3], Yuhan Hao[4,5], Gabriela Romero-Meza[1,6], Alexandra Dolynuk[1,6], Emma Almo[1,6] & Dan R. Littman[1,3,6 ✉]

Therapies that harness the immune system to target and eliminate tumour cells have revolutionized cancer care. Immune checkpoint blockade (ICB), which boosts the anti-tumour immune response by inhibiting negative regulators of T cell activation[1–3], is remarkably successful in a subset of cancer patients. Yet a significant proportion do not respond to treatment, emphasizing the need to understand factors influencing the therapeutic efficacy of ICB[4–9]. The gut microbiota, consisting of trillions of microorganisms residing in the gastrointestinal tract, has emerged as a critical determinant of immune function and response to cancer immunotherapy, with several studies demonstrating association of microbiota composition with clinical response[10–16]. However, a mechanistic understanding of how gut commensal bacteria influence the efficacy of ICB remains elusive. Here we use a gut commensal microorganism, segmented filamentous bacteria (SFB), which induces an antigen-specific T helper 17 ($T_H$17) cell effector program in the small intestine lamina propria (SILP)[17], to investigate how colonization with this microbe affects the efficacy of ICB in restraining distal growth of tumours sharing antigen with SFB. We find that anti-programmed cell death protein 1 (PD-1) treatment effectively inhibits the growth of implanted SFB antigen-expressing melanoma only if mice are colonized with SFB. Through T cell receptor (TCR) clonal lineage tracing, fate mapping and peptide–major histocompatability complex (MHC) tetramer staining, we identify tumour-associated SFB-specific T helper 1 ($T_H$1)-like cells derived from the homeostatic $T_H$17 cells induced by SFB colonization in the SILP. These gut-educated ex-$T_H$17 cells produce high levels of the pro-inflammatory cytokines interferon (IFN)-γ and tumour necrosis factor (TNF) within the tumour microenvironment (TME), enhancing antigen presentation and promoting recruitment, expansion and effector functions of CD8[+] tumour-infiltrating cytotoxic lymphocytes and thereby enabling anti-PD-1-mediated tumour control. Conditional ablation of SFB-induced IL-17A[+]CD4[+] T cells, precursors of tumour-associated $T_H$1-like cells, abolishes anti-PD-1-mediated tumour control and markedly impairs tumour-specific CD8[+] T cell recruitment and effector function within the TME. Our data, as a proof of principle, define a cellular pathway by which a single, defined intestinal commensal imprints T cell plasticity that potentiates PD-1 blockade, and indicate targeted modulation of the microbiota as a strategy to broaden ICB efficacy.

Although specific bacterial taxa have been associated with favourable clinical responses to immune checkpoint blockade (ICB) in cancer patients[12,13,18–22], the mechanisms by which the intestinal microbiota influences anti-tumour immune responses remain poorly defined. Products of the microbiota, including metabolites[23–25] and innate receptor ligands[26], may reprogramme myeloid cells[27], lowering the activation threshold for antigen presentation and thereby facilitating priming and activation of tumour-reactive T cells. Alternatively, T cells that recognize antigens shared between commensals (microbial-associated antigens (MAAs)) and tumours (tumour-associated antigens (TAAs)) may become activated in the setting of ICB, thus enhancing anti-tumour immune responses. Because the gut microbiome encodes an enormous antigenic repertoire, commensal-derived antigens can elicit T cell responses that, in some cases, cross-react with tumour epitopes—a plausible mechanism for commensal-driven tumour control[28]. Despite correlative clinical data[29], the causality of such antigenic mimicry has

[1]Department of Cell Biology, New York University School of Medicine, New York, NY, USA. [2]Division of Advanced Research Technologies, Applied Bioinformatics Laboratories, New York University School of Medicine, New York, NY, USA. [3]Perlmutter Cancer Center, New York University Langone Health, New York, NY, USA. [4]Center for Genomics and Systems Biology, New York University, New York, NY, USA. [5]New York Genome Center, New York, NY, USA. [6]Howard Hughes Medical Institute, New York, NY, USA. ✉e-mail: dan.littman@med.nyu.edu

not yet been demonstrated definitively in vivo. It is possible, however, to test this premise, particularly the relationship of microbe-specific T cells and intratumoural T cells, in animal models. An immunization model with skin-associated *Staphylococcus epidermidis* engineered to express antigens shared with implanted tumours was shown to elicit effective anti-tumour responses[30], but the ability of gut commensals that elicit stereotyped T cell responses to program anti-tumour immunity has not been explored. Here we studied how a small intestine-resident commensal microbe, SFB, which induces a regulatory-like T helper 17 ($T_H17$) cell response that enhances intestinal barrier integrity[31,32], influences efficacy of ICB in controlling growth of distal tumours that share antigen with the bacterium. We found that tumour-specific $T_H17$ cells, primed by SFB in the gut, infiltrate the tumour as trans-differentiated pro-inflammatory T helper 1 ($T_H1$)-like cells following ICB. These cells remodel the tumour microenvironment (TME), promoting recruitment, expansion and maturation of $CD8^+$ effector T cells that contribute critically to anti-tumour immunity. Our results indicate that defined constituents of the intestinal microbiota can be harnessed to elicit desired effector T cell programs that restrain tumour growth.

## SFB promotes ICB-mediated tumour control

To explore how gut commensal microbiota influence immune-mediated tumour control, we developed a synthetic neoantigen mimicry tumour model in mice by engineering B16-F10 (B16-3340) melanoma cells to express an immunodominant protein fragment of the gut-colonizing commensal microbe SFB (Fig. 1a,b). We chose SFB because it reliably colonizes specific-pathogen-free (SPF) mice and elicits a robust, well-characterized $CD4^+$ T cell response that can be tracked with peptide-MHCII tetramers and TCR-transgenic mice[17,33,34]. Lysates from B16-3340, but not from empty vector controls (B16-EV), robustly activated TCR-transgenic T cells ($TCR^{7B8}$) ex vivo, demonstrating effective processing and presentation of the SFB-derived epitope (Fig. 1c).

Next, we examined the effect of SFB colonization on tumour growth in SPF mice (Jackson Laboratories) bearing subcutaneously implanted B16-3340 and B16-EV tumours, with (Fig. 1d) or without (Extended Data Fig. 1a) anti-programmed cell death protein 1 (PD-1) treatment. In the absence of PD-1 blockade, there was no notable difference in tumour growth between mice implanted with either B16-3340 or B16-EV, regardless of SFB colonization status ($SFB^+$ or $SFB^-$) (Extended Data Fig. 1b,c). However, when animals were treated with anti-PD-1 antibody, the growth of B16-3340 tumours was markedly reduced in $SFB^+$ mice compared with $SFB^-$ mice. There was no notable difference in the growth of control B16-EV tumours between $SFB^+$ and $SFB^-$ mice receiving anti-PD-1 treatment (Fig. 1e–g and Extended Data Fig. 1d). The combination of SFB colonization and anti-PD-1 treatment of B16-3340 tumours also conferred a survival advantage compared with the other groups (Fig. 1h). Mice that survived the primary B16-3340 challenge subsequently rejected tumour re-challenge even without additional anti-PD-1 treatment, demonstrating durable, memory-like protection mediated by SFB colonization in concert with ICB (Fig. 1h). This robust SFB-dependent enhancement of anti-tumour immunity prompted further investigation into the underlying mechanisms by which SFB modulates tumour-directed immune responses and potentiates the efficacy of ICB.

To test whether SFB can act therapeutically to enhance anti-PD-1 efficacy after tumour establishment, we performed staged post-implantation gavage experiments (Extended Data Fig. 1e). Mice bearing B16-3340 were treated with anti-PD-1 (days 4–10) and gavaged with SFB at defined intervals (days 8–12, 12–16 or 15–19) or left SFB-free. Early gavage (days 8–12, group 1) produced the largest reduction in tumour growth, with progressively diminished benefit for later administrations (groups 2 and 3) (Extended Data Fig. 1f), indicating a narrow post-implantation window in which microbial antigen exposure most

effectively synergizes with PD-1 blockade. These data further show that tumour expression of the SFB-derived neoantigen is required for microbiota-dependent augmentation of anti-PD-1 and that therapeutic SFB colonization is most effective when delivered early.

We next determined whether antigenic mimicry promotes microbiota-mediated control of additional tumours, extending our study to Lewis lung carcinoma (LLC1-3340) and MC-38 colon adenocarcinoma (MC-3340). SFB-colonized ($SFB^+$) or SFB-free ($SFB^-$) C57BL/6J mice were implanted subcutaneously with these engineered tumour cells and treated with anti-PD-1 beginning at the earliest stage of palpable tumour growth (Extended Data Fig. 1g,i). In both tumour models, $SFB^+$ mice exhibited substantially delayed tumour growth compared with $SFB^-$ cohorts (Extended Data Fig. 1h,j).

## SFB alters tumour T cell profile

Profiling of T cells from B16-3340 and B16-EV tumours in anti-PD-1 treated $SFB^+$ and $SFB^-$ mice showed that SFB colonization significantly increased the intratumoural $CD8^+$ to regulatory T ($T_{reg}$) cell ratio compared with either $SFB^-$ mice with B16-3340 tumours or $SFB^+$ mice with B16-EV tumours subjected to anti-PD-1 treatment (Fig. 2a,b). In contrast, SFB colonization did not alter $CD8^+$ to $T_{reg}$ cell ratio in the small intestine lamina propria (SILP) of anti-PD-1 treated mice (Extended Data Fig. 2a). Concurrently, $CD8^+$ tumour-infiltrating lymphocytes (TILs) from anti-PD-1 treated, B16-3340 tumours from $SFB^+$ exhibited markedly enhanced effector functions, with higher frequencies of $IFN\gamma^+$, $TNF^+IFN\gamma^+$ and $Gzm-B^+TNF^+$ $CD8^+$ TILs compared with either $CD8^+$ TILs in anti-PD-1 treated B16-3340 tumours in $SFB^-$ or B16-EV tumours in $SFB^+$ mice (Fig. 2c and Extended Data Fig. 2b). Similar results were demonstrated previously in the microbiota-mediated response to PD-1 blockade[35].

Next, because SFB colonization induces antigen-specific $T_H17$ cells in the ileal lamina propria, we compared the $CD4^+$ T cell phenotypes in the remotely located B16-3340 tumours. First, using a panel of antibodies specific for TCR Vβs, we found a greater proportion of $V\beta14^+$ $CD4^+$ T cells in tumours from $SFB^+$ compared with $SFB^-$ mice (Extended Data Fig. 2c). This bias is consistent with the known preferential interaction of this subset of TCRs with immunodominant SFB peptides in the SILP of SFB-colonized mice[36] (Extended Data Fig. 2d). Second, using SFB-3340 peptide-loaded MHCII tetramers revealed that colonization with SFB caused increased infiltration of SFB-3340-specific $CD4^+$ T cells into B16-3340 tumours (Fig. 2d). Remarkably, tumour-resident tetramer$^+$ $CD4^+$ T cells displayed an IFNγ producing $T_H1$-like phenotype, unlike SILP tetramer$^+$ T cells that, as expected, were IL-17A producing $T_H17$ cells (Fig. 2e–h and Extended Data Fig. 2e–g). Although tumour-resident tetramer$^+$ $CD4^+$ T cells in both $SFB^+$ and $SFB^-$ mice were T-bet$^+$, a minor fraction from $SFB^+$ mice co-expressed RORγt, consistent with gut imprinting (Fig. 2e and Extended Data Fig. 2g). By contrast, the bulk of tetramer$^-$ $CD4^+$ T cells in both B16-3340 and B16-EV tumours, irrespective of SFB colonization, were regulatory-like T cells, expressing both T-bet and Foxp3 (Extended Data Fig. 2h,i), a phenotype associated with strong suppression of anti-tumour immune responses[37].

ELISpot assays confirmed significant enrichment of IFNγ-producing $CD4^+$ TILs in $SFB^+$ B16-3340 tumours compared with $SFB^-$ cohorts (Extended Data Fig. 3a) and, accordingly, tetramer$^+$ $CD4^+$ TILs from those tumours exhibited a robust $T_H1$ cytokine (IFNγ and TNF) response following ex vivo stimulation (Extended Data Fig. 3b). Furthermore, whereas the frequencies of both T-bet$^+$Foxp3$^-$ and T-bet$^+$Foxp3$^+$ cells in the tetramer$^-$ $CD4^+$ T cell population were comparable across groups (Extended Data Fig. 2h,i), B16-3340 tumours in $SFB^+$ mice contained a significantly higher fraction of tetramer$^-$ $CD4^+$ T cells that produced moderate amounts of IFNγ and TNF following ex vivo stimulation compared with either B16-3340 tumours in $SFB^-$ mice or B16-EV tumours in $SFB^+$ mice (Extended Data Fig. 3c,d). Together, these findings suggest that SFB-specific pro-inflammatory $CD4^+$ T cells contribute to

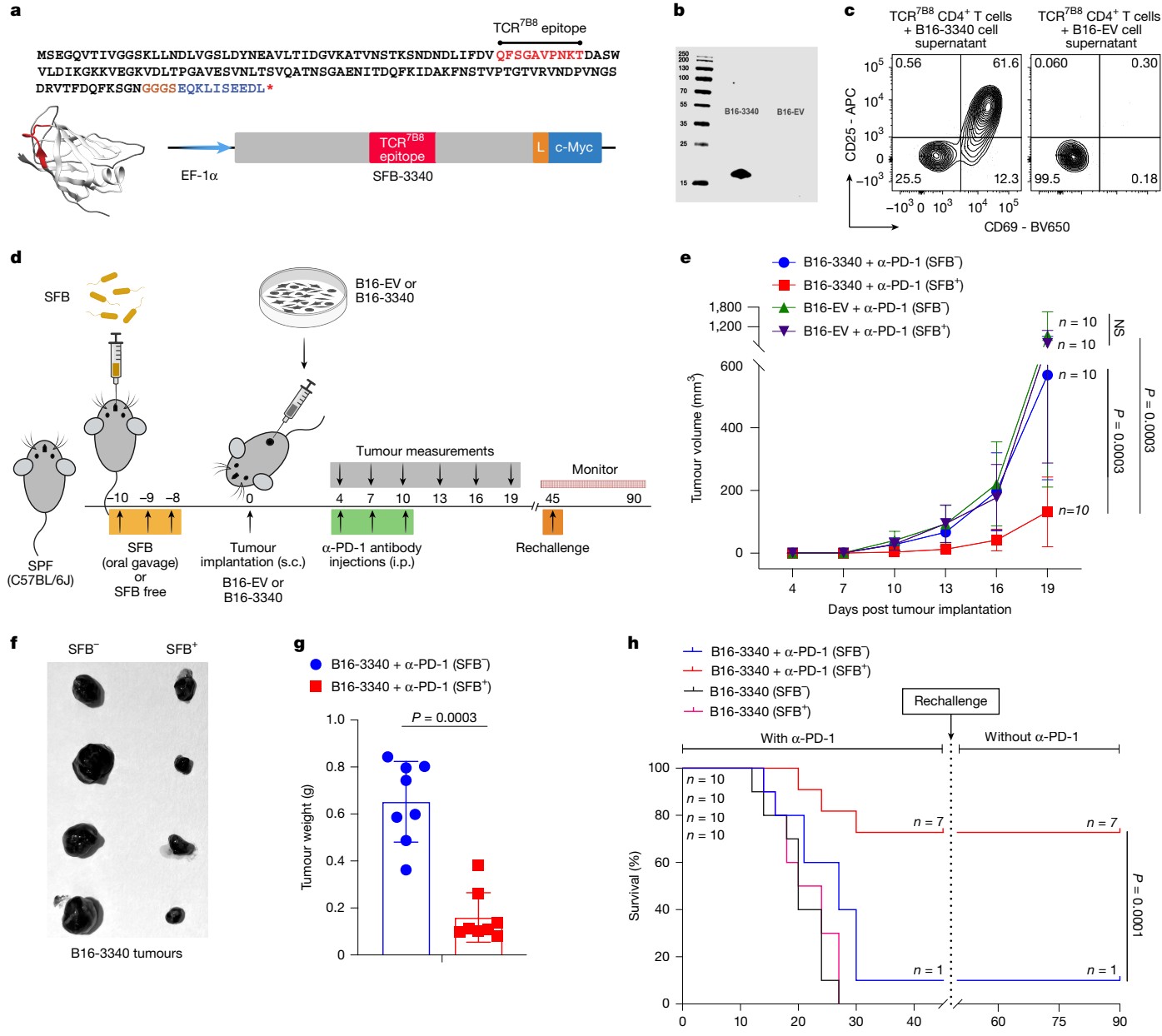

**Fig. 1 | Development of a synthetic microbiota-based tumour antigen mimicry model to evaluate response to anti-PD-1 therapy. a**, Amino acid sequence of the SFB-3340 protein fragment containing the CD4[+] T cell epitope recognized by TCR[7B8] (red). The codon-optimized gene was fused to an EF-1α promoter (5′) and c-Myc tag (blue, 3′) by a flexible linker (L). s.c., subcutaneous. **b**, Immunoblot showing expression of the SFB-3340 fragment in transfected B16-F10 (B16-3340) cells, detected with anti-c-Myc antibody. Empty vector transfected cells (B16-EV) served as control. Size markers (kDa) are shown on the left. **c**, Ex vivo activation of naive SFB-specific CD4[+] T cells from TCR[7B8] transgenic mice co-cultured with syngeneic splenocytes plus lysates from B16-3340 or B16-EV cells. Surface expression of activation markers CD69 and CD25 was analysed 24 h later by flow cytometry. **d**, Experimental design comparing SFB-colonized (SFB[+]) and SFB-free (SFB[−]) C57BL/6J mice implanted with B16-3340 or B16-EV tumours. **e**, Tumour growth curves from caliper measurements (*n* = 10 mice per group). Mice received anti-PD-1 antibody (250 µg per mouse, i.p.) on days 4, 7 and 10 post-implantation. Data represent mean ± s.d.;

significance determined by two-way analysis of variance (ANOVA) with Sidak's correction. **f**, Representative B16-3340 tumours excised from SFB[+] and SFB[−] mice on day 14 post tumour implantation. **g**, Excised tumour weights from SFB[+] and SFB[−] mice on day 14 (*n* = 8 mice per group); mean ± s.d., unpaired two-sided Mann–Whitney test. **h**, Kaplan–Meier survival curves of SFB[+] and SFB[−] mice (*n* = 10 mice per group) bearing B16-3340 tumours, with or without anti-PD-1 therapy. Following the initial challenge, surviving mice were re-challenged with the same tumour cells, and monitored without further anti-PD-1 antibody treatment. *P* values were determined by log-rank (Mantel–Cox) test. All experiments in **e**–**h** were repeated independently at least twice with similar results. In panel **a**, the ribbon-helix model was generated using AlphaFold2 to illustrate the predicted structure of the SFB-3340 protein fragment containing the TCR[7B8] epitope. This fragment was used to engineer cancer cell lines (B16-F10, MC-38 and LLC1) to stably express the SFB-3340 antigen, resulting in the generation of B16-3340, MC-3340 and LLC1-3340 cell lines. Schematics in **a** and **d** were created using BioRender (https://biorender.com).

remodelling the TME, thereby increasing its responsiveness to PD-1 blockade.

The T cell composition in the MC-38 tumours expressing the SFB antigen (MC-3340) was similarly altered, with SFB colonization promoting

accumulation of tetramer[+] CD4[+] T cells exhibiting a T-bet[+]IFNγ[+] $T_H1$-like program, and an enrichment of IFNγ[+]Gzm-B[+] CD8[+] TILs (Extended Data Fig. 3e–h). These data show that SFB-induced antigen-specific CD4[+] T cell priming and $T_H1$-like polarization, together with enhanced CD8[+]

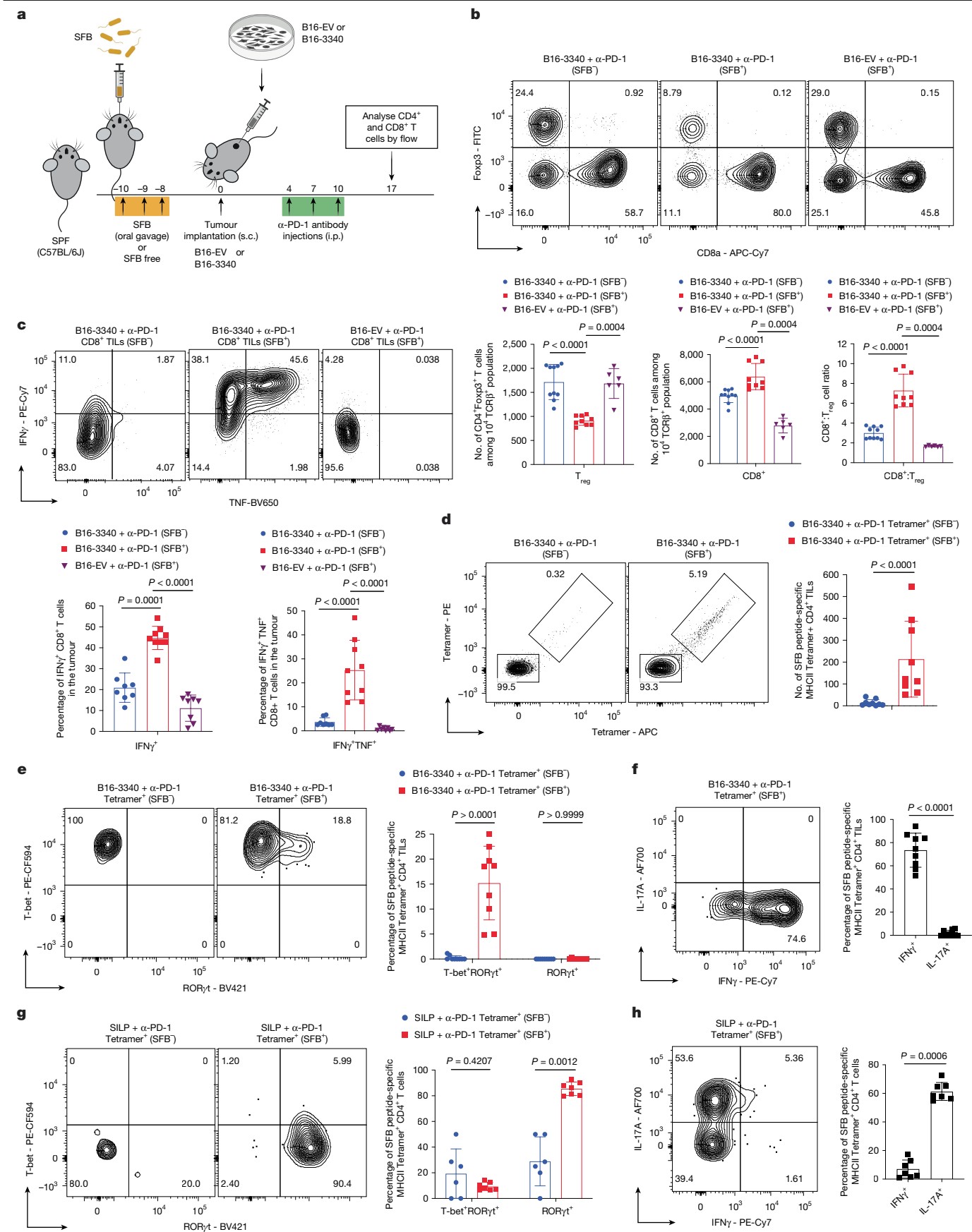

**Fig. 2 | See next page for caption.**

**Fig. 2 | SFB colonization modulates CD4⁺ and CD8⁺ T cell effector programs in antigen-expressing tumours. a**, Schematic of the synthetic mimicry model used to evaluate how gut SFB colonization alters the distal immune TME. **b**, Top, representative flow cytometry plots of tumour-infiltrating CD8⁺ T cells and Foxp3⁺ CD4⁺ T$_{reg}$ cells from B16-3340 (SFB⁻, $n$ = 10 mice and SFB⁺, $n$ = 9 mice) and B16-EV (SFB⁺; $n$ = 6) following anti-PD-1 treatment. Bottom, quantification of absolute counts of T$_{reg}$ cells, CD8⁺ T cells, and CD8:T$_{reg}$ ratio per tumour. **c**, Top, representative cytokine flow cytometry plots (TNF⁺IFNγ⁺) of CD8⁺ TILs from B16-3340 tumours in SFB⁻ and SFB⁺ mice, and B16-EV tumours in SFB⁺ mice. Bottom, frequencies of TNF⁺IFNγ⁺CD8⁺ TILs: B16-3340 (SFB⁻, $n$ = 8), B16-3340 (SFB⁺, $n$ = 9) and B16-EV (SFB⁺, $n$ = 8). **d**, Left, SFB-peptide-specific MHCII tetramer staining of CD4⁺ TILs from B16-3340 tumours (SFB⁻ versus SFB⁺). Right, absolute counts of tetramer⁺ CD4⁺ TILs per tumour ($n$ = 9 mice per group). **e**, Left, expression of RORγt and T-bet in tetramer⁺ CD4⁺ TILs from B16-3340 tumours of SFB⁻ and SFB⁺ mice. Right, frequencies of T-bet⁺RORγt⁺ and RORγt⁺ subsets ($n$ = 9 mice per group). **f**, Left, IFNγ and IL-17A expression in tetramer⁺ CD4⁺ TILs from B16-3340 tumours in SFB⁺ mice. Right, frequencies of IFNγ⁺ and IL-17A⁺ tetramer⁺ CD4⁺ TILs ($n$ = 9 mice per group). **g**, Left, RORγt and T-bet expression in SILP tetramer⁺ CD4⁺ T cells (SFB⁻ versus SFB⁺). Right, quantification of transcription factor expression in SILP tetramer⁺ CD4⁺ T cells from SFB⁻ ($n$ = 6) and SFB⁺ ($n$ = 7) mice. **h**, Left, IFNγ and IL-17A expression in SILP tetramer⁺ CD4⁺ T cells (SFB⁺). Right, frequencies of IFNγ⁺ and IL-17A⁺ SILP tetramer⁺ CD4⁺ T cells ($n$ = 7 mice per group). Data are mean ± s.d., each data point representing an individual mouse. Statistical comparisons were determined by unpaired two-sided Mann–Whitney $t$-test, with $P$ values indicated. All experiments shown were repeated at least twice with similar results. Schematic in **a** was created using BioRender (https://biorender.com).

T cell effector function, potentiate PD-1 blockade across melanoma, lung and colon tumour models when the tumour expresses the cognate microbial epitope.

## CD4⁺ and CD8⁺ TILs required for response

Next, given that the effective anti-tumour immune response requires synergistic cooperation of CD4⁺ and CD8⁺ T cells in ICB-mediated tumour control[38–40], we aimed to investigate whether the combination of SFB-induced CD4⁺ T cells and tumour-infiltrating CD8⁺ T cells, together with anti-PD-1 therapy, is essential for controlling B16-3340 tumour growth in SFB⁺ mice. In vivo depletion of either CD4⁺ (Extended Data Fig. 4a,b) or CD8⁺ (Extended Data Fig. 4a,g) T cells in B16-3340 tumour-bearing mice significantly impaired the efficacy of anti-PD-1 treatment (Extended Data Fig. 4c and 4h). CD8⁺ TILs from CD4-depleted, SFB⁺ mice exhibited a marked functional impairment, with significant reductions in T-bet⁺IFNγ⁺, TNF⁺Gzm-B⁺ and IFNγ⁺TNF⁺ cells relative to controls (Extended Data Fig. 4d–f), indicating that microbiota-dependent CD4⁺ T cells are critical for the acquisition of full CD8⁺ TIL effector function in tumours. Conversely, depletion of CD8⁺ T cells modestly reduced the proportion of T-bet⁺IFNγ⁺ CD4⁺ TILs, consistent with reciprocal but asymmetric cross-talk between these T cell lineages (Extended Data Fig. 4i,j). Together, these results demonstrate that SFB colonization enhances the efficacy of PD-1 blockade through a coordinated CD4–CD8 T cell axis: microbiota-induced, pro-inflammatory CD4⁺ T cells provide critical help for CD8⁺ TIL maturation and cytotoxic function, whereas both T cell subsets are jointly required for durable, SFB-dependent tumour control under anti-PD-1 therapy.

## Shared T cell clonality in gut and tumours

To examine the relationship of intestinal and tumour-infiltrating T cells in SFB⁻ and SFB⁺ mice with B16-3340 tumours, we performed paired single-cell RNA sequencing (scRNA-seq) and TCR repertoire analysis (scTCR-seq) on sorted CD4⁺ T cells from SILP and B16-3340 tumours (Fig. 3a). Unsupervised clustering of the scRNA-seq data resolved transcriptionally distinct CD4⁺ T cell subsets in each tissue (nine clusters in SILP and ten in B16-3340 tumours), defined by canonical lineage markers (Extended Data Fig. 5a). As anticipated, SFB colonization selectively expanded an IL-17A⁺ T$_H$17 cluster in the SILP (cluster 2) of SFB⁺ mice (Fig. 3b). In contrast, tumours from SFB⁺ mice were enriched for an IFNγ⁺ T$_H$1-like subset (cluster 1), a population absent from tumours of SFB⁻ mice (Fig. 3c), highlighting the divergent, tissue-specific programs of antigen-specific CD4⁺ T cells.

Analysis of paired TCR α and β chain transcripts revealed extensive clonal relationships of SILP CD4⁺ T cells with a T$_H$17 or follicular helper (T$_{FH}$) phenotype and B16-3340 tumour-infiltrating T$_H$1-like cells in SFB⁺ mice, supporting an intestinal origin of the trans-differentiated

CD4⁺ TILs (Fig. 3d–f). In contrast, SFB-free mice exhibited minimal clonal overlap between SILP and tumour (B16-3340) T cells, with most cells displaying a T$_H$1 phenotype in the SILP and a memory-like phenotype in the tumour (Fig. 3d, Extended Data Fig. 5b,c and Supplementary Table 1). In SFB-colonized mice, CD4⁺ TILs sharing clonotypes with SILP CD4⁺ T cells showed upregulation of genes associated with cell trafficking (such as *Cxcr6*), chemoattraction including *Ccl3* and *Ccl4* (potent chemo-attractants for various immune cells, including cytotoxic T cells, dendritic cells (DCs), natural killer cells and macrophages), pro-inflammatory cytokines (*Ifng* and *Tnf*) and cytolytic functions including *Prf1*, *Klrc1* and *Klrd1*, collectively promoting anti-tumour immunity (Fig. 3g and Extended Data Fig. 5d,e). A comparable cytotoxic CD4⁺ T cell subset has been identified across human cancers, including melanoma, breast, head and neck, and liver tumours[41], and a cytotoxic CD4⁺ T cell gene signature in bladder cancer has been associated with favourable responses to neoadjuvant anti-PD-L1 immunotherapy[42].

## SFB-specific TILs had expressed IL-17A

scTCR-seq identified a clonal relationship in SFB⁺ mice between SILP CD4⁺ T cells and CD4⁺ TILs, suggesting a potential migratory pathway. To validate whether SFB-elicited gut CD4⁺ T cells migrate to antigen-matched tumours and adopt a different effector fate, we combined fate mapping and adoptive transfer approaches in SFB-colonized, anti-PD-1 treated mice. This approach allowed us to specifically track the progeny of IL-17A-expressing SFB-specific T cells that migrate from intestinal lamina propria or mesenteric lymph nodes to distal tumour sites. Using IL-17A-GFP reporter mice, we first confirmed that CD4⁺ T cells in SFB⁺, anti-PD-1 treated B16-3340 tumours and the tumour-draining lymph node do not actively produce IL-17A, unlike small intestine cells (Extended Data Fig. 6a). We then used IL-17A-Cre mice bred to a reporter strain (*tdTomato-ON$^{ΔIL-17a}$* mice) to profile SFB-specific CD4⁺ T cells in the gut and distal B16-3340 tumours in SFB⁻ and SFB⁺ mice receiving anti-PD-1 therapy (Fig. 4a). In SFB-colonized reporter mice, a large fraction of intratumoural SFB-3340 tetramer⁺ cells and most Vβ14⁺ CD4⁺ T cells, were tdTomato⁺, indicating previous IL-17A expression. These cells were not detected either in B16-3340 tumours in SFB⁻ mice or B16-EV tumours in SFB⁺ mice (Fig. 4b,c). As expected, tdTomato⁺, tetramer⁺/Vβ14⁺ T cells were found in the SILP only in mice colonized with SFB (Extended Data Fig. 6b,c).

To further track migration, naive CD4⁺ T cells from TCR$^{7B8}$ *tdTomato-ON$^{ΔIL-17a}$* reporter donor mice were transferred adoptively into SFB-colonized B6 wild-type hosts. Three weeks later, B16-3340 tumours were implanted to track the migration of TCR$^{7B8}$ T$_H$17 T cells from the gut to distal tumour tissue (Fig. 4d). A substantial fraction (roughly 50%) of the tumour-infiltrating adoptively transferred TCR$^{7B8}$ T cells were ex-T$_H$17 (tdTomato⁺), consistent with gut-to-tumour migration. Within the tumour, a large percentage of these donor-derived ex-T$_H$17

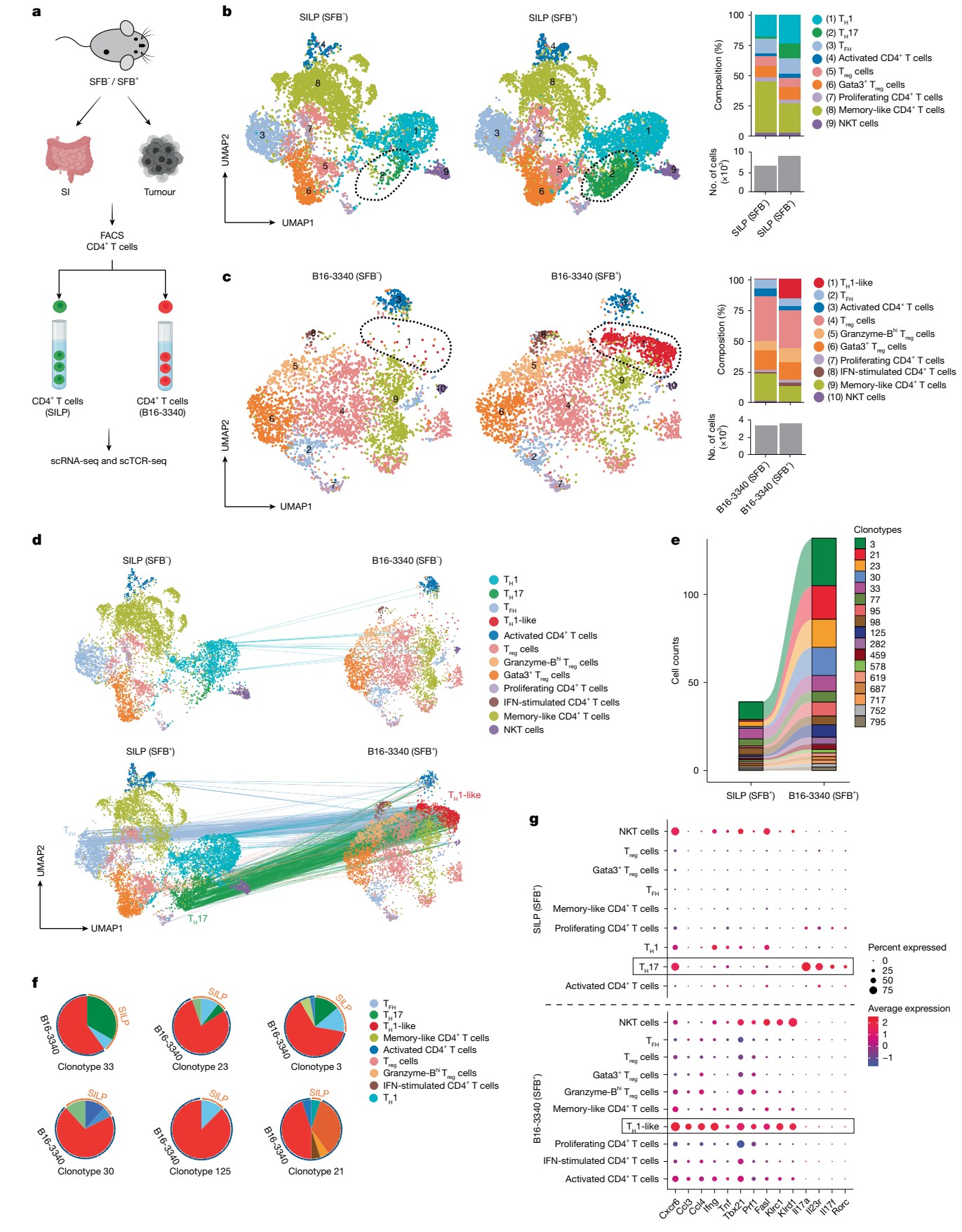

**Fig. 3** | See next page for caption.

**Fig. 3 | Shared TCR clonotypes in SILP and B16-3340 tumours of SFB-colonized and SFB-free mice. a**, Experimental workflow for scRNA-seq and scTCR-seq of CD4+ T cells from B16-3340 tumours and SILP of anti-PD-1 treated mice with and without SFB colonization. **b,c**, Uniform manifold approximation and projection (UMAP) embeddings of transcriptionally defined CD4+ T cell clusters from the SILP (**b**) and from B16-3340 tumours (**c**) in SFB⁻ (left) and SFB+ (middle) animals. Dotted ovals highlight cluster(s) of interest ($T_H17$ to $T_H1$-like transition). The distribution and total numbers of CD4+ T cell subsets is shown on the right. **d**, UMAPs illustrating the clonal connectivity between CD4+ T cells within the SILP and tumour tissues, providing a qualitative representation of their inter-relationship. **e**, Alluvial (stream) plot summarizing shared clonotypes between SILP and tumour in SFB+ mice. Each block represents a

distinct clonotype (classified by identical paired CDR3α/β sequences); branch widths indicate cell counts contributed by that clonotype to each tissue (SILP and B16-3340 tumour). **f**, Representative pie charts showing the phenotypic composition of select expanded clonotypes (shown by clonotype ID) in SILP versus tumour; colours denote transcriptionally annotated states. **g**, Dot plot of key differentially expressed genes across annotated CD4+ subsets; dot size shows percentage of cells expressing the gene in that cluster, colour scale shows mean expression (z-score). Boxes highlight genes that distinguish clonally related SILP $T_H17$ cells from tumour $T_H1$-like counterparts (for example, *Il17a, Rorc, Ifng, Tbx21*). Schematic in **a** was created using BioRender (https://biorender.com).

cells produced IFNγ compared with endogenous host CD4+ T cells (Fig. 4e). In contrast, adoptively transferred cells that remained in the SILP retained the $T_H17$ phenotype (tdTomato+) (Fig. 4f). Collectively, these results demonstrate that SFB-specific intestinal $T_H17$ cells migrate to distal, antigen-matched tumours, and trans-differentiate into $T_H1$-like effectors, thus mediating microbiota-driven enhancement of anti-PD-1 responses.

## $T_H17$ cells required for tumour control

To directly evaluate the contribution of SFB-induced IL-17A+ $T_H17$ cells to anti-PD-1 efficacy, we used a mouse model to conditionally deplete IL-17A-expressing cells in SFB+ mice[43]. *DTA-ON*$^{ΔIL-17a}$ mice (*Il17a-Cre; ROSA-LSL-DTA*) and control littermates were colonized with SFB, implanted with B16-3340 tumour cells and treated with anti-PD-1 antibody (Fig. 5a). Compared with *ROSA-LSL-DTA* controls, *DTA-ON*$^{ΔIL-17a}$ mice exhibited significantly impaired tumour control (Fig. 5b), demonstrating that SFB-induced, antigen-specific IL-17A+ $T_H17$ cells in the gut are required for therapeutic benefit.

*DTA-ON*$^{ΔIL-17a}$ mice had markedly reduced frequencies and absolute numbers of tetramer+ CD4+ T cells in the SILP, and near-complete ablation of IL-17A-producing antigen-specific T cells relative to *LSL-DTA* controls (Fig. 5c). Intratumoural tetramer+ CD4+ T cells were also reduced in *DTA-ON*$^{ΔIL-17a}$ mice compared with controls (Fig. 5d). In the SILP, Foxp3+ $T_{reg}$ and CD8+ T cell numbers were unchanged after depletion of IL-17A+ cells (Fig. 5e), but in tumours there were reduced CD8+ T cell and increased $T_{reg}$ cell frequencies (Fig. 5f). This was reflected in a pronounced deficit in the frequency of IFNγ+CD8+ TILs in *DTA-ON*$^{ΔIL-17a}$ mice (Extended Data Fig. 7a,b). Thus, SFB-induced ex-IL-17A+ effector T cells selectively support cytotoxic T cell recruitment and maintenance in the TME.

## *H. hepaticus* cannot control tumour growth

To test whether other gut commensals could augment ICB in an antigen-dependent fashion similar to SFB, we tested colonization with *Helicobacter hepaticus* (Hh), which induces $T_{reg}$ and $T_{FH}$ cells in the large intestine lamina propria (LILP) at homeostasis[44,45]. We colonized mice with Hh (Hh+) or left them Hh-free (Hh⁻), implanted B16-F10 cells expressing the Hh7-2 epitope grafted onto the SFB-3340 scaffold, (B16-eHh7-2) and treated with anti-PD-1 (Extended Data Fig. 8a,b). Tumour growth was indistinguishable between Hh+ and Hh⁻ cohorts, indicating that Hh colonization did not enhance ICB in this antigen-matched setting (Extended Data Fig. 8c).

Although Hh colonization had no measurable effect on tumour growth, it significantly expanded Hh7-2 MHCII tetramer+ CD4+ T cells within B16-eHh7-2 tumours of Hh+ mice (Extended Data Fig. 8d). These tumour-resident tetramer+ cells, however, exhibited a mixed phenotype, with many co-expressing Foxp3 and T-bet, but producing minimal IFNγ and TNF upon ex vivo stimulation (Extended Data Fig. 8e,f). In the LILP, the expanded tetramer+ population was

skewed toward Foxp3+RORγt+ cells, consistent with Hh propensity to elicit a mucosal $T_{reg}$ cell program at steady state (Extended Data Fig. 8g,h). Although Foxp3+ cells were increased in the LILP of Hh+ mice (Extended Data Fig. 8i), the broader intratumoural cytotoxic compartment remained unaltered: CD8+:Foxp3+ ratios, total CD8+ T cell numbers and the frequency of IFNγ/TNF-producing CD8+ TILs were comparable between Hh+ and Hh⁻ mice (Extended Data Fig. 8j,k).

Fate-mapping experiments showed directly that gut-primed, Hh-specific Foxp3-lineage CD4+ T cells migrate to distal B16-eHh7-2 tumours but fail to acquire robust $T_H1$-like effector function. Adoptively transferred *TCR*$^{Hh7-2}$;*Foxp3-Cre;ROSA-LSL-tdTomato* (*TCR*$^{Hh7-2}$ *tdTomato-ON*$^{ΔFoxp3}$) cells gave rise to a substantial tdTomato+ population among CD4+ TILs, confirming previous Foxp3 expression and probable gut origin, but most remained functionally non-effector and produced little IFNγ following ex vivo stimulation (Extended Data Fig. 9a–c). Together, these data show that Hh robustly expands antigen-specific, Foxp3-lineage CD4+ T cells that traffic to distal tumours but these cells do not undergo $T_H1$-like effector conversion, and their persistent regulatory phenotype probably limits productive anti-tumour immunity and explains the lack of improved responsiveness to PD-1 blockade.

## Discussion

Antigenic mimicry, with microbial antigens resembling self-antigens, has profound implications for both autoimmune disease[46] and cancer immunotherapy. Previous studies have highlighted the potential significance of cross-reactivity between microbial antigens and tumour-associated antigens in cancers[29,47] or autoantigens in autoimmune diseases such as myocarditis, lupus and rheumatoid arthritis[46,48–51]. In patients, clinical responses to immune ICB have been correlated with the presence of distinct bacterial taxa in the gastrointestinal tract[10–16]. In a model of skin colonization with *S. epidermidis* engineered to express a model tumour antigen, antigen mimicry elicited T-cell-mediated tumour control, suggesting that TCR cross-reactivity may contribute to tumour rejection in humans[30]. Yet there is limited understanding of how gut microbiota can be optimally enlisted to enhance immune control of distal tumours.

In this study, we established an experimental system enabling mechanistic investigation of intestinal microbiota-driven anti-tumour immunity and demonstrated that SFB colonization markedly augments PD-1 blockade efficacy when the tumour expresses the matching antigen. The tumour-associated cells were derived largely from small intestine SFB-specific $T_H17$ cells, and acquired $T_H1$-like properties that were probably critical for enhancing mobilization and effector functions of tumour-infiltrating CD8+ T cells and other tumour-associated CD4+ T cells, thereby contributing to tumour control (Extended Data Fig. 10). scTCR-seq and fate-mapping experiments established a clonal link between SFB-specific $T_H17/T_{FH}$ cells in the small intestine and trans-differentiated $T_H1$-like cells

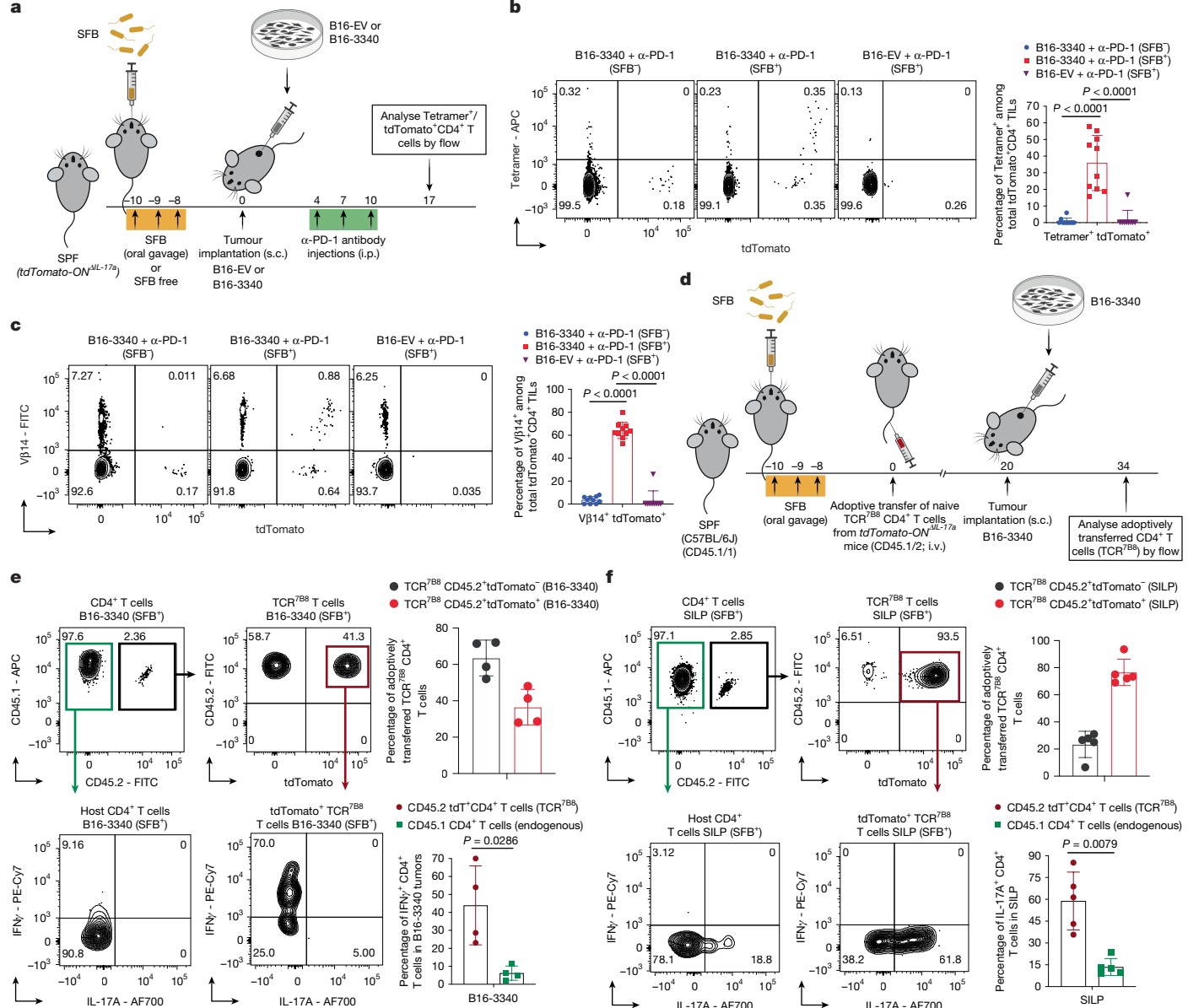

**Fig. 4 | Tracking of SFB-induced T cells in gut mucosa and distal tumour tissue by MHC tetramer staining and *Il17a* fate mapping. a**, Schematic representation of fate mapping in *Il17a-cre;ROSA-LSL-tdTomato* (*tdTomato-ON^ΔIL-17a*) mice, illustrating the identification of tumour-infiltrating CD4⁺ T cells that previously expressed IL-17A following colonization with SFB. **b**, Left, ex-T_H17 cell (tdTomato⁺) representation among SFB tetramer⁺ CD4⁺ TILs either in B16-3340 tumours in SFB⁻ or SFB⁺ mice (*n* = 10 mice per group) and B16-EV tumours in SFB⁺ mice (*n* = 9) (all anti-PD-1 treated). Right, quantification of the fraction of tetramer⁺ cells that are tdTomato⁺ (tdTomato-ON/ex-T_H17) per tumour. **c**, Left, representative flow plots showing Vβ14 staining among tdTomato⁺ CD4⁺ TILs from B16-3340 tumours in SFB⁻ or SFB⁺ mice (*n* = 10 mice per group) and B16-EV tumours in SFB⁺ mice (*n* = 9) (all anti-PD-1 treated). Right, percentage of Vβ14⁺ cells among tdTomato⁺ CD4⁺ TILs. **d**, Schematic representation of the adoptive transfer experiment to identify tumour-infiltrating SFB-specific TCR

transgenic (TCR^7B8) mouse T cells that previously expressed *Il17a*. Naive T cells from *Il17a-cre;ROSA-LSL-tdTomato* mice bred to TCR^7B8 transgenic mice (TCR^7B8 *tdTomato-ON^ΔIL-17a*) were transferred into SFB-colonized mice, and fate-mapped cells were characterized in the SILP and B16-3340 tumours. **e,f**, Total CD4⁺ T cells, including ex-T_H17 TCR^7B8 (tdTomato⁺) cells, were isolated from B16-3340 tumours (*n* = 4 mice) (**e**) and SILP (*n* = 5 mice) (**f**) 5 weeks post-adoptive transfer of naive SFB-specific TCR^7B8 CD4⁺ T cells and activated ex vivo for cytokine analysis. Top, Representative gating strategy; bottom, intracellular cytokine plots. Data are plotted as mean ± s.d., with each point representing a recipient mouse. Statistical significance was determined using unpaired two-sided Mann–Whitney *t*-test, with *P* values indicated on the corresponding graphs. Data are representative of two independent experiments. Schematics in **a** and **d** were created using BioRender (https://biorender.com).

(ex-T_H17 cells) in the tumour, confirming gut-to-tumour migration and phenotypic reprogramming. Therapeutic colonization experiments revealed that this microbiota-ICB synergy operates within a narrow early post-implantation window, underscoring the importance of timing in microbial antigen exposure.

Functional dissection using *DTA-ON^ΔIL-17a* mice revealed that IL-17A⁺ T_H17 cells are essential for both CD4⁺ and CD8⁺ T cell effector responses

and synergy with PD-1 blockade. The effector functions of ex-T_H17 cells appear superior to those of CD4⁺ T cells generated locally in the tumour-draining lymph node, probably reflecting earlier priming and acquisition of effector memory function, although immunosuppressive factors within the tumour may limit the functionality of locally primed T cells. Moreover, ex-T_H17 cells, unlike conventional T_H1 and T_H17 cells, are reported to be highly resistant to T_reg-cell-mediated suppression,

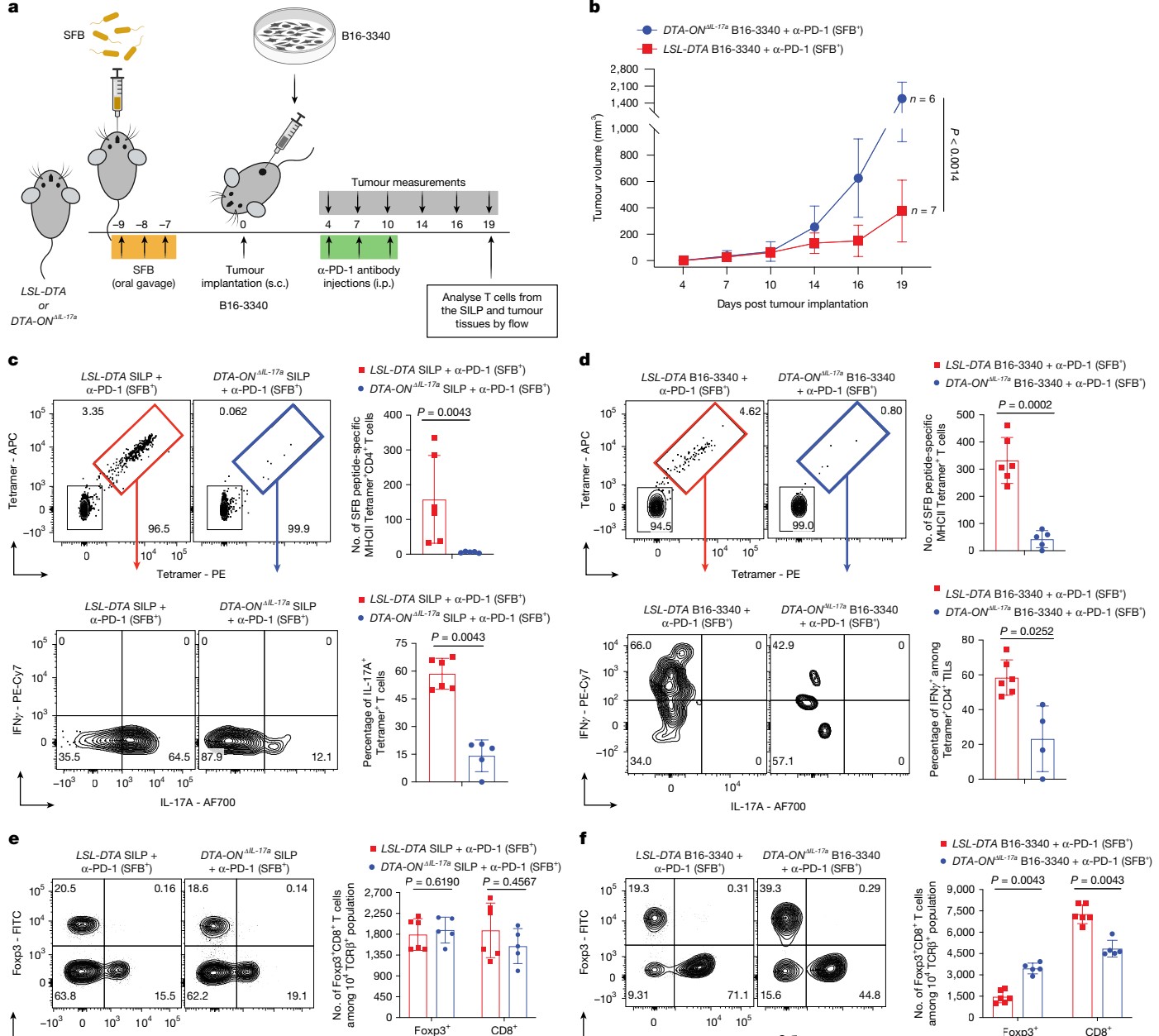

**Fig. 5 | IL-17A-lineage cells are required for SFB-dependent expansion of tumour-specific T cells and therapeutic response to PD-1 blockade.**
**a**, Schematic of the experimental design. *Il17a-Cre;ROSA-LSL-DTA* (*DTA-ON^ΔIL-17a*) and *ROSA-LSL-DTA* (control, *LSL-DTA*) mice were colonized with SFB, implanted subcutaneously with B16-3340 cells, and treated with anti-PD-1 antibody. Tumour growth was monitored, and T cells were isolated from the SILP and tumours on day 19 for flow cytometric analysis. **b**, Tumour growth curves of B16-3340 tumours in SFB⁺ *LSL-DTA* (red, *n* = 7) and *DTA-ON^ΔIL-17a* (blue, *n* = 6) mice treated with anti-PD-1. Data are shown as mean tumour volume ± s.d.; statistical significance was calculated using two-way ANOVA with Sidak's multiple comparisons. **c**,**d**, Top, representative flow cytometry plots of SFB-peptide-specific MHCII tetramer⁺ CD4⁺ T cells from the SILP (**c**) and B16-3340 (**d**) tumours of *LSL-DTA* (*n* = 6) and *DTA-ON^ΔIL-17a* (*n* = 5) mice (left) and quantification of

absolute tetramer⁺ cell numbers (right). Bottom, representative flow cytometry plots showing cytokine expression in tetramer⁺ CD4⁺ T cells from SILP and tumours with quantification of the fraction of IL-17A⁺ cells in the SILP (*n* = 6 for *LSL-DTA* and *n* = 5 for *DTA-ON^ΔIL-17a* mice) and IFNγ⁺ cells in the tumour (*n* = 6 for *LSL-DTA* and *n* = 4 for *DTA-ON^ΔIL-17a* mice) within the tetramer⁺ pool. **e**,**f**, Representative flow cytometry plots of Foxp3⁺ CD4⁺ T_reg cells and CD8⁺ T cells from SILP (**e**) and tumours (**f**) of *LSL-DTA* (*n* = 6) and *DTA-ON^ΔIL-17a* (*n* = 5) mice, with quantification of T_reg cells and CD8⁺ T cells. Bar graphs depict mean ± s.d.; with each data point representing an individual mouse. Statistical significance was determined using two-sided Mann–Whitney *t*-tests, and *P* values are indicated on the corresponding graphs. Data are representative of two independent experiments. Schematic in **a** was created using BioRender (https://biorender.com).

underscoring their potential importance in initiating and sustaining robust anti-tumour immune responses[52].

The contrasting outcomes observed with SFB and Hh colonization indicate that features of the microbiota-driven CD4⁺ T cell program, whether potentially pro-inflammatory or regulatory, critically determine

therapeutic synergy with ICB efficacy. Our findings with these models provide an important mechanistic framework, but further studies are needed to establish the presence, phenotype and functional relevance of commensal-specific T_H17 cells in human cancer immunity, to enable rational design of microbiota-assisted immunotherapeutic strategies.

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

# Methods

## Mice

SPF C57BL/6J (B6) mice (Jax, catalogue no. 000664, both sexes) were obtained from The Jackson Laboratories. CD45.1 congenic mice (B6.SJL-*Ptprc^a Pepc^b*/BoyJ, JAX, catalogue no. 002014), Rosa-CAG-LSL-tdTomato reporter mice (B6;129S6-*Gt(ROSA)26Sor^tm14(CAG-tdTomato)Hze*/J, JAX, catalogue no. 007908), *Il17a*-Cre mice (STOCK *Il17a^tm1.1(icre)Stck*/J, JAX, catalogue no. 016879) and *ROSA26-eGFP-DTA* (B6.129S6(Cg)-*Gt(ROSA)26Sor^tm1(DTA)Jpmb*/J, JAX, catalogue no. 032078) were purchased from The Jackson Laboratory. TCR[7B8] (C57BL/6-Tg(Tcra,Tcrb)2Litt/J) and TCR[Hh7-2] (C57BL/6-Tg(Tcra,Tcrb)5Litt/J) mice were generated in house as described previously. All transgenic lines were bred and maintained under SPF conditions at the Alexandria Center for Life Sciences animal facility, New York University School of Medicine.

For IL-17A fate mapping experiments, sex-matched littermates (both male and female) were used. Experimental cohorts were 6–8 weeks old at treatment onset. Sample sizes were determined by power analysis (power = 0.9, $\alpha$ = 0.05) using mean and s.d. estimates from previous and pilot studies (four to five animals per group). All animal procedures were conducted in compliance with protocols approved by the Institutional Animal Care and Use Committee (IACUC) of New York University School of Medicine.

## Antibodies, intracellular staining and flow cytometry

Monoclonal antibodies were obtained from eBioscience, BD Pharmingen, BioLegend, Thermo Fisher, Tonbo Bioscience and Invitrogen. The following fluorochrome-conjugated antibodies were used: CD4 BUV395 (GK1.5, BD, catalogue no. 563790, 1:400), CD25 APC (PC61, Thermo, catalogue no. 17-0251-82, 1:400), CD69 PE-Cy7 (H1.2F3, BioLegend, catalogue no. 104512, 1:200), CD44 AF700 (IM7, BD, catalogue no. 560567, 1:200) or BV510 (IM7, BD, catalogue no. 563114, 1:200), CD45.1 BV650 (A20, BD, catalogue no. 563754, 1:400), CD45.2 FITC (104, eBioscience, catalogue no. 11-0454-85, 1:400), CD19 PerCP-Cy5.5 (1D3, Tonbo, catalogue no. 65-0193-U100, 1:400), B220 PerCP-Cy5.5 (RA3-6B2, Invitrogen, catalogue no. 45-0452-82, 1:400), CD11c PerCP-Cy5.5 (N418, Invitrogen, catalogue no. 45-0114-82, 1:400) or PE-Cy7 (N418, BioLegend, catalogue no. 117318, 1:400), CD11b PerCP-Cy5.5 (M1/70, Invitrogen, catalogue no. 45-0112-82, 1:400) or BUV395 (BD, catalogue no. 563553, 1:400), MHCII I-A/I-E PerCP-Cy5.5 (M5/114.15.2, BioLegend, catalogue no. 107626, 1:400), NK1.1 PerCP-Cy5.5 (PK136, Invitrogen, catalogue no. 45-5941-82, 1:200), TCRβ BV711 (H57-597, BD, catalogue no. 563135, 1:200), TCRγδ PerCP-Cy5.5 (GL3, BioLegend, catalogue no. 118117, 1:400), FOXP3 FITC (FJK-16s, eBioscience, catalogue no. 11-5773-82, 1:200), RORγt BV421 (Q31-378, BD, catalogue no. 562894, 1:200), T-bet PE-CF594 (O4–46, BD, catalogue no. 562467, 1:70), IL-17A AF700 (TC11-18H10.1, BioLegend, catalogue no. 506914, 1:200), IFN-γ PE-Cy7 (XMG1.2, BioLegend, catalogue no. 505826, 1:200), Granzyme B AF700 (QA16A02, BioLegend, catalogue no. 372222, 1:200), TNF BV650 (MP6-XT22, BioLegend, catalogue no. 506333, 1:200), CXCR6 PE/Dazzle594 (SA051D1, BioLegend, catalogue no. 151117, 1:200), CD62L PE (MEL-14, BD Pharmingen, catalogue no. 553151, 1:400), and TCR Vβ14 FITC (14-2, BD Pharmingen, catalogue no. 553258, 1:400). Dead cells were excluded using 4′,6-diamidino-2-phenylindole (Sigma) or LIVE/DEAD Fixable Blue dye (Thermo Fisher).

For scTCR-seq coupled with scRNA-seq, cells were labelled with TotalSeq-C hashtag antibodies (BioLegend): Hashtag 1 (M1/42; 30-F11, catalogue no. 155861, 1:100), Hashtag 2 (catalogue no. 155863, 1:100), Hashtag 3 (catalogue no. 155865, 1:100), and Hashtag 4 (catalogue no. 155867, 1:100).

For transcription factor staining, cells were first stained for surface markers, then fixed and permeabilized using the FOXP3 staining buffer set (eBioscience), followed by nuclear staining. For intracellular cytokine analysis, cells were stimulated for 3 h in RPMI-1640 culture medium supplemented with 10% fetal bovine serum (FBS), plus phorbol 12-myristate 13-acetate (50 ng ml$^{-1}$, Sigma), ionomycin (500 ng ml$^{-1}$, Sigma) and GolgiStop (BD Biosciences), then stained for surface markers, fixed, permeabilized and subjected to intracellular/nuclear staining with eBioscience buffers.

Flow cytometry was performed using BD LSR II or Aria II instruments (BD Biosciences), data acquisition was carried out using BD FACSDiva software (v.8.0.1; BD Biosciences) and data were analysed with FlowJo software (v.10.10.0) (Tree Star).

## Design of SFB-3340 antigen construct and generation of cancer cell lines expressing SFB-3340

To establish a synthetic neoantigen mimicry model, we designed a construct encoding a small, independently folded domain containing a well-characterized immunogenic CD4$^+$ T cell epitope (hereafter referred to as SFB-3340) derived from a large membrane protein of SFB (SFBNYU_003340, GenBank: EGX28318.1)[36]. A mammalian codon-optimized gene fragment encoding SFB-3340 antigen, fused via a flexible linker to a c-Myc tag, was synthesized chemically (GenScript) and cloned into the pEF1α-IRES-Neo vector (Addgene, catalogue no. 28019) between *Nhe*I and *Sal*I restriction sites. Expression was driven by the constitutive EF1α promoter.

To establish stable B16-F10 and LLC1 cell lines expressing the neoantigen construct SFB-3340 (referred to as B16-3340 and LLC1-3340, respectively), B16-F10 cells (ATCC, catalogue no. CRL-6475) and LLC1 cells (ATCC, catalogue no. CRL-1642) were transfected with the expression plasmid complexed with TransIT-293 transfection reagent (Mirus Bio). The following day (approximately 18–22 h after transfection), the culture medium (DMEM supplemented with heat-inactivated 10% (v/v) FBS, 100 U ml$^{-1}$ penicillin and 0.1 mg ml$^{-1}$ streptomycin) was replaced with selection medium, which is same culture medium containing 1 mg ml$^{-1}$ Neomycin (G-418) for selection. The cell cultures were then incubated for an additional 4–5 days, with the selection medium changed every 2 days to select for stably transfected clones. Single clones were isolated and further expanded in selection medium. The cells were passaged several times before assessing antigen expression by western blot and ex vivo activation assays. As a control cell lines, B16-F10 and LLC1 cells were transfected with the empty vector (pEF1α-IRES-Neo without the SFB-3340 gene fragment) (referred to as B16-EV and LLC1-EV respectively), and also selected in 1 mg ml$^{-1}$ G-418-containing medium following the same protocol as for the B16-3340 and LLC1-3340 cell lines.

For generation of MC-38 cells expressing SFB-3340 antigen, the coding sequence of the designed SFB-3340 construct was cloned into an MSCV-IRES-Thy1.1 retroviral vector. Retrovirus was produced by transient transfection of Plat-E packaging cells; viral supernatants were collected, filtered and used to transduce MC-38 cells in the presence of 8 μg ml$^{-1}$ polybrene with spinoculation. At 72 h after transduction, Thy1.1$^+$ cells were purified by FACS, expanded and maintained at >95% Thy1.1$^+$.

## Immunoblotting and ex vivo activation assay

To confirm stable expression of SFB-3340 in B16-3340, LLC1-3340 and MC-3340 cells, and to test whether the expressed antigen could stimulate SFB-3340 antigen-specific TCR[7B8] CD4$^+$ T cells ex vivo[36], antigen-expressing and empty vector control cell lines were lysed in M-PER reagent (Thermo Fisher Scientific) supplemented with a protease inhibitor cocktail (Complete Mini EDTA-free; Roche). Lysates were centrifuged at 17,000*g* to pellet cellular debris, and supernatants were stored at −20 °C.

For western blotting, normalized amounts of protein from the resulting cell lysates were resolved by SDS–PAGE and transferred to nitrocellulose membrane using the iBlot 2 Dry Blotting System (Invitrogen). Membranes were blocked in PBS blocking buffer (LI-COR) and incubated overnight with anti-*c-myc* antibody (1:2,000, dilution, Cell Signalling) at 4 °C. Next day, after washing in PBST (PBS and 0.1% Tween-20), membranes were incubated with fluorescently conjugated

secondary antibodies (LI-COR) at 1:10,000 dilution in PBS blocking buffer for 1 h at room temperature and imaged using the LI-COR Odyssey CLx Imaging system in the 800-nm channel (LI-COR).

For the ex vivo activation assay, spleens from female SPF (SFB-free) CD45.2 mice (6–7 weeks, Jackson Laboratories) were dissociated into single-cell suspension using the GentleMACS Spleen Dissociation Kit (Miltenyi Biotec) and DCs were isolated with CD11c microbeads (Miltenyi). Naive antigen-specific CD4$^+$ T cells were purified from spleens and lymph nodes of CD45.1 TCR$^{7B8}$ transgenic mice by mechanical dissociation. Red blood cells were lysed using ACK lysis buffer (Lonza). Naive TCR$^{7B8}$ CD4$^+$ T cells were sorted as CD4$^+$TCRβ$^+$CD44$^{lo}$CD62L$^{hi}$CD25$^-$Vβ14$^+$ using a FACSAria II (BD Biosciences).

In 96-well round-bottom plates (CELLTREAT), $2 \times 10^4$ DCs were incubated in RPMI + 10% FBS for 2 h at 37 °C and 5% $CO_2$ with one of the following: 10 μl PBS, 10 μl of cell lysates either from $1 \times 10^6$ B16-3340, LLC1-3340 or MC-3340 cells, 10 μl of cell lysates from corresponding empty vector control cells or 500 nM of chemically synthesized SFB-3340 peptide (GenScript). After the 2-h incubation (antigen loading), $1 \times 10^5$ naive TCR$^{7B8}$ CD4$^+$ T cells were added to each well and co-cultured for an additional 20–24 h. T cell activation was assessed by flow cytometry based on CD69 and CD25 expression.

## Colonization of mice with SFB by oral gavage

SFB colonization was achieved through three consecutive oral gavages using faecal pellets from SFB mono-associated mice, following previously described methods[36,53]. Briefly, fresh faecal pellets were homogenized through a 100-μm filter, pelleted at 3,400 rpm for 10 min, and re-suspended in PBS. Each animal was administered one-quarter pellet by oral gavage. Colonization was confirmed by quantitative PCR (qPCR) with SFB-specific primers using universal 16S primers as control. Primers used were: 16S F, CGGTGAATACGTYCGG; 16S R, GGWTACCTTGTTACGACTT[54]; SFB F, GACGCTGAGGCATGAGAGCAT; SFB R, GACGGCACGGATTGTTATTCA.

## Hh culture and oral infection

Hh was provided by J. Fox (MIT) and cultured as described previously[45]. Frozen Hh stocks were maintained in Brucella broth with 20% glycerol at −80 °C. For culture, bacteria were streaked onto blood agar plates (Thermo Scientific Blood Agar with 5% sheep blood; Thermo Fisher) and incubated at 37 °C in a hypoxia chamber (Billups–Rothenberg) under a micro-aerobic atmosphere (80% $N_2$, 10% $H_2$, 10% $CO_2$; Airgas) adjusted to 3–5% $O_2$. After 4 days, bacteria were harvested with a pre-moistened sterile cotton swab, re-suspended in Brucella broth and administered to mice at a density of $1 \times 10^8$ colony-forming units of Hh (equivalent to around 1 optical density unit) by oral gavage. A second inoculation was performed 3 days later. Colonization was confirmed by qPCR with Hh-specific primers using universal 16S primers as control. Primers used were: 16S F, CGGTGAATACGTYCGG; 16S R, GGWTACCTTGTTA CGACTT[54]; Hh F, CAACTAAGGACGAGGGTTG; Hh R, TTCGGGGAGCT TGAAAAC.

## In vivo tumour models and antibody treatments

SPF female C57BL/6J mice (6–7 weeks; Jackson Laboratories) were either maintained SFB-free (SPF, SFB$^-$) or colonized with SFB by oral gavage as described above[36,53]. For subcutaneous tumour studies, B16-F10, LLC1 and MC-38 cell lines expressing the SFB-3340 protein fragment (B16-3340, LLC1-3340, MC-3340) or matched empty vector control cells were cultured in complete growth medium and 200 μg ml$^{-1}$ of G-418 was added during maintenance of the B16-F10 and LLC1 transductants. Tumour cells, harvested freshly at 50–60% confluence after three to four passages, were washed and re-suspended in sterile PBS. Mice were inoculated subcutaneously in the right flank with $2.5 \times 10^5$ cells in 100 μl PBS or with $5.0 \times 10^5$ cells in 100 μl PBS for LLC1 and MC-38 (day 0).

For checkpoint blockade (in vivo anti-PD-1) experiments, mice received 100 μl intraperitoneal (i.p.) injections of 250 μg of anti-PD-1 antibody (clone RMP1-14, BioXCell, catalogue no. BP0146) diluted in 1× PBS when tumours were palpable (for B16-F10 on days 4, 7 and 10; for LLC1 days 7, 10 and 13; and for MC-38 on days 5, 8 and 11) post tumour implantation. Tumour growth was monitored by caliper measurements, and tumour volume was calculated using the ellipsoid volume formula ($0.5 \times D \times d^2$, where $D$ represents the longer diameter and $d$ is the shorter diameter). Sample sizes were not predetermined but were based on standards commonly practiced in the field. Allocation to experimental groups was random. To minimize microbiota-related variability, control mice were from the same litter, of the same sex, and housed in the same room. Experiments were conducted blinded where feasible. For some tumour studies, investigators responsible for SFB colonization and tumour measurement remained blinded until study completion. Blinding was not possible in some experiments owing to the risk of SFB cross-contamination. Mice were humanely euthanized if tumours reached a volume of 2,000 mm$^3$ or if any signs of discomfort were observed by investigators or identified by the animal care staff, in accordance with institutional IACUC guidelines and daily monitoring.

For in vivo depletion of CD4$^+$ and CD8$^+$ T cells, mice received i.p. injections of 200 μg of either anti-CD4 (clone GK1.5, BioXCell, catalogue no. BE0003-1) or anti-CD8a (clone 2.43, BioXCell, catalogue no. BE0061) antibody per mouse. Injections were initiated 2 days before tumour implantation and continued twice weekly thereafter until experimental end-points. Control mice were injected with PBS. In parallel, mice received three i.p. injections of anti-PD-1 antibody (250 μg per mouse) on days 4, 7, and 10 post tumour implantation as described above. The depletion efficiency was >95% in all the mice as monitored by flow cytometry of peripheral blood/spleen.

## Isolation of lymphocytes from tumour, intestinal tissues and lymphoid organs

For tumour-infiltrating lymphocyte isolation, tumours were harvested 17–20 days post-implantation, minced and digested in RPMI containing collagenase type 1 (250 U ml$^{-1}$; STEMCELL Technologies), DNase I (100 μg ml$^{-1}$; Sigma), dispase (0.1 U ml$^{-1}$; Worthington) and 10% FBS with constant stirring at 37 °C for 30 min. The resulting cell suspension was filtered, and lymphocytes were isolated using a 40%/80% Percoll density gradient (GE Healthcare) and centrifuged at 800$g$ for 20 min without brake. Cells at the interface were collected for downstream analysis.

For isolation of lymphocytes from the SILP and LILP, the entire small intestine or colon were dissected from mice. Mesenteric fat and Peyer's patches were removed carefully from these tissues. Intestinal tissues were opened longitudinally, washed thoroughly to remove faecal matter and treated sequentially with 1× Hank's Balanced Salt Solution containing 1 mM dithiothreitol at 37 °C for 10 min with gentle shaking (200 rpm), followed by two incubations in 5 mM EDTA at 37 °C for 10 min each to remove epithelial cells. The remaining tissues were then minced with scissors and digested in RPMI containing 10% FBS, dispase (0.05 U ml$^{-1}$; Worthington), collagenase II (1 mg ml$^{-1}$; Roche) and DNase I (100 μg ml$^{-1}$; Sigma) at 37 °C for 45 min with constant shaking (175 rpm). The digested tissues were then filtered through a 70-μm strainer to remove large debris. Viable lamina propria lymphocytes were collected at the interface of a 40%/80% Percoll/RPMI gradient (GE Healthcare). For isolation of cells from lymph nodes and spleens, tissues were dissociated mechanically with the plunger of a 1 ml syringe and filtered through 70-μm cell strainers. Red blood cells were lysed with ACK buffer (Thermo Fisher) before downstream applications[45].

## MHCII tetramer production and staining

Fluorophore phycoerythrin (PE) and allophycocyanin (APC) conjugated, I-A$^b$/3340-A6 (SFB-peptide-specific) and I-Ab/HH-E2 (Hh_1713-E2 peptide-specific) MHCII tetramers were synthesized at the NIH tetramer core facility[55]. In brief, immunodominant epitopes QFSGAVPNKT (3340-A6) and QESPRIAAAYTIKGA (HH_1713-E2), validated with the corresponding hybridoma (TCR$^{7B8}$ and TCR$^{Hh7-2}$ respectively) stimulation

assay, were covalently linked to I-A[b] via a flexible linker to produce pMHCII monomers. Soluble monomers were purified, biotinylated and tetramerized with PE- or APC-labelled streptavidin[36]. Analysis of tetramer[+] cells was performed as previously described with minor modifications[56]. Briefly, cells were first re-suspended in FACS buffer with FcR block (anti-mouse CD16/32), 2% mouse serum and 2% rat serum. Cells were then stained with PE- and APC-conjugated tetramers (10 nM) at room temperature for 1 h in the dark. Subsequently, the cells were washed and subjected to antibody staining against surface molecules at 4 °C.

### IFNγ ELISPOT assay

IFNγ ELISPOT assay was performed using a mouse IFNγ ELISPOT kit (R&D systems) according to the manufacturer's instructions. Briefly, $5 \times 10^4$ CD4[+] T cells, extracted and sorted from either B16-3340 tumour tissue or SILP of SFB[+] and SFB[−] mice as described above, were stimulated with either SFB-3340 peptide (specific peptide) or Hh7-2 peptide (non-specific peptide). CD11c[+] APCs ($2 \times 10^4$), purified from the spleen of SPF (SFB-free) mice as described above, were used for antigen presentation in this assay. Dots (IFNγ producing cells) were enumerated automatically using ImmunoSpot software (v.5.0).

### Adoptive transfer of naive TCR[7B8] and TCR[Hh7-2] CD4[+] T cells

Spleens were harvested from donor TCR[7B8] *tdTomato-ON*[ΔIl-17a] or TCR[Hh7-2] *tdTomato-ON*[ΔFoxp3] mice, disassociated mechanically, and treated with ACK lysis buffer (Lonza) to remove red blood cells. Naive CD4[+] T cells (TCR[7B8] or TCR[Hh7-2]) were sorted by flow cytometry (FACS Aria II, BD Biosciences) based on the following surface markers: CD4[+]CD3[+]CD44[lo] CD62L[hi]CD25[−]TCRVβ14[+] (7B8) or TCRVβ6[+] (Hh7-2). Sorted cells were then re-suspended in PBS on ice and injected intravenously (i.v.) into the tail vein in congenic isotype-labelled recipient mice colonized with SFB or Hh. Cells from indicated tissues were analysed 5 weeks post-transfer.

### scRNA-seq and scTCR-seq experiment

scRNA-seq and scTCR-seq were performed using the Chromium Single Cell 5′ v.2 reagent kit and Chromium Single Cell Mouse TCR Amplification Kit (10x Genomics). Tumour-infiltrating lymphocytes from B16-3340 tumours and lymphocytes from the SILP of SFB[−] and SFB[+] mice ($n$ = 5 in each group) were isolated as described above. CD4[+] T cells were then sorted from pooled cells of either SFB[−] or SFB[+] tumour tissues or SILP of individual mice using FACS Aria II (BD Biosciences). Sorted CD4[+] T cells from each group (Tumour SFB[−], Tumour SFB[+], SILP SFB[−] and SILP SFB[+]) were resuspended in PBS containing 0.05% BSA and stained with cell hashing antibodies, TotalSeq-C0301 to C0304 (BioLegend, catalogue nos. 155861, 155863, 155865 and 155867)[57] for 20 min on ice. Cells were then washed three times with MACS buffer. CD4[+] T cells from both SFB[−] and SFB[+] tumours were combined at a 1:1 ratio. Similarly, CD4[+] T cells from the SILP of SFB[−] and SFB[+] mice were combined at a 1:1 ratio. Approximately $1.5 \times 10^4$ cells per sample were loaded onto the Chromium Controller (10x Genomics) and libraries were prepared with the Chromium Single Cell 5′ kit following the manufacturer's instructions. Libraries were sequenced using the NovaSeq 6000 system with a sequencing depth of more than 20,000 paired-end reads per cell. Sequencing reads were aligned to the mouse reference genome (mm10-2020-A, 10x Genomics) using Cell Ranger (v.7.1.0; 10x Genomics). Downstream data were processed and analysis were performed with the R packages Seurat v.5.1.0 (ref. 58).

### Data processing of scRNA-seq

To preprocess single-cell data, raw scRNA-seq data were processed with Cell Ranger 'multi' software (v.7.1.0, 10x Genomics) using the mouse reference genome (mm10 2020-A, 10x Genomics). For scTCR-seq, data were aligned and quantified with CellRanger 'multi' software (v.6.6.1, 10x Genomics) against the reference vdj_GRCm38_alts_ensembl-5.0.0, using default parameters.

For scRNA-seq analysis, cells with fewer than 200 detected genes or more than 5% mitochondrial gene content were excluded. Hashtag oligonucleotides (HTO) counts were normalized using centred log ratio transformation and demultiplexing with Seurat::HTODemux function (positive quantile set to 0.99). Doublets mapped to several HTO tags were removed. RNA counts were normalized with Seurat::SCTransform function, regressing out cell cycle, ribosomal and mitochondrial scores[59]. Paired SFB[−] and SFB[+] samples from the same tissue were integrated using the Seurat standard scRNA-seq integration workflow with 3,000 anchor genes. A shared nearest neighbour graph was constructed using the first 40 principal components and Leiden clustering (Seurat::FindClusters function) was applied at several resolutions to identify potential rare subsets[60]. Clusters were annotated based on canonical markers and differentially expressed genes identified with Seurat::FindAllMarkers (logistic regression model). Cells were then projected onto a UMAP for visualization[61].

TCR sequence data were processed using Cell Ranger vdj pipeline to identify TCR genes and CDR3 sequences. For each sample, full length, productive TRB and TRA chains were retained for downstream analysis. Clonal expansions were defined as clonotypes with identical CDR3 nucleotide sequences of both chains present in at least three cells across all samples. TCR metadata were merged with the scRNA-seq Seurat object by cell barcodes and sample ID. Phenotypic characterization of TCR clonotypes was performed by exporting metadata from the Seurat object and analysed and quantified in Microsoft Excel (v.16.73).

Differential gene expression between groups was tested with the MAST package (MAST_1.28.0) as implemented in Seurat v.5.1.0 (ref. 62), which applies to hurdle model adapted to scRNA-seq data. Genes with Bonferroni-adjusted $P$ value < 0.05 were considered as statistically significant.

### Statistical analysis

Statistical tests including unpaired two-sided $t$-test, paired two-sided $t$-test, one-way ANOVA with Bonferroni correction, two-way ANOVA with Sidak's multiple comparisons, Mann–Whitney test and the Mantel-Cox test for survival curves were all performed to compare the results using GraphPad Prism v.9 (GraphPad). No samples were excluded from analysis. Exact $P$ values are reported where possible, and $P < 0.05$ were considered statistically significant.

### Reporting summary

Further information on research design is available in the Nature Portfolio Reporting Summary linked to this article.

## Data availability

All mouse sequencing data generated and assembled for this project are available at Zenodo (https://doi.org/10.5281/zenodo.17399749)[63]. Reference genome mm10-2020-A was used for mapping. Source data are provided with this paper.

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

**Acknowledgements** We thank members of the Littman laboratory for valuable discussion. We thank S. Gottesman for valuable discussion and critical reading of the manuscript and A. Lund, S. Naik and S. Schwab for valuable feedback. We thank the NIH Tetramer Core Facility (NIH Contract 75N93020D00005 and RRID:SCR_026557) for providing MHCII tetramers and the NYU Genome Technology Center (GTC) for scRNA-seq and scTCR-seq. The GTC is partially supported by NYU Cancer Center Support Grant NIH/NCI P30CA016087 at the Laura and Isaac Perlmutter Cancer Center, S10 RR023704-01A1 and NIH S10 ODO019974-01A1. This work was supported by a Merieux Foundation grant (D.R.L.), the Helen and Martin Kimmel Center for Biology and Medicine (D.R.L.), NIH grants R01AI158687 and R01CA255635 (D.R.L.) and the Howard Hughes Medical Institute (D.R.L.).

**Author contributions** T.A.N. and D.R.L. designed the study and analysed the data. T.A.N. performed all the experiments with assistance from G.R.-M., A.D and E.A. T.A.N., Yuan Hao and Yuhan Hao did bioinformatics analysis. T.A.N. and D.R.L. wrote the manuscript, with input from the other authors. D.R.L. supervised the research.

**Competing interests** D.R.L. is co-founder of Vedanta Biosciences and ImmunAI, on the advisory boards of Nilo Therapeutics, IMIDomics, Sonoma Biotherapeutics and Evommune, and on the board of directors of Pfizer Inc. Yuhan Hao is co-founder and equity holder of Neptune Bio. The other authors declare no competing interests.

**Additional information**
**Correspondence and requests for materials** should be addressed to Dan R. Littman.

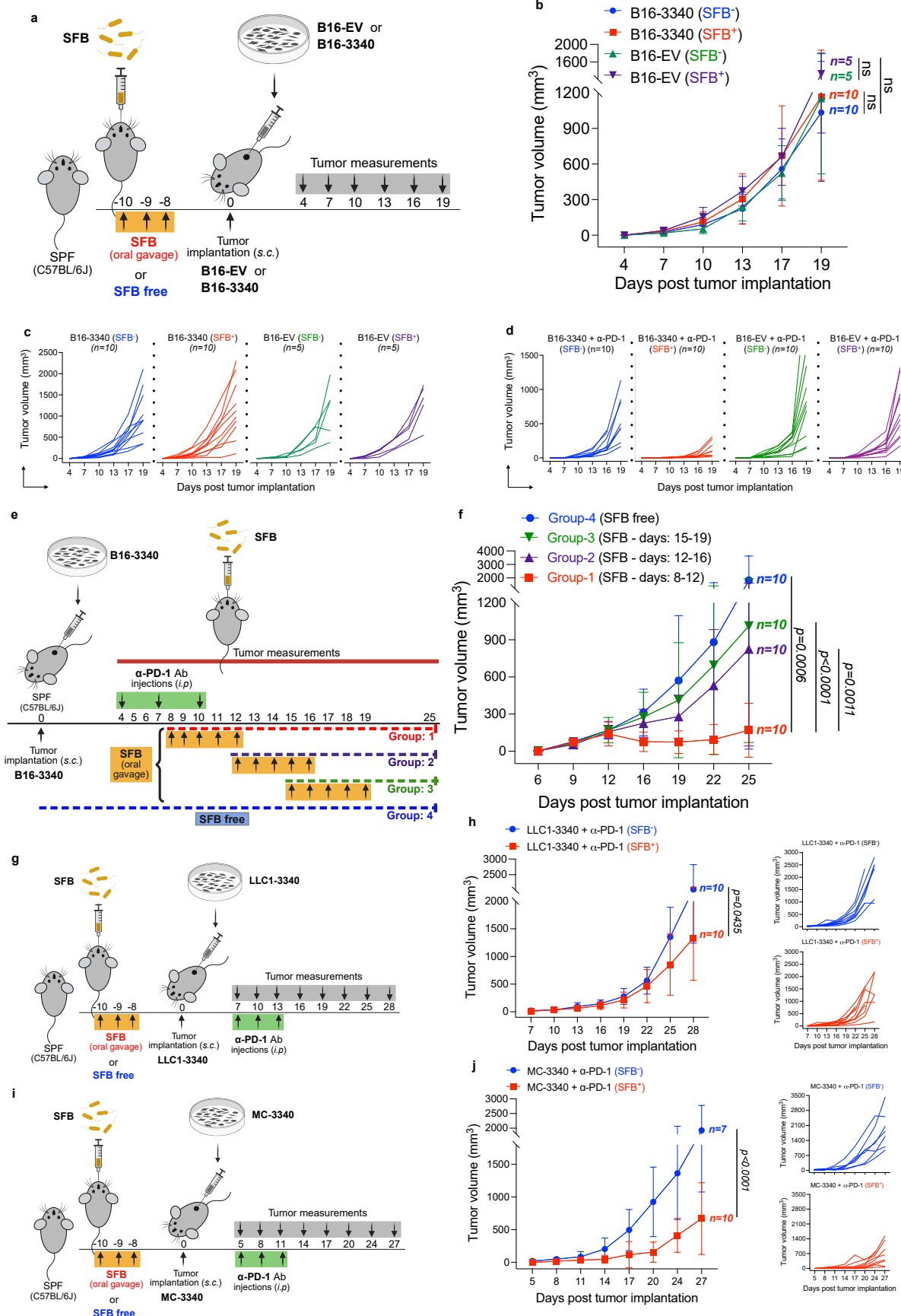

**Extended Data Fig. 1** | See next page for caption.

**Extended Data Fig. 1 | SFB colonization does not alter tumor growth without anti-PD-1 therapy. (a)** Experimental design for the synthetic mimicry model in C57BL/6 J mice without anti-PD-1 treatment. Mice were colonized with SFB or kept SFB-free, followed by implantation of B16-3340 or B16-EV tumor cells. No anti-PD-1 antibody was given. **(b)** Tumor growth curves (mean ± s.d.) for B16-3340 (n = 10 per group) or B16-EV (n = 5 per group) in SFB⁺ and SFB⁻ mice. No significant differences in tumor growth were observed between groups (two-way ANOVA with Sidak's multiple comparisons). **(c)** Individual tumor growth curves over time (B16-3340 SFB⁻, blue; B16-3340 SFB⁺, red; B16-EV SFB⁻, green; B16-EV SFB⁺, magenta). **(d)** Related to Fig. 1e, individual tumor growth curves over time after anti-PD-1 treatment for the same groups. Data in (b-d) are representative of ≥2 independent experiments. **(e,f)** Therapeutic SFB gavage experimental design **(e)** with treatment (SFB gavage) starting at various time points post tumor implantation, and corresponding tumor growth curves **(f)** for C57BL/6 J SPF mice subcutaneously implanted with B16-3340 and receiving anti-PD-1 antibody (*i.p.*). Four groups (n = 10 mice per group) differed only in timing of oral SFB gavage (Group 1: days 8–12; Group 2: days 12–16; Group 3: days 15–19) or remained SFB-free (Group 4: no gavage). **(g,i)** Experimental designs for LLC1-3340 **(g)** and MC-3340 **(i)** tumor models. **(h,j)** Left: Tumor growth curves (mean ± s.d.) for LLC1-3340 (n = 10 mice per group) **(h)** and MC-3340 **(j)** in SFB⁻ (blue; n = 7 mice) and SFB⁺ (red; n = 10 mice) mice with corresponding individual tumor growth traces (right) for SFB⁻ (top) and SFB⁺ (bottom) mice. Statistical comparisons were determined using two-way ANOVA with Sidak's multiple comparisons, and *P*-values are indicated on the corresponding graphs. Data are representative of two independent experiments. Schematics in **a**, **e**, **g** and **i** were created using BioRender (https://biorender.com).

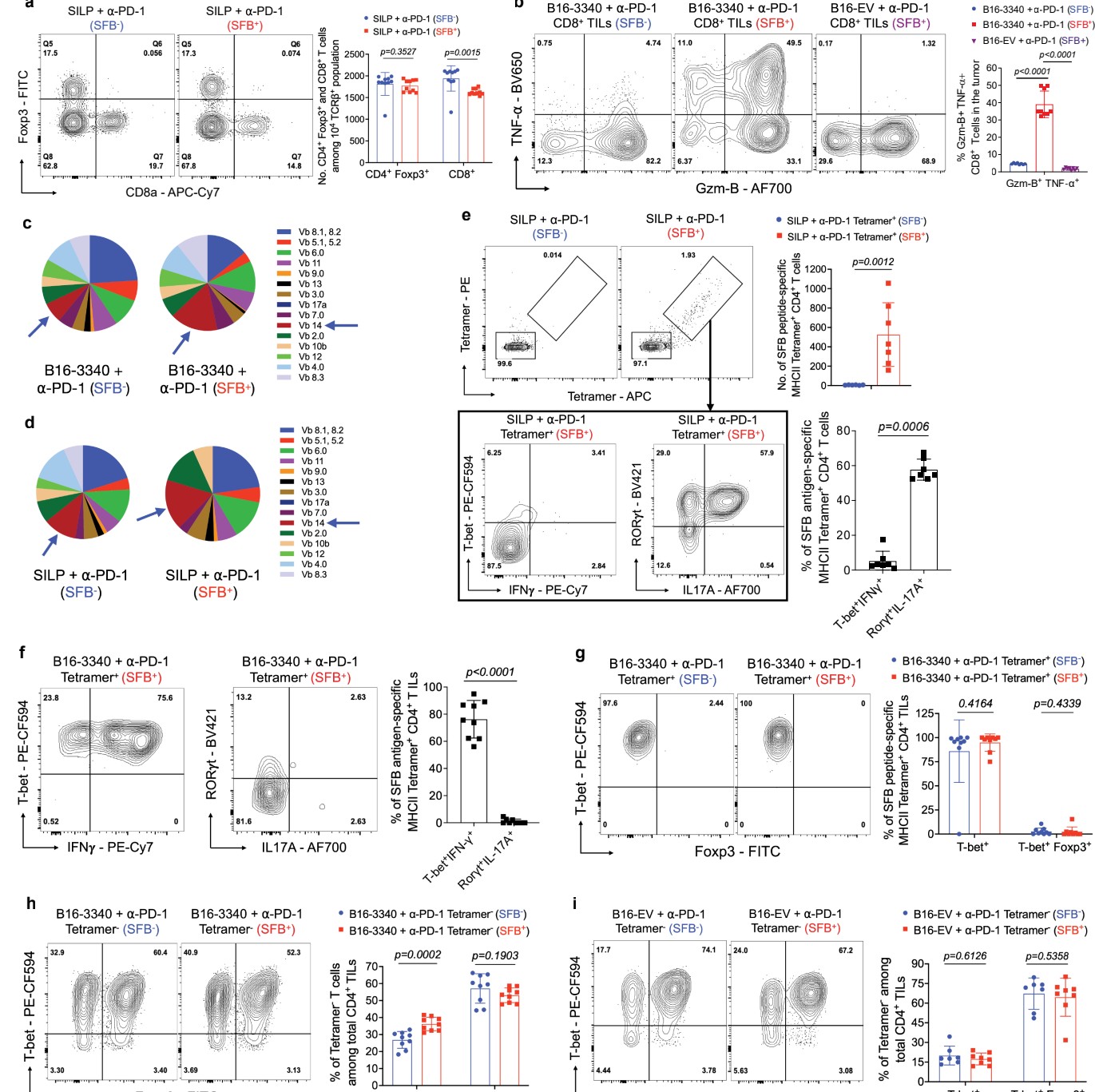

**Extended Data Fig. 2 | Differential immune responses in the SILP and tumor microenvironment of SFB-colonized mice following anti-PD-1 therapy.**
**(a)** Representative flow cytometry plots of CD8$^+$ T cells and Foxp3$^+$ CD4$^+$ Tregs in the SILP of SFB-free (SFB$^-$) and SFB-colonized (SFB$^+$) mice (left panel) with quantification of CD8$^+$ T cells and Treg frequencies (right panel; n = 10 mice per group). **(b)** Left: representative flow cytometry plots showing expression of effector gene products (Gzm-B$^+$ and TNF-α$^+$) in CD8$^+$ TILs isolated from B16-3340 (SFB$^-$ and SFB$^+$) and B16-EV (SFB$^+$) tumors. Right: frequencies of TNF-α$^+$ Gzm-B$^+$ CD8$^+$ TILs are quantified: B16-3340 (SFB$^-$, n = 8), B16-3340 (SFB$^+$, n = 9) and B16-EV (SFB$^+$, n = 8). **(c,d)** TCR Vβ repertoire composition among CD4$^+$ T cells in B16-3340 tumors **(c)** and SILP **(d)** from SFB$^-$ and SFB$^+$ mice (coloured pie charts indicate proportional usage of Vβ families; arrows highlight Vβ segments enriched in SFB$^+$ samples). **(e)** Top: representative SFB peptide-specific MHCII tetramer staining of SILP CD4$^+$ T cells from SFB$^-$ (n = 6) and SFB$^+$ (n = 7) mice, and quantification of tetramer$^+$ cells (right). Bottom: phenotype of SILP

tetramer$^+$ cells from SFB$^+$ mice showing T-bet/IFN-γ and RORγt/IL-17A profiles (representative plots, left; quantification, right; n = 7). **(f)** Phenotype of tumor-resident tetramer$^+$ CD4$^+$ T cells from B16-3340 SFB$^+$ tumors showing T-bet/IFN-γ and RORγt/IL-17A expression (representative plots, left; quantification, right; n = 9 mice group). **(g)** Foxp3 and T-bet expression in tumor tetramer$^+$ CD4$^+$ TILs from SFB$^-$ and SFB$^+$ mice (left) with quantification (right, n = 9 mice per group). **(h,i)** Foxp3 and T-bet expression among tetramer$^-$ CD4$^+$ TILs from B16-3340 **(h)** and B16-EV **(i)** tumors in SFB$^-$ and SFB$^+$ mice (representative plots, left; quantification, right). Group sizes: B16-3340 SFB$^-$, n = 9; B16-3340 SFB$^+$, n = 9; B16-EV SFB$^-$, n = 7; B16-EV SFB$^+$, n = 8. Data are shown as mean ± s.d., with each data point representing an individual mouse. Statistical significance was determined using unpaired two-sided Mann-Whitney t-test, and P-values are indicated on the corresponding bar graphs. All experiments were independently repeated at least twice with similar results.

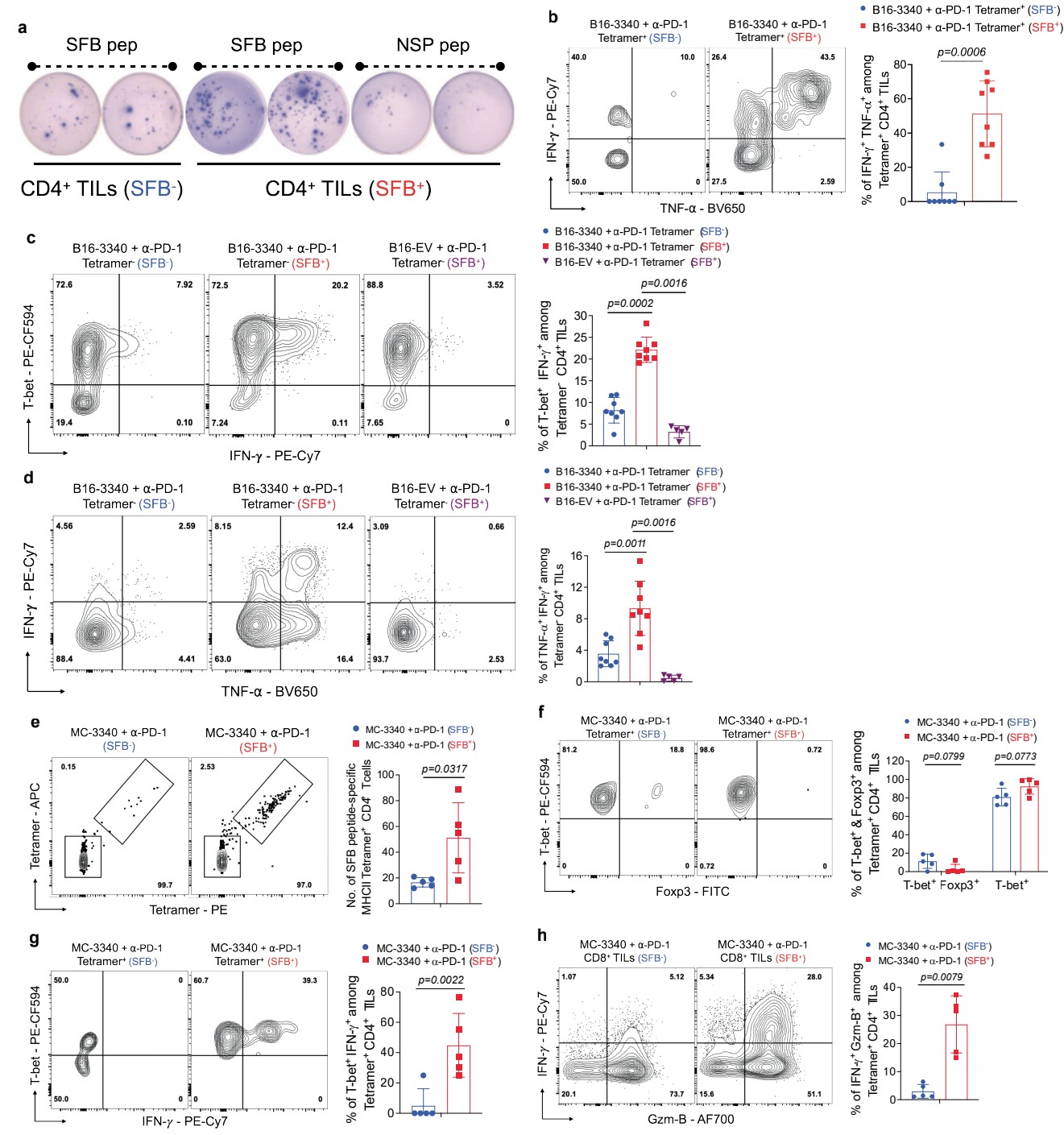

**Extended Data Fig. 3 | Phenotypic analysis of SFB-induced CD4⁺ T cell responses in tumors. (a)** IFN-γ ELISpot of CD4⁺ TILs isolated from B16-3340 tumors of SFB⁻ and SFB⁺ mice and stimulated for 24 hrs with SFB-3340 peptide (SFB peptide recognized by TCR[7B8]) or non-specific peptide (NSP) ex vivo. Representative wells (left) and quantification (right). **(b)** TNF-α and IFN-γ expression in tetramer⁺ CD4⁺ TILs isolated from B16-3340 tumors in SFB⁻ and SFB⁺ mice (n = 8 per group). **(c,d)** Expression of T-bet, IFN-γ and TNF-α in tetramer⁻ CD4⁺ T cells from B16-3340 tumors of SFB⁻ (n = 8) and SFB⁺ (n = 9) and from B16-EV tumors in SFB⁺ (n = 5) mice. **(e)** SFB-peptide MHC-II tetramer⁺ CD4⁺ T cells recovered from MC-3340 tumors following α-PD-1 treatment in

SFB⁻ (blue) and SFB⁺ (red) mice (n = 5 mice per group). **(f-h)** Phenotypic and functional characterization of intratumoral tetramer⁺ CD4⁺ T cells and bystander CD8⁺ TILs: T-bet/Foxp3 (**f**), T-bet/IFN-γ (**g**) in tetramer⁺ CD4⁺ T cells, and IFN-γ/Granzyme-B in CD8⁺ TILs (**h**) (n = 5 mice per group). Representative flow cytometry plots are shown in the left panels and quantification on the right (**b-h**). Data are plotted as mean ± s.d., with each data point representing an individual mouse. Statistical significance was determined using unpaired two-sided Mann-Whitney t-test, and P-values are indicated on the respective graphs. All experiments were independently repeated at least twice with similar results.

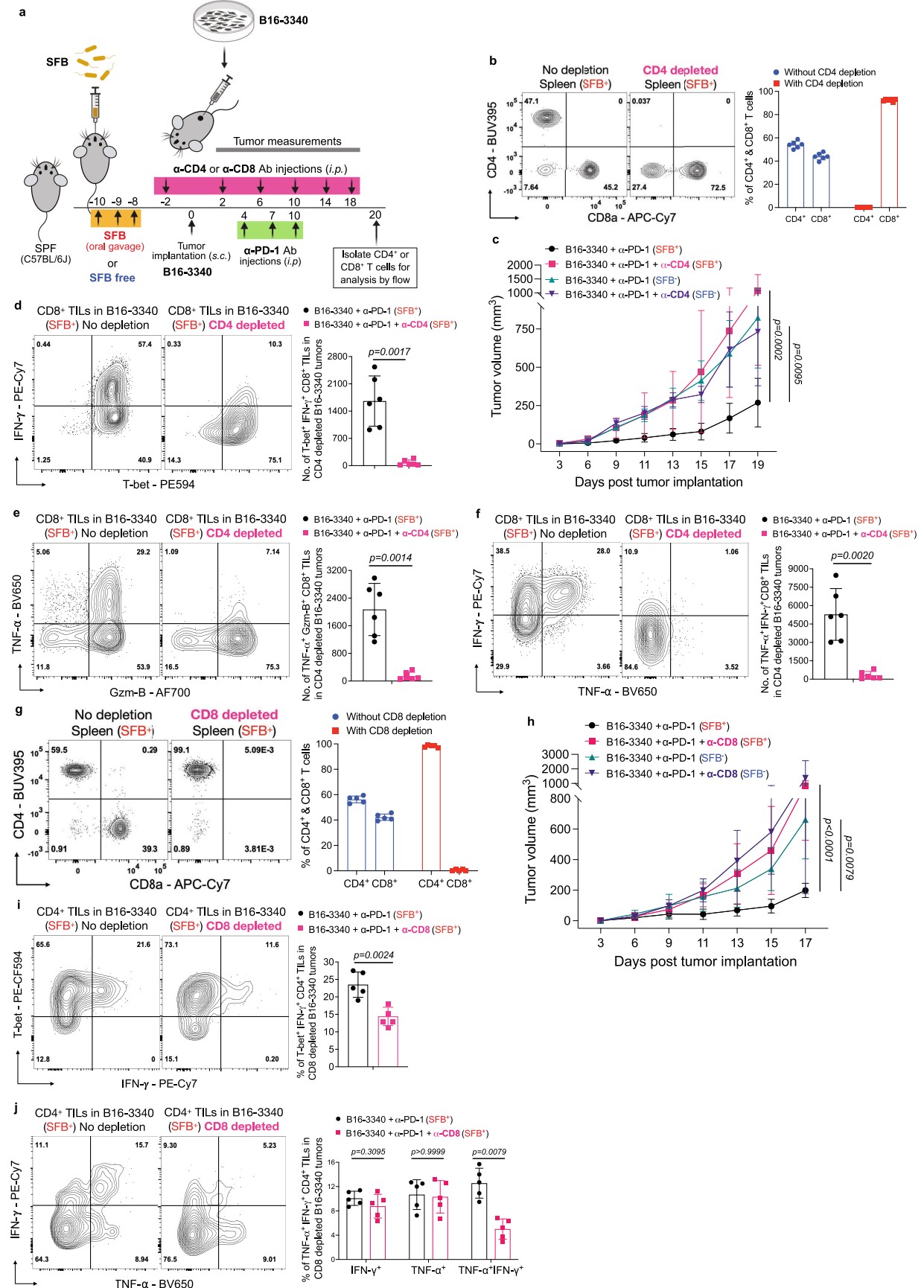

**Extended Data Fig. 4 |** See next page for caption.

**Extended Data Fig. 4 | Effects of in vivo CD4 and CD8 T cell depletion on tumor growth and TIL effector function in SFB⁺ and SFB⁻ mice treated with anti-PD-1 antibody.** (**a**) Schematic representation of in vivo CD4 or CD8 T cell depletion in SFB⁺ and SFB⁻ mice implanted with B16-3340 tumors and treated with anti-PD-1 antibody. Panels (**b**, n = 6) and (**g**, n = 5) show the efficacy of CD4⁺ or CD8⁺ T cells depletion, respectively. All mice received 3 injections of anti-PD-1 Ab (250 μg/mouse *i.p.* on days 4, 7 and 10 post tumor implantation) with or without in vivo depleting anti-CD4 or anti-CD8 monoclonal antibody administered twice per week (200 μg/mouse *i.p.*, on days −2, 2, 6, 10, 14 and 18 post-tumor implantation). (**c**) Growth curves (mean ± s.d.) of B16-3340 tumors in SFB⁺ and SFB⁻ mice with or without depletion of CD4⁺ T cells (n = 5 mice per group). (**d-f**) Representative flow plots (left) and quantification (right) showing T-bet and effector cytokine expression in CD8⁺ TILs from CD4⁺-depleted SFB⁺ mice treated with anti-PD-1 (n = 6 mice per group). (**g-j**) Reciprocal analysis following CD8 T cell depletion: (**h**) Tumor growth curves (mean ± s.d.) in SFB⁺ and SFB⁻ mice with or without depletion of CD8⁺ T cells (n = 5 mice per group) and (**i,j**) representative flow plots (left) and quantification (right) of T-bet and effector cytokine expression in CD4⁺ TILs from CD8⁺-depleted SFB⁺ mice treated with anti-PD-1 (n = 5 mice per group). Statistical significance for panels (**c**) and (**h**) was determined using two-way ANOVA and Sidak's multiple comparisons. Bar graphs show mean ± s.d., and each data point representing an individual mouse. Statistical analysis used unpaired two-sided Mann-Whitney t-test, and exact *P*-values are shown on the graphs. Schematic in **a** was created with BioRender (https://biorender.com).

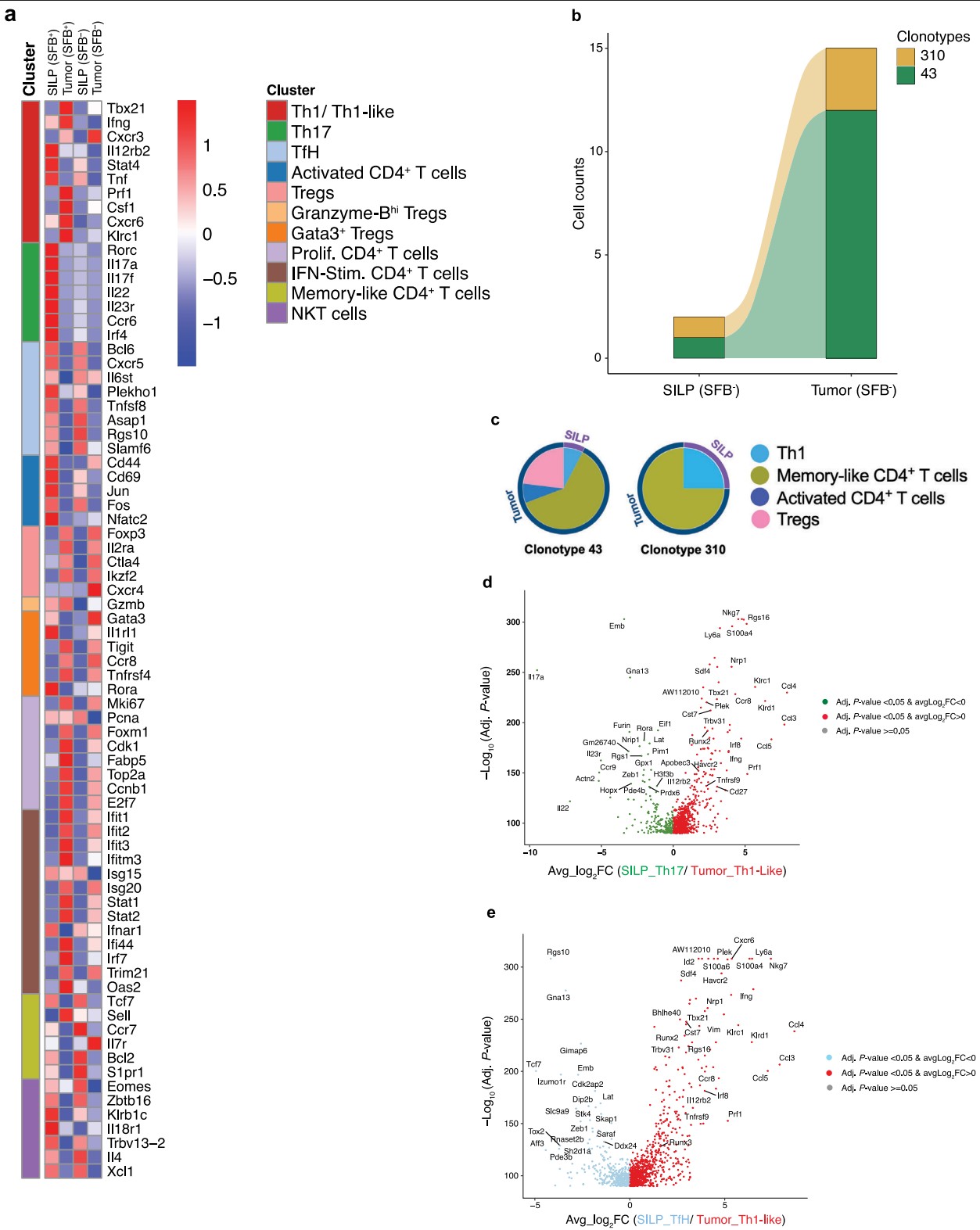

**Extended Data Fig. 5 | Gene expression in CD4⁺ T cells within SILP and tumors.**
(a) Heatmap showing normalized gene expression scaled by row (gene) of top differentially expressed and key cell lineage marker genes. (b) An alluvial graph depicting the clonal connectivity of CD4⁺ T cell clonotypes between the SILP and tumor tissues in SFB-free mice. Each block in the bar diagram represents cell counts within a distinct CD4⁺ T cell clonotype, with branches in the graph illustrating the shared clonotypes between SILP and tumor compartments. (c) Clonal expansion with phenotypic switching (represented by color) within

the tumor for two shared clonotypes originating from the gut in SFB-free mice. (d,e) Volcano plots highlighting gene expression differences between Th17 (b) and TfH (c) cells in the SILP and Th1-like cells in the tumor of SFB⁺ mice. Statistical significance was determined using the MAST package, with color coding indicating the magnitude of change: Red for upregulated genes and green or light-blue for downregulated genes (only significant genes are shown, genes with adjust-p-value >=0.05 are not included in the plot).

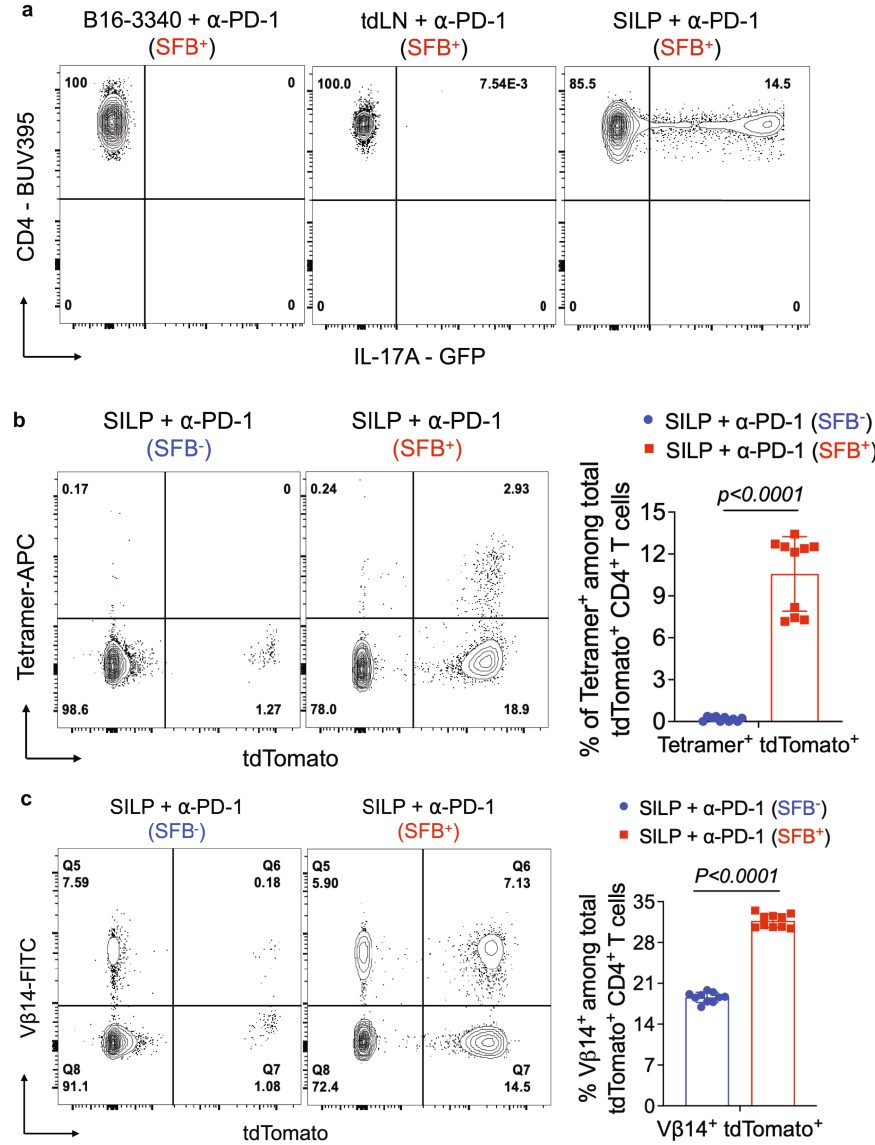

**Extended Data Fig. 6 | Characterization of *Il17a* fate mapped T cells in the small intestine of SFB-colonized mice. (a)** GFP expression in CD4+ T cells isolated from B16-3340 tumor tissue, tumor draining lymph node (TdLN) and SILP of IL-17A-GFP reporter mice colonized with SFB and treated with anti-PD-1 antibody. **(b)** Current or previous *Il17a* expression (tdTomato+) among SFB tetramer+ CD4+ T cells in the SILP of *tdTomato-ON*^ΔIL-17a fate-mapped mice with and without SFB colonization (n = 10 per group). **(c)** Vβ14+ T cells among *tdTomato-ON*^ΔIL-17a fate-mapped CD4+ cells in SILP of SFB+ mice compared to SFB− mice (n = 10 in each group). In (**b**) and (**c**), mice in both the SFB− and SFB+ groups received three doses of anti-PD-1 antibody. Statistical significance shown in (**b**) and (**c**) was determined using unpaired two-sided Mann-Whitney t-test. Error bars denote mean ± SD, and *P*-values are indicated on the corresponding graphs. Data are representative of at least two independent experiments.

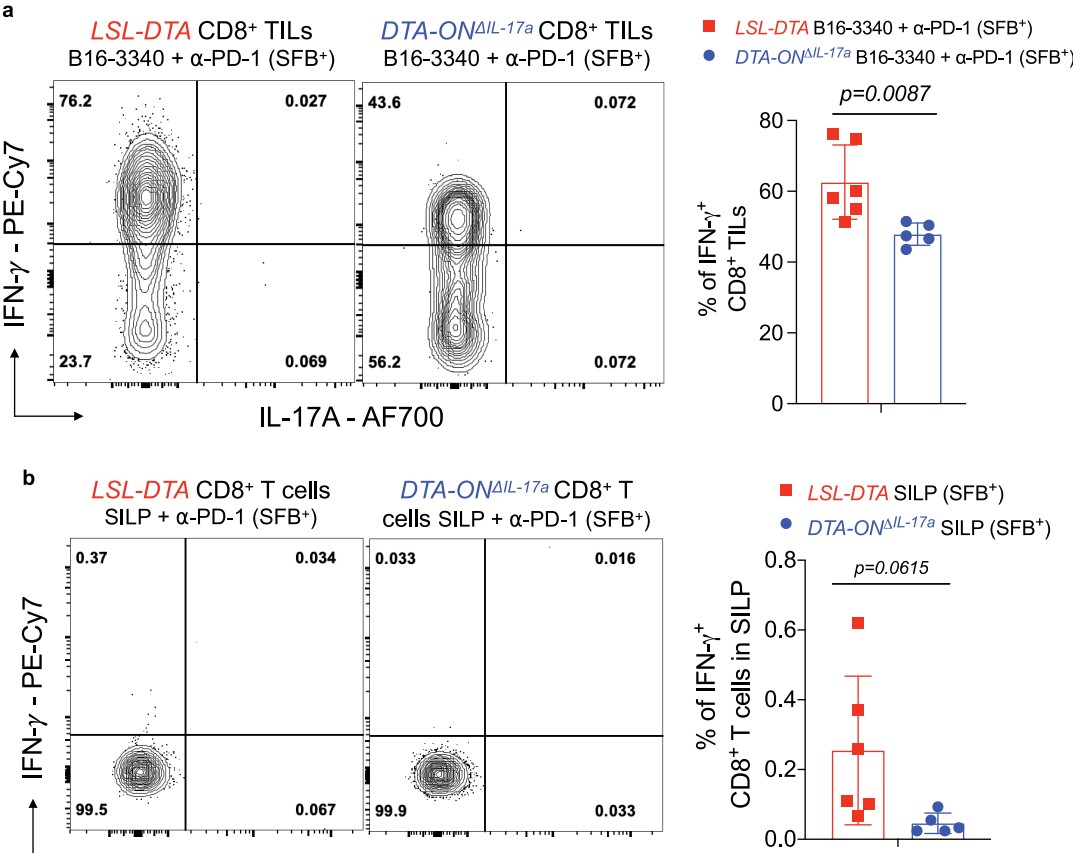

**Extended Data Fig. 7 | Loss of IL-17A-lineage cells reduces CD8+ IFN-γ responses in B16-3340 tumors. (a,b)** Intracellular cytokine staining for IFN-γ and IL-17A in CD8+ TILs from B16-3340 tumors (**a**) and in CD8+ T cells from SILP (**b**) of SFB-colonized *LSL-DTA* control (n = 6) and *DTA-ON^{IL-17A}* (n = 5) mice following anti-PD-1 treatment. Left panels show representative flow cytometry plots; right panels display quantification of cytokine-expressing cell frequencies. Bar graphs represent mean ± s.d., with each data point representing an individual mouse. Statistical comparisons were performed using two-sided, unpaired Mann-Whitney t-test, with *P*-values shown on the graphs. Data are representative of at least two independent experiments.

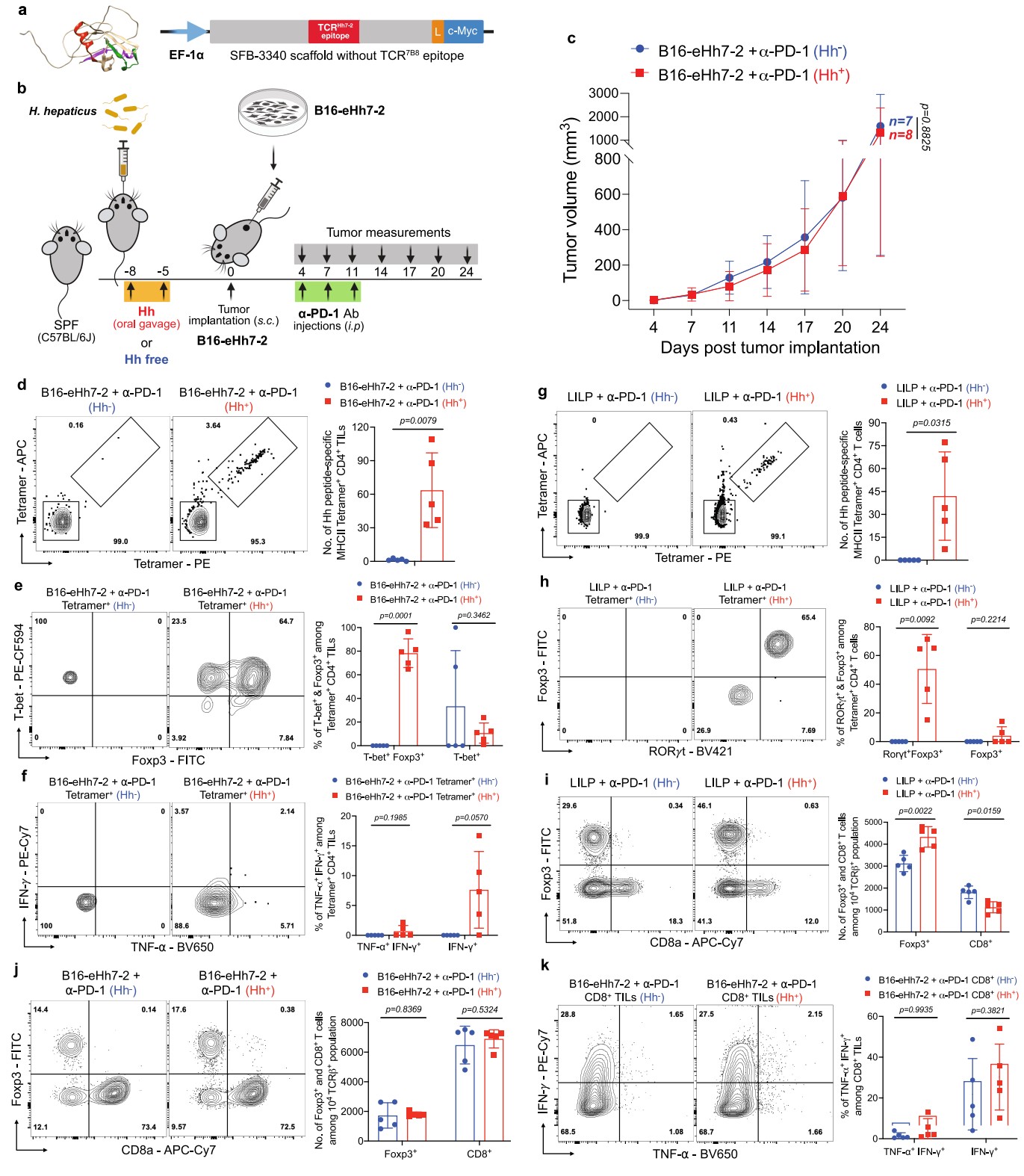

**Extended Data Fig. 8 |** See next page for caption.

**Extended Data Fig. 8 | *H. hepaticus* expands Hh-specific CD4+ T cells in distal tumors but fails to enhance anti-PD-1 efficacy.** (**a**) Schematic of engineered antigen construct: the Hh7-2 epitope from *H. hepaticus* is grafted onto an SFB-3340 scaffold (the native SFB TCR[7B8] epitope on the scaffold was disrupted) for expression in B16-F10 cells (B16-eHh7-2). (**b**) Experimental design testing the *H. hepaticus* artificial mimicry model in SPF C57BL/6 J mice. B16-eHh7-2 cells were subcutaneously implanted in mice colonized with Hh (Hh+) or Hh-free (Hh−), treated with anti-PD-1 antibody. (**c**) Tumor growth curves (mean ± s.d.) of B16-eHh7-2 tumor bearing mice treated with anti-PD-1: Hh− (blue, n = 7) and Hh+ (red, n = 8). Statistical significance was determined by two-way ANOVA with Sidak's multiple comparisons. (d) Hh peptide-specific MHC-II tetramer staining of tumor-infiltrating CD4+ T cells from Hh− and Hh+ mice treated with anti-PD-1 (n = 5 mice per group). (**e,f**) Representative flow cytometry analysis of tetramer+ CD4+ TILs showing T-bet and Foxp3 expression (**e**), and TNF-α and IFN-γ production following ex vivo stimulation (**f**) (n = 5 mice per group). (**g,h**) Hh peptide-specific MHC-II tetramer staining (**g**) and analysis of RORγt and Foxp3 expression (**h**) in tetramer+ CD4+ T cells from the LILP of Hh− and Hh+ mice following anti-PD-1 treatment (n = 5 mice per group). (**i-k**), Foxp3 and CD8 staining (left) and quantification among $10^4$ TCRβ+ T cells in LILP (**i**) and B16-eHh7-2 tumors (**j**) of Hh− and Hh+ mice, and TNF-α and IFN-γ expression following ex vivo stimulation of CD8+ TILs (**k**) (n = 5 mice per group). In (**d-j**), bar graphs show the mean ± s.d. corresponding to the representative flow cytometry plots, with each point representing an individual mouse. Statistical significance was determined using unpaired two-sided Mann-Whitney t-test, with *P*-values indicated in the corresponding graphs. Data are representative of two independent experiments. Schematics in **a** and **b** were created using BioRender (https://biorender.com).

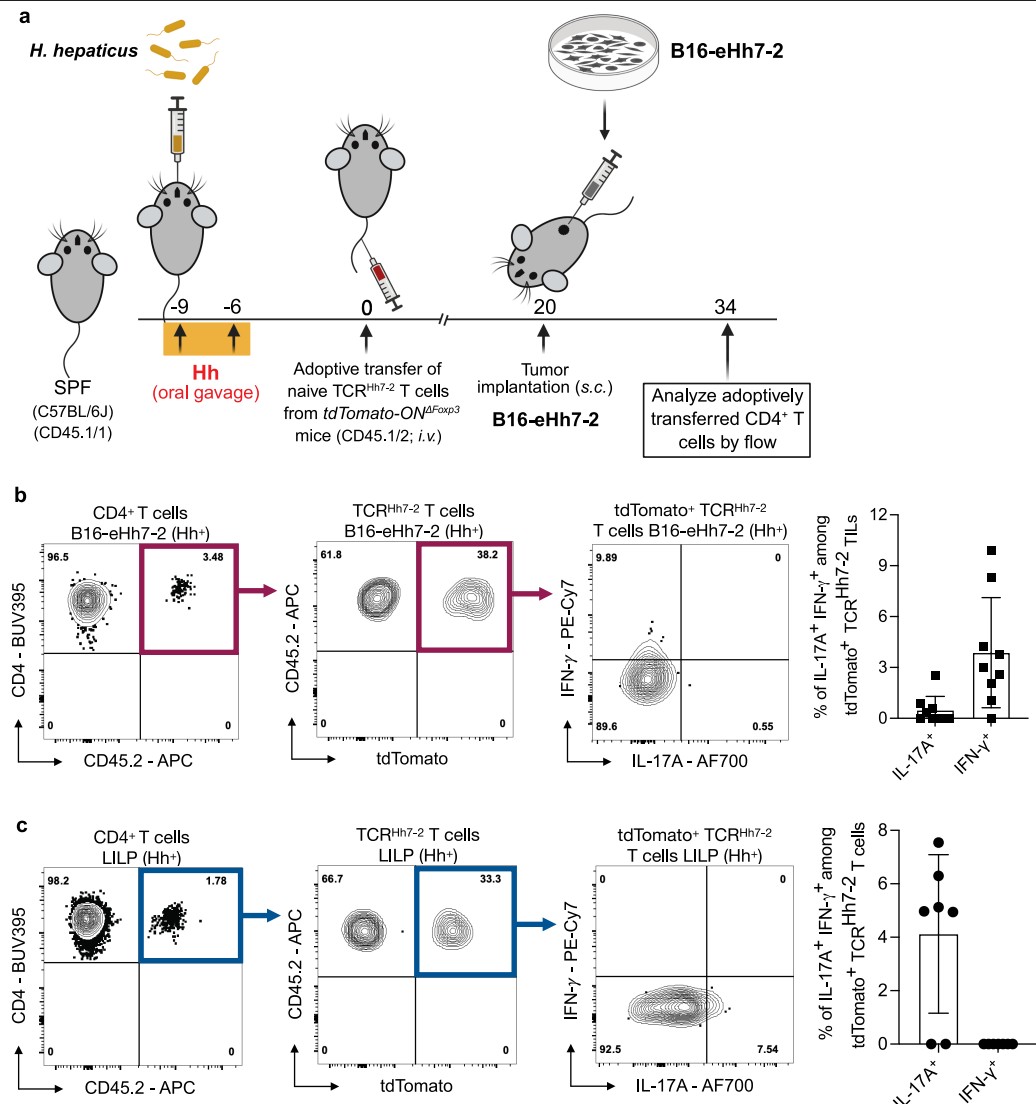

**Extended Data Fig. 9 | Tracking of *TCR^{Hh7-2} tdTomato-ON^{ΔFoxp3}* CD4⁺ T cells in gut mucosa and distal tumor tissue by Foxp3 fate-mapping. (a)** Schematic of the Foxp3 fate-mapping strategy using *Foxp3-Cre;ROSA-LSL-tdTomato TCR^{Hh7-2}* transgenic donor mice (*TCR^{Hh7-2} tdTomato-ON^{ΔFoxp3}*). Naïve Hh-specific TCR^{Hh7-2} CD4⁺ T cells were FACS sorted from donor mice and adoptively transferred into Hh-colonized (Hh⁺), congenic C57BL/6 recipients. Three weeks post-transfer, recipient mice were implanted with B16-eHh7-2 tumors and analyzed five weeks after adoptive transfer. **(b,c)** Analysis of adoptively transferred donor cells recovered from B16-eHh7-2 tumors (n = 9 mice) and LILP (n = 7 mice) of Hh⁺ recipients. Donor cells were identified by congenic markers (CD45.2/CD45.1) and gated as CD45.2 tdTomato⁺ TCR^{Hh7-2} CD4⁺ T cells. Representative flow plots show intracellular cytokine staining (IFN-γ and IL-17A) among gated tdTomato⁺ TCR^{Hh7-2} donor CD4⁺ TILs with quantification of IL-17A⁺ and IFN-γ⁺ cell frequencies in tumors **(c)** and LILP **(d)**. Bar graphs represent mean ± s.d.; each point denotes an individual mouse. Statistical significance was determined by unpaired two-sided Mann-Whitney test, with *P*-values indicated on the graphs. Schematic in **a** was created using BioRender (https://biorender.com).

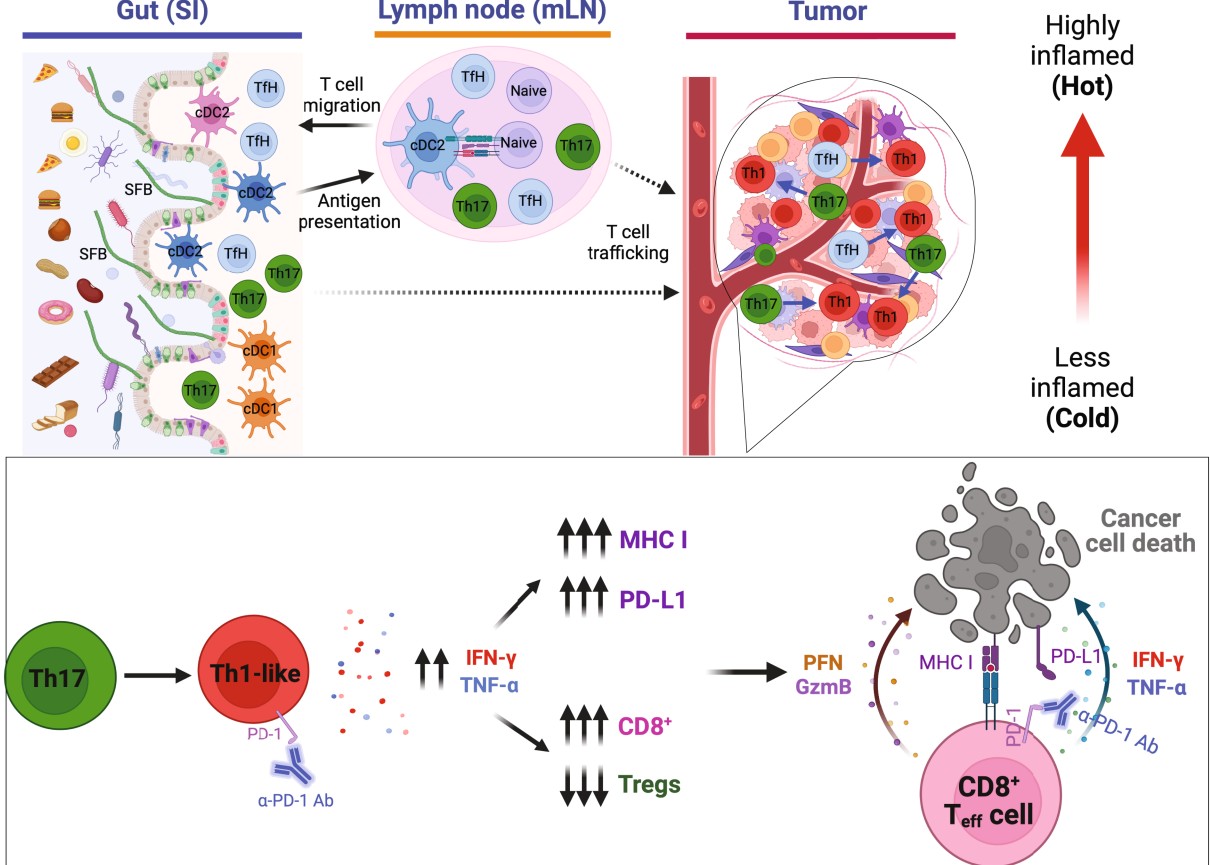

**Extended Data Fig. 10 | Schematic illustration of the role of gut commensal-primed T cells in enhancing immune checkpoint blockade through antigenic mimicry.** SFB colonization induces antigen-specific CD4[+], and likely CD8[+] T cells, which then distribute to other parts of the body but are retained and subsequently expand only in the tumor tissue which expresses SFB antigen (B16-3340 tumors). Th17 cells specific for SFB-3340 antigen transdifferentiate into Th1 cells, likely in the tumor tissue under the influence of the tumor microenvironment and produce proinflammatory cytokines IFN-γ and TNF-α, which aid in the infiltration and effector capabilities of cytotoxic T cells specific for the tumor. Schematic was created using BioRender (https://biorender.com).

# Reporting Summary

## Statistics

For all statistical analyses, confirm that the following items are present in the figure legend, table legend, main text, or Methods section.

| n/a | Confirmed | |
|---|---|---|
| ☐ | ☒ | The exact sample size (*n*) for each experimental group/condition, given as a discrete number and unit of measurement |
| ☐ | ☒ | A statement on whether measurements were taken from distinct samples or whether the same sample was measured repeatedly |
| ☐ | ☒ | The statistical test(s) used AND whether they are one- or two-sided<br>*Only common tests should be described solely by name; describe more complex techniques in the Methods section.* |
| ☐ | ☒ | A description of all covariates tested |
| ☐ | ☒ | A description of any assumptions or corrections, such as tests of normality and adjustment for multiple comparisons |
| ☐ | ☒ | A full description of the statistical parameters including central tendency (e.g. means) or other basic estimates (e.g. regression coefficient) AND variation (e.g. standard deviation) or associated estimates of uncertainty (e.g. confidence intervals) |
| ☐ | ☒ | For null hypothesis testing, the test statistic (e.g. *F*, *t*, *r*) with confidence intervals, effect sizes, degrees of freedom and *P* value noted<br>*Give P values as exact values whenever suitable.* |
| ☒ | ☐ | For Bayesian analysis, information on the choice of priors and Markov chain Monte Carlo settings |
| ☒ | ☐ | For hierarchical and complex designs, identification of the appropriate level for tests and full reporting of outcomes |
| ☒ | ☐ | Estimates of effect sizes (e.g. Cohen's *d*, Pearson's *r*), indicating how they were calculated |

*Our web collection on statistics for biologists contains articles on many of the points above.*

## Software and code

Policy information about availability of computer code

| | |
|---|---|
| Data collection | Flow cytometry data collection was performed with LSR II and Aria using FACSDiva v8.0.1 (BD Biosciences). scRNA and scTCR sequencing data was collected on a NovaSeq 6000. Immunoblot imaging data were collected using the LI-COR Odyssey CLx imaging system with Image Studio software (v1.0.18, LI-COR Biosciences). |
| Data analysis | Flow cytometry data were analyzed using Flowjo (v10.10.0). Satistical analyses was performed using Graphpad Prism V9. Immunoblot images were processed and quantified using LI-COR Image Studio software (v1.0.18). scRNA sequencing data were processed by Cell Ranger mkfastq from 10 Genomics (v7.1.0) using a custom reference package based on mouse reference genome mm10 2020-A (10x Genomics). For single-cell T cell receptor sequencing (scTCR-seq), data were aligned and quantified with CellRanger "multi" software (v.6.6.1, 10x Genomics) against the reference vdj_GRCm38_alts_ensembl-5.0.0, using default parameters. Gene expression matrix was processed ans analyzed using Seurat (v5.1.0) and Differential gene expression between groups was tested with the MAST package (MAST_1.28.0) as implemented in Seurat v5.1.0. Downstream analyses were performed in R (V4.4). All code used for analysis in this manuscript is publicly available at Zenodo (https://doi.org/10.5281/zenodo.17399749) . |

For manuscripts utilizing custom algorithms or software that are central to the research but not yet described in published literature, software must be made available to editors and reviewers. We strongly encourage code deposition in a community repository (e.g. GitHub). See the Nature Portfolio guidelines for submitting code & software for further information.

## Data

Policy information about availability of data

All manuscripts must include a data availability statement. This statement should provide the following information, where applicable:
- Accession codes, unique identifiers, or web links for publicly available datasets
- A description of any restrictions on data availability
- For clinical datasets or third party data, please ensure that the statement adheres to our policy

Mouse scRNA-seq and scTCR-seq data generated for this project, along with all code used for computational analysis, are publicly available at Zenodo (https://doi.org/10.5281/zenodo.17399749). Reference genomes mm10-2020-A (mouse) were used for mapping.

## Research involving human participants, their data, or biological material

Policy information about studies with human participants or human data. See also policy information about sex, gender (identity/presentation), and sexual orientation and race, ethnicity and racism.

| | |
|---|---|
| Reporting on sex and gender | N/A |
| Reporting on race, ethnicity, or other socially relevant groupings | N/A |
| Population characteristics | N/A |
| Recruitment | N/A |
| Ethics oversight | N/A |

Note that full information on the approval of the study protocol must also be provided in the manuscript.

# Field-specific reporting

Please select the one below that is the best fit for your research. If you are not sure, read the appropriate sections before making your selection.

☒ Life sciences    ☐ Behavioural & social sciences    ☐ Ecological, evolutionary & environmental sciences

For a reference copy of the document with all sections, see nature.com/documents/nr-reporting-summary-flat.pdf

# Life sciences study design

All studies must disclose on these points even when the disclosure is negative.

| | |
|---|---|
| Sample size | Sample sizes were not predetermined and the precise number of animals used are given in the figure legend. 5-10 mice per group were chosen based on previous experience of the intragroup validation of tumor growth upon similar treatments and common practice in the field, and animal welfare guidelines and availability of animals, while minimizing the use of animals in accordance with animal care guidelines from the NYU School of Medical Standing Committee on Animals and the National Institutes of Health. Similarly, group sizes for ex vivo experiments were determined from our previous experience in evaluating T cell driven immune responses to commensal microbes and from what is generally accepted in the field. (DOI: 10.1038/nature13279; https://doi.org/10.1038/nature25500; https://doi.org/10.1038/s41586-022-05089-y ) |
| Data exclusions | No samples were excluded from analysis. |
| Replication | All the findings on the main figures were replicated at least twice. The precise number of repeats are given in the figure legend. All attempts were successful. |
| Randomization | Allocation into sample groups was random. In addition, all control mice were from the same litter, same sex and housed in the same room to minimize microbiota-related variability. Both males and females were used for the experiments. |
| Blinding | Experiments were performed blinded whenever feasible. For certain tumor studies, the investigators responsible for colonizing mice with SFB and those measuring tumor sizes remained blinded until the experiment concluded. Tumor growth curves were only plotted after all measurements were collected. Blinding was not possible in some experiments due to potential SFB cross-contamination issues previously observed. Data were plotted and analyzed at the end of the experiment and measurements and analysis were performed by multiple people. |

# Reporting for specific materials, systems and methods

We require information from authors about some types of materials, experimental systems and methods used in many studies. Here, indicate whether each material, system or method listed is relevant to your study. If you are not sure if a list item applies to your research, read the appropriate section before selecting a response.

## Materials & experimental systems

| n/a | Involved in the study |
|---|---|
| ☐ | ☒ Antibodies |
| ☐ | ☒ Eukaryotic cell lines |
| ☒ | ☐ Palaeontology and archaeology |
| ☐ | ☒ Animals and other organisms |
| ☒ | ☐ Clinical data |
| ☒ | ☐ Dual use research of concern |
| ☒ | ☐ Plants |

## Methods

| n/a | Involved in the study |
|---|---|
| ☒ | ☐ ChIP-seq |
| ☐ | ☒ Flow cytometry |
| ☒ | ☐ MRI-based neuroimaging |

# Antibodies

| | |
|---|---|
| Antibodies used | In vivo anti-PD-1 monoclonal antibody (clone RMP1-14) was purchased from BiXcell. The following monoclonal antibodies were purchased from eBiosciences, BD Pharmingen or BioLegend: CD4 BUV395(GK1.5), BD 563790, 1:400; CD25 APC (PC61), Thermo Scientific 17-0251-82, 1:400; CD69 PE-Cy7 (H1.2F3), BioLegend 104512, 1:200; CD44 AF700 (IM7), BD 560567, 1:200; CD44 BV510 (IM7), BD 563114, 1:200; CD45.1 BV650(A20), BD563754, 1:400; CD45.2 FITC (104), eBioscience 11-0454-85, 1:400; CD19 PerCP-Cyanine5.5 (1D3), Tonbo Bioscience 65-0193-U100, 1:400; CD45R/B220 PerCP-Cyanine5.5 (RA3-6B2), Invitrogen 45-0452-82, 1:400; CD11c PerCP-Cyanine5.5 (N418) Invitrogen 45-0114-82, 1:400;  CD11b PerCP-Cyanine5.5 (M1/70) Invitrogen 45-0112-82, 1:400; MHCII I-A/I-E PerCP-Cyanine5.5 (M5/114.15.2), BioLegend 107626, 1:400; NK1.1 PerCP-Cyanine5.5 (PK136), Invitrogen 45-5941-82, 1:200; TCRβ BV711 (H57-597), BD 563135, 1:200; TCRγδ PerCP-Cyanine5.5 (GL3), BioLegend 118117, 1:400; FOXP3 FITC (FJK-16s), eBioscience 11-5773-82, 1:200; RORγt BV421 (Q31-378), BD 562894, 1:200; T-BET PE-CF594 (O4–46), BD 562467, 1:70; IL-17A AF700 (TC11-18H10.1) BioLegend 506914, 1:200; IFN-⯑ PE-Cy7 (XMG1.2), BioLegend 505826, 1:200; Granzyme B AF700 (QA16A02), BioLegend 372222, 1:200; TNF-⯑ BV650 (MP6-XT22), BioLegend 506333, 1:200; CD11c PE-Cy7 (N418), BioLegend 117318, 1:400; CD11b BUV395 (M1/70), BD 563553, 1:400; CXCR6 PE/dazzle 594 (SA051D1), BioLegend 151117, 1:200; CD62L PE (MEL-14) BD, Pharmingen 553151, 1:400; TCR Vβ14 FITC (14-2), BD, Pharmingen 553258, 1:400; 4′,6-diamidino-2-phenylindole (DAPI) or Live/dead fixable blue (ThermoFisher) was used to exclude dead cells.<br>For single-cell TCR sequencing (scTCR-seq) coupled with scRNA-seq, the following antibodies were used:<br>TotalSeq-C0301 anti-mouse Hashtag 1 Antibody (M1/42; 30-F11) BioLegend 155861, 1:100;<br>TotalSeq-C0302 anti-mouse Hashtag 2 Antibody (M1/42; 30-F11) BioLegend 155863, 1:100;<br>TotalSeq-C0303 anti-mouse Hashtag 3 Antibody (M1/42; 30-F11) BioLegend 155865, 1:100;<br>TotalSeq-C0304 anti-mouse Hashtag 4 Antibody (M1/42; 30-F11) BioLegend 155867, 1:100;<br>TotalSeq-C0305 anti-mouse Hashtag 5 Antibody (M1/42; 30-F11) BioLegend 155869, 1:100;<br>TotalSeq-C0306 anti-mouse Hashtag 6 Antibody (M1/42; 30-F11) BioLegend 155871, 1:100; |
| Validation | Only commercially available antibodies were used in the entire study. All commercially available antibodies are routinely tested by the vendor. |

# Eukaryotic cell lines

Policy information about cell lines and Sex and Gender in Research

| | |
|---|---|
| Cell line source(s) | B16-F10 (ATCC #CRL-6475), LLC1 (ATCC #CRL-1642) purchased from ATCC and MC-38 (kindly gifted by Dr. Kwok-Kin Wong from NYU School of Medicine) mouse tumor cell lines were used in this study. |
| Authentication | Cell lines were verified by the manufacturer's websites and regularly checked by morphology. B16-F10 and LLC1 cell lines used in this study were purchased from ATCC. MC-38 cell line was kindly provided by Dr. Kwok-Kin Wong from NYU School of Medicine. MC-38 cell line had been previously validated through whole exome sequencing by Dr. Kwok-Kin Wong lab. |
| Mycoplasma contamination | All cell lines used in this study were periodically tested for Mycoplasma contamination using the PCR Mycoplasma Detection Kit from Applied Biological Materials (abm). The tests consistently confirmed that the cell lines were Mycoplasma-free. |
| Commonly misidentified lines (See ICLAC register) | None |

# Animals and other research organisms

Policy information about studies involving animals; ARRIVE guidelines recommended for reporting animal research, and Sex and Gender in Research

| | |
|---|---|
| Laboratory animals | Specific pathogen-free (SPF) C57BL/6J (B6) mice (Jax #000664, both sexes, 6-7 weeks old) were obtained from The Jackson Laboratories. 7-8 weeks old CD45.1 congenic mice. (B6.SJL-Ptprca Pepcb/BoyJ, JAX #002014), Rosa-CAG-LSL-tdTomato reporter mice (B6;129S6-Gt(ROSA)26Sortm14(CAG-tdTomato)Hze/J, JAX #007908), Il17a-Cre mice (STOCK Il17atm1.1(icre)Stck/J, JAX #016879) and ROSA26-eGFP-DTA (B6.129S6(Cg)-Gt(ROSA)26Sortm1(DTA)Jpmb/J, JAX #032078) were purchased from The Jackson Laboratory. TCR7B8 (C57BL/6-Tg(Tcra,Tcrb)2Litt/J) and TCRHh7-2 (C57BL/6-Tg(Tcra,Tcrb)5Litt/J) mice were generated in-house as previously |

| | described. |
|---|---|
| Wild animals | No wild animals were used in the study |
| Reporting on sex | Both male and female mice were used; however, each experiment included same-sex controls, and no sex-specific differences were observed. |
| Field-collected samples | No Field-collected samples were used in the study |
| Ethics oversight | All animal procedures were performed in accordance with protocols approved by the Institutional Animal Care and Usage Committee of New York University School of Medicine. |

Note that full information on the approval of the study protocol must also be provided in the manuscript.

## Plants

| | |
|---|---|
| Seed stocks | N/A |
| Novel plant genotypes | N/A |
| Authentication | N/A |

## Flow Cytometry

### Plots

Confirm that:

☒ The axis labels state the marker and fluorochrome used (e.g. CD4-FITC).

☒ The axis scales are clearly visible. Include numbers along axes only for bottom left plot of group (a 'group' is an analysis of identical markers).

☒ All plots are contour plots with outliers or pseudocolor plots.

☒ A numerical value for number of cells or percentage (with statistics) is provided.

### Methodology

| | |
|---|---|
| Sample preparation | For isolation of cells from lymph nodes and spleens, tissues were mechanically disrupted with the plunger of a 1-ml syringe and passed through 70-µm cell strainers. Red blood cells were lysed with ACK buffer (Thermo Fisher).<br>For tumor-infiltrating lymphocyte isolation, tumors were collected around days 17–18 after implantation, minced and dissociated in digestion buffer, (RPMI containing collagenase (250 U ml–1 of type 1 collagenase; STEMCELL technologies), DNase I (100 ug/mL; Sigma), dispase (0.1 U/ml; Worthington) and 10% FBS with constant stirring at 37°C 30 min. After filtration, the lymphocytes were then isolated by Percoll density gradient (40%/80%) centrifugation at 800g for 20 min without brake. The interface of the Percoll layers were recovered for further analyses.<br>For isolation of lymphocytes from the SILP, the entire small intestine was dissected from mice. Mesenteric fat and Peyer's patches were carefully removed from these tissues. Intestinal tissue was opened and extensively cleaned of fecal matter. This tissue was sequentially treated with HBSS 1× (1 mM DTT) at 37°C for 10 min with gentle shaking (200 rpm), and twice with 5 mM EDTA at 37°C for 10 min to remove epithelial cells. The remaining tissue was then minced with scissors and dissociated in RPMI containing 10% FBS, dispase (0.05 U/ml; Worthington), collagenase (1 mg/ml collagenase II; Roche) and DNase I (100 ug/ml; Sigma) with constant shaking at 37°C for 45 min (175 rpm). The digested tissue was then filtered through a 70-µm strainer to remove large debris. Viable lamina propria lymphocytes were collected at the interface of a 40%/80% Percoll/RPMI gradient (GE Healthcare). |
| Instrument | Flow cytometric analysis was performed on an LSR II (BD Biosciences) or an Aria II (BD Biosciences). |
| Software | We used FACSDiva software to collect data, and performed analysis using FlowJo software (Tree Star). |
| Cell population abundance | Sort purity was determined to be greater than 95% by running post sort sample. |
| Gating strategy | Naïve 7B8 TCR-tg and Hh7-2 TCR-tg T cells were sorted as DAPI-CD4+TCRβ+CD44low/-CD62L+CD25−Vβ14+ (7B8) and Vβ6+ (Hh7-2) |

☒ Tick this box to confirm that a figure exemplifying the gating strategy is provided in the Supplementary Information.

