## [Peer Review File · Nature]

Microbiota-induced T cell plasticity enables immune-mediated tumor control

Corresponding Author: Dr Dan Littman

Version 0:

Reviewer comments:

Referee #1

(Remarks to the Author)

In this study, the authors investigated whether molecular mimicry between a commensal microbial antigen and a tumor antigen would be beneficial for anti-tumor immunity and response to immune checkpoint blockade. B16-F10 melanoma cells were engineered to express an antigen from the commensal bacteria segmented filamentous bacteria (SFB). Specific pathogen-free (SPF) mice were colonized SFB (or left SFB-free), inoculated with the SFB-Ag-expressing B6-F10 melanoma or the parental cell line and treated +/- with anti-PD-L1 (ICB). The authors found that only SFB+ mice given SFB-Ag-expressing B6-F10 cells responded to ICB leading to reduced tumor size. In the absence of ICB, the presence or absence of SFB did not affect tumor growth. In line with reduced tumor growth, there was an increase in intra-tumor CD8+ T cells and a reduction in Treg in SFB+ mice given SFB-Ag-expressing B6-F10, leading to a reduced ratio of intra-tumoral (but not small intestinal) CD8:Treg. Similarly, there was increased expression of effector molecules (TNFa, IFNg, Granzyme B) in intratumoral T cells in SFB+ mice given SFB-Ag-expressing B6-F10. The authors also found increased infiltration of SFB-specific CD4+ T cells into the tumor. These TIL CD4+ T cells expressed Tbet or Tbet and RORgt and produced IFNg, in contrast to the small intestine where the SFB-Specific CD4+ T cells predominantly expressed RORgt and produced IL-17 (as expected by SFB colonization). Depletion of CD4 or CD8 led to loss of tumor control and reduced effector function of the remaining TIL. To better assess the relationship between CD4+ T cells in the SI and TIL, the authors FACS-sorted CD4+ T cells from both sites (in SFB+ and SFB- mice bearing SFB-Ag expressing melanoma). Consistent with the previous FACS data, in SFB+ mice the SI has increased Th17 cells and the tumor has increased Th1-like TIL. Interestingly, scTCRseq showed matched clonotypes between SI Th17 and TFH and Tumor Th1-like. Using IL-17 fate-mapping, the authors show that many (20-60/50%) of the Tetramer+ (therefore SFB-specific) CD4+ TIL had previously expressed IL-17. Finally, SFB-specific (TCR7B8) IL-17-fate-map CD4+ T cells were adoptively transferred into SFB+ mice and then given SFB-expressing melanoma cells. Approx 50% of the transferred cells in the tumor had previously expressed IL-17 and many of these (20-70%) now expressed IFNg.

Overall, this study convincingly shows that molecular mimicry between a gut commensal and a tumor antigen is sufficient to boost the efficacy of ICB therapy. The experiments are performed well and the data clear. I have a few comments to help clarify the presentation of the data and suggest one experiment that could provide information as to whether this pathway could be utilized therapeutically.

Comments:

- The authors comment in the abstract and introduction that 'a mechanistic understanding of how gut commensal bacteria influence the efficacy of ICB remains elusive' and 'there is little understanding of mechanisms by which the intestinal microbiota composition influences anti-tumor immune responses'. This is a bit misleading as there have been several studies published that have illustrated mechanisms by which the commensal microbiota influence the efficacy of ICB and the authors should cite these articles (eg. PMID: 34624222, PMID: 36083892, PMID: 32792462, PMID: 34861182, PMID: 35278352, to name only a few). Providing these citations does not in any way diminish their findings.
- In Fig. 1h, how many mice in SFB+ and SFB- mice were re-challenged with SFB-Ag-expressing tumor? The graph states n=10 but the initial tumor challenge was only done in 10 mice and only 10% (i.e. n=1) survived in the SFB- group. To understand the strength of the data the numbers need to be explained more clearly. The data also suggests that the SFB-

mice that survived the original tumor and anti-PD-1 treatment were as protected during re-challenge as the SFB+ mice. The authors state 'a T cell memory response elicited by earlier SFB colonization, in combination with checkpoint blockade, was sufficient to restrict tumor growth', yet it seems this memory response is elicited even in SFB- mice?

- Line 136 – the authors state that the 'frequency of both T-bet+Foxp3- and T-bet+Foxp3+ cells among total CD4+ Tetramer-T cell population was comparable' where is that data shown?
- It is interesting that Tetramer -ve CD4+ TIL in SFB+ mice express significantly increased IFN γ and TNF α compared to SFB-ve mice (both with SFB-Ag-expressing melanoma). They state that their data suggests that 'SFB-specific CD4+ proinflammatory T cells in the tumor may contribute to altering the tumor microenvironment' but at this point there is no evidence these are SFB-specific, just that they are induced only in SFB colonized mice.
- In Fig. 4, one assumes the adoptive transfer was not done in SFB-ve mice? This would have been a good control.
- In Fig. 4e a bar graph showing the quantification of CD45.2+TdTomato+ cells would be helpful to show variability. Same for Fig 4f.
- It would have been interesting to see if the therapeutic addition of SFB to SFB-ve mice that already had SFB-antigen expressing tumors and had failed anti-PD-1 treatment could rescue anti-tumor immunity. Alternatively (and perhaps easier to perform) would be to inoculate the tumor, wait until it was palpable and then provide SFB together (or just prior) to ICB. This would provide excellent evidence that molecular mimicry could be utilized therapeutically and provide insight into the strength of the response if the T cells are primed in the gut at the same time as the therapy.
- In lines 234-235 the authors could also add additional references for molecular mimicry and autoimmunity as there are additional important studies illustrating this (eg. PMID: 27621416, PMID: 31237334, PMID: 29053971 to name just a few).
- Throughout the study, the number of times each experiment was repeated is not provided in the figure legends and I could not find this information in the Methods.

Minor comments

- The figure for Extended Data Fig. 3 has four panels (a-d) yet the figure legend only explains (a), lists (b) twice and there is no (d). Also, one assumes that the cells shown in panel (d) are T-bet+ but this is not clear.
- In the legend for Extended Data Fig. 4 there is no mention of T-bet staining (panel c). Are the cells shown in panels d and e pre-gated on T-bet+ cells? This is not clear.
- In the legend for Extended Data Fig. 5 there is no mention of T-bet staining (panel c). Are the cells shown in panels d pre-gated on T-bet+ cells? This is not clear. It looks like anti-CD8-depletion had no effect on CD4+ T cell production of TNF α (single positive) – the quantification this could be shown in a bar graph.
- Were control mice for anti-CD4 or anti-CD8 depletion expts injected with isotype control antibodies? The figure legend for Extended Data Fig. 4 does not state anything and the figure legend for Extended Data Fig. 5 states PBS.

Referee #2

(Remarks to the Author)

This manuscript shows that by forcing the expression of a specific commensal-derived antigen in one tumor cell line, commensal-specific CD4 T cells orchestrate an efficient anti-tumor immune response in mice treated with aPD1 mAb. To achieve this, a tumor cell line was engineered to express SFB-3340 protein. This antigen derives from the murine commensal SFB and is recognized by CD4 T cells, in particular by Th17 cells. Here, the data show that mice colonized with SFB and treated with aPD-1 mAb have low tumor volumes compared to controls. Notably, these mice develop a "long-term" memory which efficiently protects the mice even in the absence of aPD-1 mAb treatment. Next, the characterization of the tumors reveals an increase in cytotoxic CD8 T cells and of SFB-specific Th1-Th17 cells. RNA and TCR scRNAseq then reveals a common TCR repertoire among some of the CD4 T cells in the intestine and in the tumor. Finally, combining tetramer staining and fate mapping mouse models, exTh17 cells are shown to be present in the tumors and produce IFN. Overall, this is an interesting set of data, but I believe some key caveats need to be taken into consideration and potentially addressed experimentally.

1) It remains unclear to me what the biological and clinical relevance of forcing the expression of a mouse specific commensal antigen in a tumor cell line is. Probably this should be better introduced and eventually addressed experimentally.

This model does not allow to test molecular mimicry.

In contrast, I believe that the approach presented might help to test the potential clinical application of engineering tumors to express commensal bacteria-derived antigens to elicit a "preexisting but latent" antitumor immune response. If this is the aim of the manuscript, one should try to engineer the tumors directly in vivo, for example using tumor-infiltrating bacteria (PMID: 17448724) or any other means. This would support the relevance of this study, because it could be then potentially translated in humans. Finally, one might propose—and subsequently test—the hypothesis that commensal bacteria can colonize distal tumors, something that remains controversial.

2) It is remarkable that commensal specific Th17 cells become antitumorigenic in mice. However, it remains unclear what the mechanism of action of aPD-1 is in this context. What is the cellular target? Th17 cells? It also remains unclear what the mechanisms are that maintain this "long-term" anti-tumor immunity. These aspects should be further investigated.

3) Finally, data from humans may be necessary to strengthen the relevance of this study, especially considering that such a beautiful and simple SFB-Th17 cell relationship is not present in humans (PMID: 26411289). Are intestinal derived, or even better commensal-specific Th17 cells found in melanoma (or any human tumors)? Is their presence predictive of aPD1 response?

Additional specific comments:

Fig.1

The experimental plan is logical, and the results presented here are strong. I simply suggest that the authors specify in the legend how many times the experiments, shown in e and h, have been repeated.

It would be ideal to test this approach using a different type of cancer cell line in an another orthotopic tumor mouse model. It is known that the tissue matters in regard to the tumor microenvironment. Alternatively the authors should revise their conclusions restricting them to melanoma. Finally, they should also highlight the caveat of using this mouse model.

Fig.2

The characterization of the TME is well done. The results are clear and the data showing the role of both CD4 and CD8 T cells are of interest.

However, to support the conclusions on Th17 cells, it is important to test their specific role by depleting these cells or even better their capacity to produce IFN or Tbet.

Fig.3

This is a deeper characterization of the TME in relation to the intestine using the scRNA and TCRseq. The way the data in d are shown is brilliant.

Fig.4

I believe that using photoconvertible mice (PMID: 34099917) is fundamental to directly prove the intestinal origin of the tumor-infiltrating Th17 cells, especially considering the transgenic expression of the relevant antigen is outside the intestine. At the moment, beside TCR sharing which remains an interesting correlation, the conclusions are partially based on the observation that these cells are found in the intestine and in the tumors, but not in the dLN (Extended Fig. 7).

Referee #3

(Remarks to the Author)

Najar et al. present an elegant and simple interrogation of the relationship between T cell responses formed against commensals and anti-tumor immunity. In their study they use a well described and studies SFB derived CD4 epitope and overexpress it in B16F10 tumor cells. They subsequently carefully analyze the anti-tumor immune response in SFB- and SFB+ mice. Using scRNAseq, TCR clonotype analysis and fate mapping they identify that in SFB+ mice Th17 T cells from the gut re-differentiate and acquire a Th1 T cell phenotype. In contrast clonotypes in tumors of SFB- mice are found in Tregs. While the study has some major shortcoming, there is value in the careful analysis of the T cell response.

Conceptual shortcomings:

As the authors state in their introduction a previous study showed a clear correlation between Staph epi – OVA colonization and increased anti-tumor immunity. This significantly reduces the novelty. While the authors could make an argument for gut commensals the fact that SFB is introduced shortly before tumor inoculation further dampens enthusiasm. Given the SFB was described as a stable commensal why not use a SFB+ colony which has life long SFB exposure. Timing here might be very important.

While the reviewer sees value in interrogation of gut commensal, tumor T cell cross-talk at baseline the current model resembles more a acute vaccination setting. Nonetheless, for completion the authors should also consider conducting an experiment with therapeutic SFB inoculation.

Major issues:

The entirety of the observation is based on one cancer cell line and one bacterial strain. While the latter could be rationed the authors should test at least 3 more cell lines to ensure generalizability. For the lack of several cell lines it should be carefully discussed how other bacterial strains might affect these dynamics.

The authors based their entire framework on the fact that cancer cells acquire a shared peptide with bacteria. But translationally there seems to be little evidence of this. Thus it would be important to include a SFB expressing GP100 for instance and determine whether similar observations can be made with a tumor associated self-antigen.

Minor issues:

The authors elegantly show that combination of SFB + anti-PD-1 can induced robust anti-tumor immunity and that a protective memory is formed. Given their fate mapping it would be interesting to determine whether Th1 T cells form memory or disappear and only a th17 gut population is maintained which subsequently aids a CD8 recall response.

Version 1:

Reviewer comments:

Referee #1

(Remarks to the Author)

In this revised study, the authors investigated whether molecular mimicry between a commensal microbial antigen and a tumor antigen would be beneficial for anti-tumor immunity and response to immune checkpoint blockade. The authors engineered a tumor cell line (B16-F10 melanoma) to express an antigen (the 3340 epitope) from the commensal bacteria segmented filamentous bacteria (SFB). They then performed a series of elegant experiments to show that the presence of SFB significantly enhanced the efficacy of anti-PD-1 immune checkpoint therapy. In the revised manuscript, they tested two additional tumor cell lines (Lewis lung carcinoma and MC38 colon adenocarcinoma) that they also engineered to express the SFB epitope 3340 and confirmed that the presence of SFB also led to significantly delayed tumor growth. They show that SFB colonization induces a homeostatic Th17 response in the small intestine, which then traffic to the tumor and convert to Th1-like cells with production of IFN-g and TNF- α within the tumor microenvironment. Alterations in the tumor microenvironment also led to increased recruitment, expansion and effector functions of tumor-specific CD8+ T cells. In the revised manuscript, the authors now also show that depletion of these Th17 cells led to loss of anti-PD-1 mediated tumor control, showing the requirement for SFB commensal induction of Th17 cells. In the revision, the authors also tested the therapeutic potential of adding SFB at the same time as anti-PD-1 therapy, or various timepoints after the initiation of the treatment, and showed that gavage with SFB increased efficacy of anti-PD-1 even when given as an adjuvant therapy, with the best effect achieved when given early. Finally, the authors tested the ability of a different bacteria (*Helicobacter hepaticus*) to increase ICB when the B16-F10 tumor cells expressed the Hh7-2 epitope. While *H. hepaticus* induced antigen-specific CD4+ T cells in the gut and these cells trafficked to distal tumors, they did undergo Th1-like effector conversion and failed to enhance ICB.

In this revised manuscript the authors adequately addressed all of my comments and performed several complex new experiments that provide additional insights into the ability of commensal bacteria to drive a CD4 T cell response that promotes ICB efficacy. I have no additional concerns and commend the authors for such a detailed response to reviews.

Referee #2

(Remarks to the Author)

I appreciate the new experiments performed overall, but believe that some aspects (see below) still need to either be experimentally addressed or the conclusions toned down.

General points:

1) "... not as a literal model of naturally occurring peptide sharing." I agree with the authors that this manuscript does not address molecular mimicry. It remains very much possible that microbiota specific Th17 cells do not respond to tumour antigens. I therefore believed that the "molecular mimicry" concept should not be part of this manuscript because it has not been tested and can even be confusing. I suggest the author focus more on their following concept: "...delivered through gut-resident bacteria, can be harnessed to stimulate durable, tumour-specific immunity and improve responses to checkpoint therapy."

2) The authors have speculated on the mechanistic actions of exTh17 cells in remodelling the tumour microenvironment in response to aPD1. However, this has not been addressed experimentally.

3) Human relevance. This point has not been experimentally addressed. I believe that this is a partial limitation and it should at least be discussed in the manuscript.

Specific points:

Fig.1

I would suggest showing either SD or SEM, but not a mix of the two, as, for example, in e and g. This should be applied throughout the entire manuscript.

Fig.2

The use of the DTA-ONDIL-17A is very well received. However, the authors should be cautious with their direct conclusions on IL-17A Th17 cells, since this construct is not specific for Th17 cells but affects all IL-17A-producing cells even before the tumour is inoculated. I would simply acknowledge this aspect and discuss why it is very likely that the effect is due to SFB Th17 cells.

Fig.4

I disagree that the tools used here allow the authors to "more directly trace the intestinal origin of tumour-infiltrating ex-Th17 cells". I believe that the use of mouse models with photoconvertible cells is a more direct approach. Nevertheless, based on all that is known about SFB Th17 cells, it is very likely that the ones found in the tumour come from the intestine and/or MALTs. Yet, there is no direct experiment testing this migration path in this manuscript, and it cannot be formally excluded that SFB Th17 cells, in a mouse model where tumour cells express SFB antigens, are primed outside the MALTs. So, if the photoconversion experiment is not going to be performed, this aspect should at least be addressed in the discussion, clearly stating the reasons why it is very plausible for YFP Th17 cells to be derived from the MALTs.

Ext data Fig.3

I believe there is no need to show a statistic in the presence of two replicates as was done in A. The results are clear based

on these few replicates and the stats do not add anything.

Ext Data Fig.4 and 5.

I would include in the manuscript the validation of the efficacy of depletion using aCD4 and aCD8 mAb.

Discussion

See above about molecular mimicry. What is the point to start the discussion with this topic?

Finally there is no ref 52 reported in the reference list.

Referee #3

(Remarks to the Author)

The authors have sufficiently addressed my concerns

Response to Reviewers' Comments.

Microbiota-induced plastic T cells enhance immune control of antigen-sharing tumors

Najar *et al.*

We thank the editors for the opportunity to respond to the reviewers' comments. In each case, our response is shown in blue text below the reviewer comment/question.

Referee #1 (Remarks to the Author):

In this study, the authors investigated whether molecular mimicry between a commensal microbial antigen and a tumor antigen would be beneficial for anti-tumor immunity and response to immune checkpoint blockade. B16-F10 melanoma cells were engineered to express an antigen from the commensal bacteria segmented filamentous bacteria (SFB). Specific pathogen-free (SPF) mice were colonized SFB (or left SFB-free), inoculated with the SFB-Ag-expressing B6-F10 melanoma or the parental cell line and treated +/- with anti-PD-L1 (ICB). The authors found that only SFB+ mice given SFB-Ag-expressing B6-F10 cells responded to ICB leading to reduced tumor size. In the absence of ICB, the presence or absence of SFB did not affect tumor growth. In line with reduced tumor growth, there was an increase in intra-tumor CD8+ T cells and a reduction in Treg in SFB+ mice given SFB-Ag-expressing B6-F10, leading to a reduced ratio of intra-tumoral (but not small intestinal) CD8:Treg. Similarly, there was increased expression of effector molecules (TNF α , IFN γ , Granzyme B) in intratumoral T cells in SFB+ mice given SFB-Ag-expressing B6-F10. The authors also found increased infiltration of SFB-specific CD4+ T cells into the tumor. These TIL CD4+ T cells expressed Tbet or Tbet and ROR γ t and produced IFN γ , in contrast to the small intestine where the SFB-specific CD4+ T cells predominantly expressed ROR γ t and produced IL-17 (as expected by SFB colonization). Depletion of CD4 or CD8 led to loss of tumor control and reduced effector function of the remaining TIL. To better assess the relationship between CD4+ T cells in the SI and TIL, the authors FACS-sorted CD4+ T cells from both sites (in SFB+ and SFB- mice bearing SFB-Ag expressing melanoma). Consistent with the previous FACS data, in SFB+ mice the SI has increased Th17 cells and the tumor has increased Th1-like TIL. Interestingly, scTCRseq showed matched clonotypes between SI Th17 and TFH and Tumor Th1-like. Using IL-17 fate-mapping, the authors show that many (20-60%) of the Tetramer+ (therefore SFB-specific) CD4+ TIL had previously expressed IL-17. Finally, SFB-specific (TCR7B8) IL-17-fate-map CD4+ T cells were adoptively transferred into SFB+ mice and then given SFB-expressing melanoma cells. Approx 50% of the transferred cells in the tumor had previously expressed IL-17 and many of these (20-70%) now expressed IFN γ .

Overall, this study convincingly shows that molecular mimicry between a gut commensal and a tumor antigen is sufficient to boost the efficacy of ICB therapy. The experiments are performed well and the data clear. I have a few comments to help clarify the presentation of the data and suggest one experiment that could provide information as to whether this pathway could be utilized therapeutically.

We thank the reviewer for the favorable remarks. Below, we have addressed the points raised.

Comments:

- The authors comment in the abstract and introduction that 'a mechanistic understanding of how gut commensal bacteria influence the efficacy of ICB remains elusive' and 'there is little understanding of mechanisms by which the intestinal microbiota composition influences anti-tumor immune responses'.

This is a bit misleading as there have been several studies published that have illustrated mechanisms by which the commensal microbiota influence the efficacy of ICB and the authors should cite these articles (eg. PMID: 34624222, PMID: 36083892, PMID: 32792462, PMID: 34861182, PMID: 35278352, to name only a few). Providing these citations does not in any way diminish their findings.

We thank the reviewer for highlighting these important references (PMID: 34624222, 36083892, 32792462, 34861182, 35278352). We have now cited them in the Introduction and Discussion to provide a more comprehensive overview of microbiota-immune interactions. While these studies elegantly describe the roles of microbial metabolites, innate pathways, and myeloid cell re-programming in shaping ICB responses, the specific contribution of antigenic mimicry, where commensal-derived epitopes directly cross-react with tumor antigens, has remained underexplored.

In our manuscript, the term "elusive" in the abstract and introduction specifically refers to the limited mechanistic understanding of antigenic mimicry, particularly how numerous commensal-derived peptides (such as those produced by SFB) may resemble tumor antigens and consequently activate cross-reactive T cells that modulate tumor immunity. While antigenic mimicry is a recognized concept, and previous studies have made important contributions to our understanding of the microbiota's impact on immune modulation, the specific involvement of antigen mimicry in enhancing ICB efficacy remains largely underexplored. Our study addresses this critical gap in understanding how gut commensals could modulate anti-tumor immunity, using segmented filamentous bacteria (SFB) as a model. We demonstrate that SFB colonization elicits cross-reactive Th17 cells in the gut, which then migrate to distal tumors and transdifferentiate into highly pro-inflammatory Th1-like cells. This cellular shift remodels the tumor microenvironment from an anti-PD-1 non-responsive to a responsive one characterized by reduced Treg populations and increased CD8⁺ T cell infiltration thereby promoting robust anti-tumor T cell responses. In alignment with clinical studies demonstrating that the gut microbiome influences responses to immunotherapy (PMID: 33542131; PMID: 33303685), our work highlights a distinct yet complementary mechanism: microbiota-derived antigens can elicit T cells with TCR cross-reactivity to tumor epitopes via antigenic mimicry, thereby enhancing the efficacy of PD-1 blockade. This mechanism is distinct from the pathways emphasized in earlier work and complements clinical evidence from FMT trials (PMID: 33542131, 33303685) by identifying a defined, antigen-driven route through which the microbiota can augment checkpoint therapy.

- In Fig. 1h, how many mice in SFB⁺ and SFB⁻ mice were re-challenged with SFB-Ag-expressing tumor? The graph states n=10 but the initial tumor challenge was only done in 10 mice and only 10% (i.e. n=1) survived in the SFB⁻ group. To understand the strength of the data the numbers need to be explained more clearly. The data also suggests that the SFB⁻ mice that survived the original tumor and anti-PD-1 treatment were as protected during re-challenge as the SFB⁺ mice. The authors state 'a T cell memory response elicited by earlier SFB colonization, in combination with checkpoint blockade, was sufficient to restrict tumor growth', yet it seems this memory response is elicited even in SFB⁻ mice?

We agree that the re-challenge experiment (Figure 1h) needed to be described more clearly. In the primary tumor challenge, both SFB⁺ and SFB⁻ cohorts (n=10 per group) were implanted with SFB antigen-expressing B16-3340 tumors and treated with anti-PD-1. In the SFB⁺ group, 7 out of 10 mice survived the primary challenge, while only 1 out of 10 mice survived in the SFB⁻ group. These surviving mice (7 from the SFB⁺ group and 1 from the SFB⁻ group) were subsequently re-challenged with the same tumor cells, but without further anti-PD-1 treatment. All animals, (7/7 from SFB⁺ group and 1/1

from SFB⁻ group), successfully rejected the tumor, demonstrating durable memory formation. While the rare survival of one SFB⁻ mouse suggests that checkpoint therapy alone can occasionally elicit memory, the markedly higher survival and stronger effector responses in SFB⁺ mice underscore the dominant role of SFB colonization in driving protective immunity. We have now revised Figure 1h and its legend to improve clarity. We have also updated the corresponding results section in the revised manuscript to better explain the experimental design and findings.

- Line 136 – the authors state that the ‘frequency of both T-bet+Foxp3⁻ and T-bet+Foxp3⁺ cells among total CD4⁺ Tetramer⁻ T cell population was comparable’ where is that data shown?

We apologize for any confusion regarding the location of the corresponding data. The relevant information is indeed included in the manuscript and is presented in **Extended Data Fig. 2h**. To improve clarity, we have revised both the figure legend and the main text to more explicitly direct readers to this result.

- It is interesting that Tetramer^{-ve} CD4⁺ TIL in SFB⁺ mice express significantly increased IFN γ and TNF α compared to SFB^{-ve} mice (both with SFB-Ag-expressing melanoma). They state that their data suggests that ‘SFB-specific CD4⁺ proinflammatory T cells in the tumor may contribute to altering the tumor microenvironment’ but at this point there is no evidence these are SFB-specific, just that they are induced only in SFB colonized mice.

We agree that the tetramer⁻ CD4⁺ TILs cannot be definitively assigned SFB specificity. However, multiple lines of evidence suggest they are shaped by SFB colonization: (i) scTCR-seq revealed shared clonotypes between SILP Th17/Tfh cells and tumor-infiltrating Th1-like CD4⁺ cells, (ii) fate mapping demonstrated IL-17A lineage origin for a substantial fraction of tumor CD4⁺ T cells, and (iii) adoptive transfer of TCR^{7B8} cells confirmed their Th17-to-Th1 transition. These data indicate that many tumor-infiltrating CD4⁺ T cells derive from SFB-primed precursors, and some tetramer⁻ TILs may recognize distinct epitopes within the SFB-3340 fragment (other than the 7B8 epitope on SFB-3340 that is not targeted by SFB tetramers). Endogenous tumor antigens may also be targeted by bystander CD4⁺ T cells that are influenced by the inflamed TME.

To further dissect the mechanistic relevance of SFB-induced Th17 cells, regardless of their epitope specificity, we employed IL-17A-driven DTA (*DTA-ON^{IL-17A}*) mice to selectively ablate IL-17A expressing cells in SFB⁺ mice (new **Figure 5**). Ablation of IL-17⁺ Th17 cells markedly impaired SFB-mediated tumor control in anti-PD-1 treated mice (**Figure 5a-b**), with a significant reduction in tetramer⁺ CD4⁺ T cells in the SILP and near-complete loss of IL-17A-producing antigen-specific cells (**Figure 5c**). This was accompanied by reduced numbers and IFN- γ production of tumor-infiltrating tetramer⁺ CD4⁺ T cells (**Figure 5d**), decreased CD8⁺ T cell infiltration, increased Foxp3⁺ Tregs, and diminished IFN- γ -producing CD8⁺ TILs (**Figure 5e-f; Extended Data Fig. 7a-b**). These results confirm the functional requirement of ex-Th17 cells for remodeling TME with concurrent CD8⁺ T cell recruitment and maturation, thus enhancing PD-1 therapy.

- In Fig. 4, one assumes the adoptive transfer was not done in SFB^{-ve} mice? This would have been a good control.

We agree that SFB⁻ mice would be a useful control. However, the primary objective of the experiment presented in Figure 4 was to specifically trace the migration of SFB-induced Th17 cells from the gut to

the tumor, and to examine their subsequent phenotypic transition from IL-17A-producing Th17 cells to IFN- γ -producing Th1-like cells. In SFB-colonized mice, adoptively transferred SFB-specific CD4⁺ T cells are activated in the gut in response to SFB-derived antigens, then migrate to the tumor site where they transdifferentiate into Th1-like effector cells that contribute to tumor control. In contrast, in SFB⁻ mice, the absence of SFB antigen precludes the activation of these naive TCR-transgenic cells in the gut. Without antigen-driven stimulation, these cells will fail to expand and ultimately undergo attrition over time, typically within 1-2 weeks, thus making it unfeasible to track the activation–migration–differentiation sequence that was the primary focus of this experiment. Therefore, we deliberately focused on SFB⁺ mice to capture this specific biological process in a physiologically relevant context.

- In Fig. 4e a bar graph showing the quantification of CD45.2+TdTomato⁺ cells would be helpful to show variability. Same for Fig 4f.

We thank the reviewer for this suggestion. We have now added bar graphs quantifying CD45.2+TdTomato⁺ cells in Figures 4e,f to better illustrate data variability and enhance clarity.

- It would have been interesting to see if the therapeutic addition of SFB to SFB-ve mice that already had SFB-antigen expressing tumors and had failed anti-PD-1 treatment could rescue anti-tumor immunity. Alternatively (and perhaps easier to perform) would be to inoculate the tumor, wait until it was palpable and then provide SFB together (or just prior) to ICB. This would provide excellent evidence that molecular mimicry could be utilized therapeutically and provide insight into the strength of the response if the T cells are primed in the gut at the same time as the therapy.

We thank the reviewer for this excellent suggestion. We performed staged post-implantation SFB gavage in mice bearing B16-3340 tumors (implant day 0) that received anti-PD-1 during the early palpable window (days 4-10). Cohorts were gavaged with SFB at days 8–12 (Group-1), 12-16 (Group-2) or 15-19 (Group-3), or left SFB-negative (no gavage, Group-4); results are now shown in new **Extended Data Fig. 1e,f**.

Early SFB administration (days 8–12) yielded the largest therapeutic benefit, with markedly reduced tumor growth versus SFB-negative controls and versus later gavage cohorts; the benefit declined progressively with delayed SFB delivery. These data show that (i) tumor expression of the SFB-derived neoantigen is required for microbiota-dependent enhancement of anti-PD-1 in our model, and (ii) there is a discrete post-implantation window, overlapping with early PD-1 blockade, during which microbe-driven priming of antigen-specific CD4⁺ T cells and their recruitment to the tumor site most potently synergizes with anti-PD-1 therapy.

- In lines 234-235 the authors could also add additional references for molecular mimicry and autoimmunity as there are additional important studies illustrating this (eg. PMID: 27621416, PMID: 31237334, PMID: 29053971 to name just a few).

We appreciate the recommendation to include additional references on molecular mimicry and its relevance to autoimmunity in the discussion section. We have incorporated the suggested citations (PMID: 27621416, PMID: 31237334, and PMID: 29053971) into the revised manuscript. These references help to provide a broader context for our findings and underscore the relevance of molecular mimicry not only in cancer immunotherapy but also in the pathogenesis of autoimmune diseases.

- Throughout the study, the number of times each experiment was repeated is not provided in the figure legends and I could not find this information in the Methods.

We have updated the figure legends in the revised manuscript to clearly indicate the number of independent experimental replicates performed.

Minor comments

- The figure for Extended Data Fig. 3 has four panels (a-d) yet the figure legend only explains (a), lists (b) twice and there is no (d). Also, one assumes that the cells shown in panel (d) are T-bet⁺ but this is not clear.

We thank the reviewer for noting the inconsistencies in the Extended Data Fig. 3 legend. We have revised the legend to accurately describe panels (a-d) and to eliminate any ambiguity regarding the cell populations shown. Specifically, the flow cytometry plots in Extended Data Fig. 3c and 3d display T-bet versus IFN- γ and TNF- α versus IFN- γ , respectively, gated on live TCR β^+ CD4⁺tetramer⁻ cells. We have also updated the panel labels to reflect this gating strategy clearly and have included the corresponding quantification for both plots in the revised figure and legend to improve clarity and completeness.

- In the legend for Extended Data Fig. 4 there is no mention of T-bet staining (panel c). Are the cells shown in panels d and e pre-gated on T-bet⁺ cells? This is not clear.

We thank the reviewer for pointing out the oversight and have revised the figure legend accordingly. We have now indicated that T-bet staining is included in panel (c), and clarified that the cells shown in panels (d) and (e) are **not** pre-gated on T-bet⁺ cells.

- In the legend for Extended Data Fig. 5 there is no mention of T-bet staining (panel c). Are the cells shown in panels d pre-gated on T-bet⁺ cells? This is not clear. It looks like anti-CD8-depletion had no effect on CD4⁺ T cell production of TNF α (single positive) – the quantification this could be shown in a bar graph.

Panel c displays T-bet versus IFN- γ staining in CD8-depleted and non-depleted SFB⁺ mice. To clarify further, the cytokine flow plots shown in Panel d were generated from total CD4⁺ tumor-infiltrating lymphocytes, gated on live, singlet, TCR β^+ CD4⁺ cells. These plots were not pre-gated on T-bet⁺ cells. We have updated the figure legend to explicitly describe the gating strategy used for all flow cytometry panels and note that Panel c depicts T-bet and IFN- γ co-staining.

Regarding the impact of anti-CD8 depletion on TNF- α -producing CD4⁺ T cells, we appreciate the reviewer's suggestion to quantify TNF- α single-positive populations. In the original analysis, we focused primarily on IFN- γ^+ and IFN- γ^+ TNF- α^+ double-positive CD4⁺ T cells, as this populations best reflect the Th1-like, cytotoxic-supporting helper phenotypes that were central to our mechanistic focus on CD8⁺ T cell-dependent tumor control. In contrast, the frequency of TNF- α single-positive CD4⁺ T cells remained relatively unchanged following CD8 depletion and therefore was not emphasized in the original figure. We have now included a bar graph quantifying all three populations: IFN- γ^+ , TNF- α^+ , and IFN- γ^+ TNF- α^+ CD4⁺ TILs.

- Were control mice for anti-CD4 or anti-CD8 depletion expts injected with isotype control antibodies?

The figure legend for Extended Data Fig. 4 does not state anything and the figure legend for Extended Data Fig. 5 states PBS.

In the CD4 and CD8 depletion experiments in Extended Data Figs. 4–5, we used PBS rather than isotype control antibodies. The depleting antibodies acted specifically without off-target effects, as shown in Fig. R1 below.

Fig. R1: In vivo CD4⁺ and CD8⁺ T cell depletion and gating strategy. (a-b) Validation of CD4⁺ T cell depletion. Shown are representative flow cytometry plots and gating strategy for identifying live, singlet TCRβ⁺ splenic T cells, following exclusion of DAPI⁺ dead cells and dump channel (non-T cells). CD4 and CD8 expression is subsequently analyzed. (a) Representative plot from a PBS-treated control mouse. (b) Representative plot from a mouse treated with anti-CD4 depleting antibody. Right panel: quantification of CD4⁺ and CD8⁺ T cell frequencies in PBS and anti-CD4 antibody treated mice. Anti-CD4 treatment efficiently reduced CD4⁺ T cells to near-background levels, with no off-target effect on CD8⁺ T cells. (c-d) Validation of CD8⁺ T cell depletion. Gating strategy as in (a-b). Representative flow cytometry plots are shown for PBS-treated (c) and anti-CD8 antibody treated (d) mice. Right panel: quantification of CD4⁺ and CD8⁺ T cell frequencies. Anti-CD8 treatment efficiently depleted CD8⁺ T cells with no impact on CD4⁺ T cells. In all quantification panels, each data point represents a single mouse; bars indicate mean ± s.e.m. (n = 5 mice per group).

Referee #2 (Remarks to the Author):

This manuscript shows that by forcing the expression of a specific commensal-derived antigen in one tumor cell line, commensal-specific CD4 T cells orchestrate an efficient anti-tumor immune response in mice treated with aPD1 mAb. To achieve this, a tumor cell line was engineered to express SFB-3340 protein. This antigen derives from the murine commensal SFB and is recognized by CD4 T cells, in particular by Th17 cells. Here, the data show that mice colonized with SFB and treated with aPD-1mAb have low tumor volumes compared to controls. Notably, these mice develop a “long-term” memory which efficiently protects the mice even in the absence of aPD-1 mAb treatment. Next, the characterization of the tumors reveals an increase in cytotoxic CD8 T cells and of SFB-specific Th1-

Th17 cells. RNA and TCR scRNAseq then reveals a common TCR repertoire among some of the CD4 T cells in the intestine and in the tumor. Finally, combining tetramer staining and fate mapping mouse models, exTh17 cells are shown to be present in the tumors and produce IFN. Overall, this is an interesting set of data, but I believe some key caveats need to be taken into consideration and potentially addressed experimentally.

1) It remains unclear to me what the biological and clinical relevance of forcing the expression of a mouse specific commensal antigen in a tumor cell line is. Probably this should be better introduced and eventually addressed experimentally.

This model does not allow to test molecular mimicry.

In contrast, I believe that the approach presented might help to test the potential clinical application of engineering tumors to express commensal bacteria-derived antigens to elicit a “preexisting but latent” antitumor immune response. If this is the aim of the manuscript, one should try to engineer the tumors directly *in vivo*, for example using tumor-infiltrating bacteria (PMID: 17448724) or any other means. This would support the relevance of this study, because it could be then potentially translated in humans. Finally, one might propose—and subsequently test—the hypothesis that commensal bacteria can colonize distal tumors, something that remains controversial.

We thank the reviewer for this important conceptual point. Our engineered-antigen tumor model is intended as a proof-of-principle platform to test whether gut-primed, commensal-specific CD4⁺ T cells can traffic to antigen-matched tumors, reprogram into pro-inflammatory effectors, and synergize with PD-1 blockade. Forced expression of a defined commensal epitope in tumor cells enables precise, causal dissection of these cell-biological events, migration, clonal overlap, local Th17→Th1 conversion, and downstream CD8⁺ T cell recruitment—that would be intractable in spontaneously arising, antigenically heterogeneous tumors. We therefore view this approach as a mechanistic complement to clinical microbiome–ICB observations (PMID: 33542131; PMID: 33303685), not as a literal model of naturally occurring peptide sharing.

Importantly, our data show that gut-induced T cells can be mobilized against tumors when a cognate microbial epitope is present, providing biological plausibility that commensal-derived antigens can prime or amplify tumor-reactive T cell responses in patients who show positive responses to ICB. These mechanistic insights directly inform translational strategies, such as *in-silico* epitope mining of human tumors, rational selection of donor strains for FMT, engineered probiotics that safely present tumor neoantigens, or peptide/mRNA vaccine platforms that mimic beneficial microbial epitopes. Indeed, recent successes in neoantigen vaccines (Rojas et al., Nature 2023) illustrate feasible paths to exploit peripheral microbial priming to bolster CD8⁺ T cell responses in poorly immunogenic tumors.

We also agree that *in vivo* tumor engineering via tumor-colonizing bacteria is an attractive translational route (PMID: 17448724). However, such strategies currently present nontrivial safety and technical challenges like host infection risk, systemic inflammation, and delivery control that require careful preclinical optimization. Overall, while our model uses an engineered antigen system to establish proof of concept, it highlights a potentially generalizable framework in which microbial antigens, delivered through gut-resident bacteria, can be harnessed to stimulate durable, tumor-specific immunity and improve responses to checkpoint therapy. We view this as a necessary foundational step toward the long-term goal of translating microbiota-derived antigens into therapeutic strategies.

2) It is remarkable that commensal specific Th17 cells become antitumorigenic in mice. However, it

remains unclear what the mechanism of action of aPD-1 is in this context. What is the cellular target? Th17 cells? It also remains unclear what the mechanisms are that maintain this “long-term” anti-tumor immunity. These aspects should be further investigated.

We can only speculate as to how anti-PD-1 synergizes with the commensal-induced Th17 cells. Our data suggest that the gut-primed intratumoral CD4⁺ T cells not only remodel the TME, enhancing the recruitment, priming, and expansion of tumor-specific CD8⁺ T cells, but may also induce PD-L1 expression on cancer and stromal cells through IFN- γ production. Thus, PD-1 blockade likely relieves the local inhibition (and potential exhaustion) of the CD8⁺ T cells (and possibly also CD4⁺ T cells). To reflect these mechanistic possibilities, we have updated our working model figure (**Extended Data Fig. 11**) to highlight how PD-1 blockade may act at multiple levels. It is possible that the PD-1 blockade additionally facilitates the conversion of commensal-specific Th17 cells into inflammatory Th1-like cells, but this process cannot be evaluated with currently available tools.

3) Finally, data from humans may be necessary to strengthen the relevance of this study, especially considering that such a beautiful and simple SFB-Th17 cell relationship is not present in humans (PMID: 26411289). Are intestinal derived, or even better commensal-specific Th17 cells found in melanoma (or any human tumors)? Is their presence predictive of aPD1 response?

While direct evidence linking commensal-specific Th17 cells to improved aPD-1 response in humans is currently lacking, recent studies have reported the presence of IL-17-producing CD4⁺ T cells in human tumors, including melanoma, colorectal, and lung cancers (PMID: 34944746, 37525015, 32158594, 19879162, 23159950, 21304053, 30800130, 23083809, 18354038, 28289713, 34083422, 39465401). In melanoma, for example, IL-17-producing CD4⁺ T cells have been associated with both pro- and anti-tumor roles, depending on the context. Moreover, transcriptional profiling of tumor-infiltrating CD4⁺ T cells in various human cancers, including bladder and head and neck carcinomas, has identified subsets with cytotoxic and pro-inflammatory gene signatures resembling the Th1-like phenotype described in our study (PMID: 32497499, 33637530, 39242276). Given our data showing that murine SFB-specific Th17 cells can transdifferentiate into Th1-like effector cells that promote anti-tumor immunity, it is plausible that similar functional plasticity exists within human commensal-specific Th17 populations. Our findings in this model provide important mechanistic framework, but further studies are needed to establish the presence, phenotype, and functional relevance of commensal-specific Th17 cells in human cancer immunity.

Additional specific comments:

Fig.1

The experimental plan is logical, and the results presented here are strong. I simply suggest that the authors specify in the legend how many times the experiments, shown in e and h, have been repeated. It would be ideal to test this approach using a different type of cancer cell line in an another orthotopic tumor mouse model. It is known that the tissue matters in regard to the tumor microenvironment. Alternatively the authors should revise their conclusions restricting them to melanoma. Finally, they should also highlight the caveat of using this mouse model.

We have now specified the number of independent replicates for all relevant experiments. Specifically, the experiments presented in panels (e) and (h) were each independently repeated at least three times with consistent results, and this information has been included in the revised figure legends.

To address the broader applicability of our findings, we agree that extending the study beyond the B16-F10 melanoma model is essential. While our primary mechanistic work was conducted in B16-F10 tumors, we have now expanded our analysis to two additional syngeneic tumor models: Lewis Lung Carcinoma (LLC) and MC38 colon adenocarcinoma. In both models, we engineered the tumor cells to express the SFB-3340 antigen and examined the impact of SFB colonization on anti-PD-1 responsiveness. Consistent with our findings in B16-3340 tumors, SFB colonization significantly enhanced anti-tumor responses in both LLC-3340 and MC38-3340 models. These responses were characterized by increased CD8⁺ T cell infiltration, reduced Treg frequencies, and enrichment of IFN- γ -producing CD4⁺ T cells within the tumor microenvironment. These new data further support the generalizability of our model across tumor types and are included in the revised manuscript (**Extended Data Fig. 1g-j and Extended Data Fig. 3e-h**).

We also acknowledge the limitations inherent to the B16-F10 model, including its rapid growth kinetics and immunologically "cold" baseline status. While these features make it useful for studying checkpoint-based immune modulation, they may not fully recapitulate human tumor complexity.

Fig.2

The characterization of the TME is well done. The results are clear and the data showing the role of both CD4 and CD8 T cells are of interest.

However, to support the conclusions on Th17 cells, it is important to test their specific role by depleting these cells or even better their capacity to produce IFN or Tbet.

Thank you for this valuable suggestion regarding the functional role of Th17 cells in the tumor microenvironment. We fully agree that testing their contribution is critical to strengthen our mechanistic conclusions.

In the revised manuscript, we now include new data using IL-17a-Cre \times LSL-DTA mice to conditionally ablate IL-17A⁺ Th17 cells (new **Figure 5**). These experiments reveal that SFB-induced Th17 cells are essential for the therapeutic efficacy of PD-1 blockade.

Specifically, in SFB⁺ mice treated with anti PD-1, deletion of IL-17A⁺ cells (DTA-ON ^{Δ IL-17A}) significantly impaired tumor control compared to littermate LSL-DTA controls (**Figure 5a–b**). This loss led to reduced frequencies and functionality of SFB-specific CD4⁺ T cells in both the gut and tumor, including a marked decrease in IFN- γ -producing Th1-like effectors (**Figure 5c–d**). Tumors from DTA-ON ^{Δ IL-17A} mice also exhibited lower CD8⁺ T cell infiltration, increased Foxp3⁺ Tregs, and diminished CD8⁺ effector function (**Figure 5e–f, Extended Data Fig. 8**). These findings provide direct genetic evidence that IL-17A⁺ Th17 cells are required for orchestrating microbiota-driven CD4⁺–CD8⁺ coordination and enhancing PD-1 checkpoint therapy.

Fig.3

This is a deeper characterization of the TME in relation to the intestine using the scRNA and TCRseq. The way the data in d are shown is brilliant.

Thank you for your encouraging feedback on our scRNA-seq and TCR sequencing data. We are pleased that the clonal relationships and migratory patterns illustrated, particularly in panel (d), effectively convey the gut–tumor immune axis that we aim to highlight.

Fig.4

I believe that using photoconvertible mice (PMID: 34099917) is fundamental to directly prove the intestinal origin of the tumor-infiltrating Th17 cells, especially considering the transgenic expression of

the relevant antigen is outside the intestine. At the moment, beside TCR sharing which remains an interesting correlation, the conclusions are partially based on the observation that these cells are found in the intestine and in the tumors, but not in the dLN (Extended Fig. 7).

We agree that photoconvertible mice represent a valuable tool for tracking cell migration. However, their application in gut tissues poses challenges due to the high vascularization, which increases the risk of non-specific labeling of circulating cells and complicates interpretation.

To address these concerns and more directly trace the intestinal origin of tumor-infiltrating ex-Th17 cells, we employed IL-17A fate-mapping under both endogenous (**Figure 4a-c**) and TCR transgenic conditions (**Figure 4d-f**). These experiments allowed us to track IL-17A-expressing cells from the gut and/or its draining mesenteric lymph nodes to the tumor and characterize their trans-differentiation into Th1-like effectors. In parallel, our scTCR-seq clonal tracking (**Figure 3**) further substantiates the gut-tumor lineage connection by revealing shared clonotypes between gut-resident Th17/Tfh cells and tumor-infiltrating Th1-like CD4⁺ cells.

Together, these complementary approaches offer robust evidence for the migration and functional reprogramming of gut-derived Th17 cells in shaping the tumor immune microenvironment.

Referee #3 (Remarks to the Author):

Najar et al. present an elegant and simple interrogation of the relationship between T cell responses formed against commensals and anti-tumor immunity. In their study they use a well described and studies SFB derived CD4 epitope and overexpress is in B16F10 tumor cells. They subsequently carefully analyze the anti-tumor immune response in SFB- and SFB+ mice. Using scRNAseq, TCR clonotype analysis and fate mapping they identify that in SFB+ mice Th17 T cells from the gut re-differentiate and acquire a an Th1 T cell phenotype. In contrast clonotypes in tumors of SFB- mice are found in Tregs. While the study has some major shortcoming, there is value in the careful analysis of the T cell response.

Conceptional shortcomings:

As the authors state in their introduction a previous study showed a clear correlation between Staph epi – OVA colonization and increased anti-tumor immunity. This significantly reduces the novelty. While the authors could make an argument for gut commensals the fast that SFB is introduced shortly before tumor inoculation further dampens enthusiasm. Given the SFB was described as a stable commensal why not use a SFB+ colony which has life long SFB exposure. Timing here might be very important. While the reviewer sees value in interrogation of gut commensal, tumor T cell cross-talk at baseline the current model resembles more a acute vaccination setting. Nonetheless, for completion the authors should also consider conducting an experiment with therapeutic SFB inoculation.

We thank the reviewer for the valuable comments. We acknowledge prior work showing that skin commensals such as *Staphylococcus epidermidis* can modulate tumor immunity. However, our study specifically focuses on gut-resident commensals, which represent a distinct immunological environment with direct clinical relevance, given the well-established associations between gut microbiota composition and immune checkpoint blockade efficacy in cancer patients.

A key gap in earlier studies was the lack of direct evidence that immune cells primed by bacterial colonization migrate from their site of induction to distant tumors and actively mediate tumor control.

Our work addresses this by demonstrating that SFB-specific Th17 cells, induced in the gut, traffic to tumors where they transdifferentiate into pro-inflammatory Th1-like cells and directly suppress tumor growth. This provides critical mechanistic insight into how gut microbiota drive tumor-specific immunity through antigen mimicry, thereby reinforcing and extending clinical observations linking microbiota composition to ICB responsiveness.

Regarding the timing of SFB exposure, we appreciate the reviewer's concern. To address this directly, we repeated the B16-3340 tumor experiments using mice obtained from Taconic Farms that are chronically colonized with SFB (i.e., SFB-positive from early life rather than by recent gavage). In this independent cohort ($n = 10$ mice per group), Taconic SFB⁺ mice were implanted subcutaneously with either B16-EV or B16-3340 and treated with anti-PD-1 regimen exactly the same as done with acutely colonized mice. As shown in the figure below (Fig. R2), tumors expressing the SFB-3340 mimic (B16-3340) were significantly better controlled following anti-PD-1 therapy compared with B16-EV controls. The individual mouse growth traces (lower panels) further illustrate that the effect is driven by epitope-dependent tumor control rather than outlier animals. Thus, lifelong (chronic) SFB exposure recapitulates the SFB-dependent enhancement of anti-PD-1 efficacy that we report using our gavage model, indicating that the phenomenon is not an artefact of recent colonization timing.

Fig. R2: Chronic SFB colonization (Taconic mice) recapitulates epitope-dependent enhancement of anti-PD-1 therapy. Top: Caliper measurements are shown as mean tumor growth (\pm s.e.m.) for Taconic SFB⁺ mice implanted subcutaneously with B16-EV (blue circles) or B16-3340 (orange squares) and treated with anti-PD-1 antibody according to the regimen described in Methods ($n = 10$ mice per group). Statistical comparison was performed by two-way ANOVA with repeated measures ($p < 0.0001$). Bottom left and right: Individual tumor growth trajectories for mice in the B16-EV (bottom left) and B16-3340 (bottom right) cohorts, respectively. Data show that, in mice with lifelong SFB exposure, anti-PD-1 therapy produces substantially greater tumor control when the tumor expresses the cognate SFB-3340 epitope, mirroring the results obtained with our SFB gavage model. See Methods for mouse source, confirmation of SFB status (vendor health report and fecal PCR), tumor implantation, antibody dosing schedule and statistical tests.

We also agree that evaluating therapeutic SFB administration can be a valuable translational test of our molecular-mimicry model. To address this, we performed new experiments to assess whether therapeutic SFB administration could rescue anti-PD-1 responses in mice bearing established SFB-3340 tumors. As shown in **Extended Data Fig. 1e–f**, all mice were implanted with B16-3340 tumors on day 0 and treated with anti-PD-1 during the early palpable window (days 4–10). Separate cohorts were gavaged with SFB at defined post-implantation intervals:

- Group 1: days 8–12
- Group 2: days 12–16
- Group 3: days 15–19

- Group 4: no SFB (control)

Early post-implantation SFB administration (Group 1) yielded the most significant therapeutic benefit, with markedly reduced tumor growth compared to both SFB-free controls and mice receiving later SFB gavage. Importantly, the therapeutic effect declined progressively with delayed administration. These data demonstrate that: (i) expression of the SFB-derived neoantigen by the tumor is required for microbiota-dependent augmentation of anti-PD-1 efficacy in this model, and (ii) there is a defined post-implantation window, overlapping with early PD-1 blockade, during which microbial antigen exposure most effectively synergizes with therapy. Together, these findings support a model in which timely, microbe-driven priming and recruitment of antigen-specific CD4⁺ T cells synergizes with PD-1 blockade to promote durable anti-tumor immunity. This highlights the translational potential of therapeutic microbiota modulation to enhance checkpoint responsiveness, provided it occurs within an optimal therapeutic window. We have added these data to **Extended Data Fig. 1e–f** and included a brief discussion of their implications in the revised manuscript.

Major issues:

The entirety of the observation is based on one cancer cell line and one bacterial strain. While the latter could be rationally tested, the authors should test at least 3 more cell lines to ensure generalizability. For the lack of several cell lines it should be carefully discussed how other bacterial strains might affect these dynamics.

We appreciate the reviewer's concern regarding generalizability and the need to assess additional bacterial strains and tumor models. In the revised manuscript, we have now expanded our analyses to directly address these points.

First, to determine whether mucosal commensals other than SFB can similarly augment PD-1 blockade in an antigen-dependent manner, we tested colonization with *Helicobacter hepaticus* (Hh), which is known to predominantly elicit regulatory T cell responses in the large intestine. Using an Hh-specific neoantigen mimicry platform, we found that although Hh colonization led to a robust expansion of antigen-specific CD4⁺ T cells in both the gut and tumors, these cells largely retained a Foxp3⁺ regulatory lineage and failed to acquire a Th1-like effector program. Consistently, Hh-specific T cells produced little to no IFN- γ , did not enhance intratumoral CD8⁺ responses, and ultimately failed to improve tumor control in the setting of PD-1 blockade (**Extended Data Figs. 9-10**). These findings demonstrate that the impact of commensals on tumor immunity is not uniform, but rather shaped by the immune differentiation programs they induce in the gut.

Second, to extend beyond the B16 melanoma model, we tested two additional syngeneic tumor models, Lewis lung carcinoma (LLC1) and MC38 colon adenocarcinoma, engineered to express the immunodominant SFB-3340 epitope. In both models, SFB colonization significantly delayed tumor growth and enhanced responsiveness to PD-1 therapy compared to SFB⁻ controls (**Extended Data Fig. 11**). These results establish that the microbiota-tumor antigen synergy we describe is not restricted to melanoma but extends across distinct tumor types.

Together, these new experiments directly address the reviewer's concern by demonstrating (i) that not all commensals support checkpoint efficacy, Hh, for example, expands tumor-trafficking Tregs that fail to promote tumor immunity, and (ii) that SFB-driven antigenic mimicry can enhance PD-1 blockade across multiple tumor models. These findings highlight both the specificity and generalizability of our observations, while also underscoring the importance of bacterial strain-dependent immune programming in shaping tumor responses to immunotherapy.

The authors based their entire framework on the fact that cancer cells acquire a shared peptide with bacteria. But translationally there seems to be little evidence of this. Thus, it would be important to include a SFB expressing GP100 for instance and determine whether similar observations can be made with a tumor associated self-antigen.

Our study was designed to elucidate fundamental principles of antigenic mimicry between commensal microbes and tumors, with the broader goal of understanding how peripheral priming, especially, in the gut, outside the immunosuppressive tumor microenvironment, can generate T cells that are functionally competent upon tumor entry.

To this end, we selected SFB-3340, a commensal-derived antigen that elicited only a modest immune response locally within the tumor microenvironment. This allowed us to establish a proof-of-principle that microbiota-specific T cells can be leveraged to enhance anti-tumor immunity independent of classical immunodominant tumor-associated antigens. Importantly, this strategy illustrates how T cells, conditioned in a non-inhibitory mucosal environment, may retain effector potential and resist exhaustion when they encounter the immunosuppressive cues of the tumor microenvironment.

We agree that engineering SFB to express a tumor-associated self-antigen such as GP100 would provide a compelling test of whether commensal-specific T cells can augment anti-tumor responses beyond bacterial mimicry. However, because SFB is not culturable and cannot currently be genetically modified, this approach is not technically feasible. We therefore emphasize in the revised manuscript that future studies employing alternative microbial platforms or synthetic systems will be essential to extend this strategy to tumor-associated self-antigens and to further explore its therapeutic potential.

Minor issues:

The authors elegantly show that combination of SFB + anti-PD-1 can induced robust anti-tumor immunity and that a protective memory is formed. Given their fate mapping it would be interesting to determine whether Th1 T cells form memory or disappear and only a th17 gut population is maintained which subsequently aids a CD8 recall response.

We thank the reviewer for this insightful suggestion. We agree that delineating the precise nature of memory responses in the context of SFB colonization and checkpoint blockade is an interesting question. Our current fate-mapping strategy allowed us to track the differentiation trajectory of IL-17A-derived cells and demonstrate the presence of a protective recall response; however, definitively resolving whether Th1-like cells themselves persist as long-lived memory populations, or whether Th17 cells are preferentially maintained and subsequently contribute to CD8⁺ T cell recall responses, remains technically challenging. One limitation is the rarity of memory CD4⁺ T cells in this system, which makes their detection and longitudinal tracking inherently difficult. We agree that this represents a key direction for further investigation, as it will clarify how distinct CD4⁺ T cell lineages contribute to the durability and breadth of commensal-augmented anti-tumor immunity.

Response to Reviewers' Comments.

Microbiota-induced plastic T cells enhance immune control of antigen-sharing tumors

Najar et al.

We thank the editors for the opportunity to respond to the reviewers' comments. In each case, our response is shown in "blue text" below the reviewer comment/question.

Referee #1 (Remarks to the Author):

In this revised study, the authors investigated whether molecular mimicry between a commensal microbial antigen and a tumor antigen would be beneficial for anti-tumor immunity and response to immune checkpoint blockade. The authors engineered a tumor cell line (B16-F10 melanoma) to express an antigen (the 3340 epitope) from the commensal bacteria segmented filamentous bacteria (SFB). They then performed a series of elegant experiments to show that the presence of SFB significantly enhanced the efficacy of anti-PD-1 immune checkpoint therapy. In the revised manuscript, they tested two additional tumor cell lines (Lewis lung carcinoma and MC38 colon adenocarcinoma) that they also engineered to express the SFB epitope 3340 and confirmed that the presence of SFB also led to significantly delayed tumor growth. They show that SFB colonization induces a homeostatic Th17 response in the small intestine, which then traffic to the tumor and convert to Th1-like cells with production of IFN-g and TNF-a within the tumor microenvironment. Alterations in the tumor microenvironment also led to increased recruitment, expansion and effector functions of tumor-specific CD8+ T cells. In the revised manuscript, the authors now also show that depletion of these Th17 cells led to loss of anti-PD-1 mediated tumor control, showing the requirement for SFB commensal induction of Th17 cells. In the revision, the authors also tested the therapeutic potential of adding SFB at the same time as anti-PD-1 therapy, or various timepoints after the initiation of the treatment, and showed that gavage with SFB increased efficacy of anti-PD-1 even when given as an adjuvant therapy, with the best effect achieved when given early. Finally, the authors tested the ability of a different bacteria (*Helicobacter hepaticus*) to increase ICB when the B16-F10 tumor cells expressed the Hh7-2 epitope. While *H. hepaticus* induced antigen-specific CD4+ T cells in the gut and these cells trafficked to distal tumors, they did undergo Th1-like effector conversion and failed to enhance ICB.

In this revised manuscript the authors adequately addressed all of my comments and performed several complex new experiments that provide additional insights into the ability of commensal bacteria to drive a CD4 T cell response that promotes ICB efficacy. I have no additional concerns and commend the authors for such a detailed response to reviews.

We thank the reviewer for the supportive comments.

Referee #2 (Remarks to the Author):

I appreciate the new experiments performed overall, but believe that some aspects (see below) still need to either be experimentally addressed or the conclusions toned down.

We thank the reviewer for the constructive evaluation of our manuscript and the thoughtful suggestions, which have helped us further clarify the mechanistic scope and conceptual framing of our work.

General points:

1) "... not as a literal model of naturally occurring peptide sharing." I agree with the authors that this manuscript does not address molecular mimicry. It remains very much possible that microbiota specific Th17 cells do not respond to tumour antigens. I therefore believed that the "molecular mimicry" concept should not be part of this manuscript because it has not been tested and can even be confusing. I

suggest the author focus more on their following concept: "...delivered through gut-resident bacteria, can be harnessed to stimulate durable, tumour-specific immunity and improve responses to checkpoint therapy."

We agree that our study does not evaluate molecular mimicry in the strict biological sense of naturally occurring antigenic or peptide homology between microbial and tumor antigens. Our inclusion of the term "molecular mimicry" was intended solely to contextualize the prior hypothesis regarding antigenic overlap as one potential, but untested, mechanisms linking the microbiota-tumor immune interactions in the context of ICB-responsive cancers.

Importantly, as noted by the reviewer, our study employs a synthetic neoantigen mimicry system, an artificially engineered platform designed to precisely control antigen sharing between the gut commensal SFB and tumor cells, to model and study mechanistically the immunologic influence of gut commensal microbes on the outcome of anti-PD-1 therapy. This deliberate design allowed us to precisely dissect how a defined intestinal microbe can condition systemic immunity and modulate ICB efficacy in a controlled, mechanistic manner. Such rationally designed models are important proof-of-principle approaches to dissect the mechanistic contribution of microbiota-mediated antigen experience without relying on natural epitope overlap. Nevertheless, to prevent confusion, we have revised the manuscript to soften the term "molecular mimicry" to "antigenic mimicry" and explicitly state that our model focused on engineered neoantigen systems rather than natural endogenous antigenic homology.

2) The authors have speculated on the mechanistic actions of exTh17 cells in remodelling the tumour microenvironment in response to aPD1. However, this has not been addressed experimentally.

We agree with the reviewer that we do not investigate the mechanism by which the exTh17 cells influence the tumor microenvironment, although we provide experimental evidence that recruitment of these T cells profoundly remodels the tumor microenvironment to favor effective anti-tumor immunity.. We have focused on how the exTh17 cells affect the overall T cell response in the context of anti-PD-1 therapy. A full understanding of how the exTh17 cells achieve the successful outcome will require an extensive follow-up study to examine which cytokines and chemoattractants are required, which target cells are involved beyond the required CD8+ T cells (e.g. macrophages, fibroblasts), etc. We feel that such further mechanistic work will be important, particularly in the context of commensal microbes that differ in their abilities to promote anti-tumor responses, but is beyond the scope of the current study. We clarify that future studies using conditional lineage-specific deletion or cytokine perturbation models are needed to define the precise molecular mechanisms by which exTh17 cells modulate CD8⁺ TIL function and regulatory T cell dynamics.

3) Human relevance. This point has not been experimentally addressed. I believe that this is a partial limitation and it should at least be discussed in the manuscript.

This is an important point, and we have added in the Discussion a statement as to the limitation of our study which, nevertheless, is an important step toward leveraging gut microbiota to improve cancer immunotherapy outcomes.

Specific points

Fig.1

I would suggest showing either SD or SEM, but not a mix of the two, as, for example, in e and g. This should be applied throughout the entire manuscript.

We thank the reviewer for noting the inconsistency in data presentation. We have standardized all graphical representations throughout the manuscript to display **mean ± s.d.**, ensuring uniformity across figures and extended data.

Fig.2

The use of the DTA-ONDIL-17A is very well received. However, the authors should be cautious with their direct conclusions on IL-17A Th17 cells, since this construct is not specific for Th17 cells but affects all IL-17A-producing cells even before the tumor is inoculated. I would simply acknowledge this aspect and discuss why it is very likely that the effect is due to SFB Th17 cells.

The reviewer makes a very important point, as this system ablates all IL-17A producing cells and does not exclusively target Th17 cells. However, Th17 cells are the major source of IL-17A in adult C57BL/6 mice, particularly under steady-state conditions. Importantly, SFB is a well-established and selective inducer of CD4⁺ Th17 cells in C57BL/6 mice, while innate IL-17A producing subsets, such as $\gamma\delta$ T cells and innate lymphoid cells, are not significantly induced by SFB in the steady state. Thus, the phenotypes observed align most closely with selective depletion of the SFB-driven Th17 lineage. Our results, together with extensive prior knowledge of SFB immunobiology and IL-17A cell sources, strongly suggest that the effects are predominantly mediated by induced Th17 cells rather than innate IL-17A producers.

Fig.4

I disagree that the tools used here allow the authors to “more directly trace the intestinal origin of tumour-infiltrating ex-Th17 cells”. I believe that the use of mouse models with photoconvertible cells is a more direct approach. Nevertheless, based on all that is known about SFB Th17 cells, it is very likely that the ones found in the tumour come from the intestine and/or MALTs. Yet, there is no direct experiment testing this migration path in this manuscript, and it cannot be formally excluded that SFB Th17 cells, in a mouse model where tumour cells express SFB antigens, are primed outside the MALTs. So, if the photoconversion experiment is not going to be performed, this aspect should at least be addressed in the discussion, clearly stating the reasons why it is very plausible for YFP Th17 cells to be derived from the MALTs.

We thank the reviewer for this thoughtful comment and fully agree that the IL-17A-driven *tdTomato* fate-mapping system marks cells of the Th17 lineage but does not, by itself, provide direct spatial evidence of intestinal-to-tumor migration. To address this, we complemented our fate-mapping approach with single-cell TCR sequencing and lineage tracing analysis, which revealed shared clonotypes between intestinal IL-17A⁺ CD4⁺ T cells and tumor-infiltrating Th1-like CD4⁺ T cells. These shared TCR signatures along with dual IL-17A-driven *tdTomato* fate-mapping strongly support the interpretation that the gut-derived Th17 lineage cells migrate to the tumor, where they acquire a Th1-like effector phenotype under ICB conditions.

We also recognize that more direct cell-tracking tools, such as photoactivation-based assays, can theoretically delineate cell trafficking from the gut to peripheral tissues. However, these approaches have notable caveats: photoactivation can label circulating cells, resulting in false-positive detection, and the invasive surgical procedures required to expose the intestinal mucosa can perturb tissue integrity and alter host–microbiota interactions. Such disruptions would confound the study of how commensal microbes, particularly SFB, shape systemic T cell function and ICB efficacy. Our chosen combination of fate mapping and single-cell TCR lineage tracing thus provides a robust and minimally disruptive method to infer physiological cell migration consistent with prior studies using IL-17A reporter systems.

Overall, we maintain that our integrated strategy effectively links SFB-induced Th17 lineage imprinting in the intestine to the expansion of clonally related Th1-like effector T cells in tumors, while minimizing experimental artifacts associated with more invasive migration-tracking approaches.

Ext data Fig.3

I believe there is no need to show a statistic in the presence of two replicates as was done in A. The results are clear based on these few replicates and the stats do not add anything.

We agree and have removed statistical tests where only two replicates were available, retaining the blots for visual representation only.

Ext Data Fig.4 and 5.

I would include in the manuscript the validation of the efficacy of depletion using aCD4 and aCD8 mAb.

We thank the reviewer for this suggestion. We have now included additional panels in Extended Data Fig. 4 showing validation of CD4⁺ and CD8⁺ T cell depletion efficiency following antibody treatment, confirming the effectiveness of the depletion strategy.

Discussion

See above about molecular mimicry. What is the point to start the discussion with this topic?

Finally there is no ref 52 reported in the reference list.

Again, we thank the reviewer for raising this important point and have already responded in raised comment 1.

We have also corrected the missing reference 52 and verified all other citations.

Referee #3 (Remarks to the Author):

The authors have sufficiently addressed my concerns.

We thank the reviewer for the positive feedback.